

# Rank $Q$ E-string on spheres with flux

**Chiung Hwang[1*], Shlomo S. Razamat[2,3], Evyatar Sabag[2] and Matteo Sacchi[4]**

**1** Department of Applied Mathematics and Theoretical Physics,
University of Cambridge, Cambridge CB3 0WA, UK
**2** Department of Physics, Technion, Haifa, 32000, Israel
**3** School of Natural Sciences, Institute for Advanced Study, Princeton NJ, USA
**4** Dipartimento di Fisica, Università di Milano-Bicocca & INFN,
Sezione di Milano-Bicocca, I-20126 Milano, Italy

⋆ ch911@cam.ac.uk

## Abstract

We consider compactifications of rank $Q$ E-string theory on a genus zero surface with no punctures but with flux for various subgroups of the $E_8 \times SU(2)$ global symmetry group of the six dimensional theory. We first construct a simple Wess–Zumino model in four dimensions corresponding to the compactification on a sphere with one puncture and a particular value of flux, the cap model. Using this theory and theories corresponding to two punctured spheres with flux, one can obtain a large number of models corresponding to spheres with a variety of fluxes. These models exhibit interesting IR enhancements of global symmetry as well as duality properties. As an example we will show that constructing sphere models associated to specific fluxes related by an action of the Weyl group of $E_8$ leads to the S-confinement duality of the $USp(2Q)$ gauge theory with six fundamentals and a traceless antisymmetric field. Finally, we show that the theories we discuss possess an $SU(2)_{ISO}$ symmetry in four dimensions that can be naturally identified with the isometry of the two-sphere. We give evidence in favor of this identification by computing the 't Hooft anomalies of the $SU(2)_{ISO}$ in 4d and comparing them with the predicted anomalies from 6d.

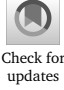

# 1   Introduction and Summary

It is possible to construct a vast landscape of 4d Quantum Field Theories by considering a 6d CFT on a Riemann surface. We can view such a construction as an RG flow across dimensions. Such a flow might have an IR dual description directly in 4d, that is there might exist a 4d weakly coupled theory which flows to the same QFT in the IR. By now there are many examples of such dualities across dimensions (for a list of examples see *e.g.* [1–17]). In certain cases there are more than one 4d dual for the same compactification of a 6d CFT. Compiling a dictionary between 4d Lagrangians and 6d compactifications can shed some light on strong coupling dynamics of the 4d theories as, at least some of, its features can be encoded in the geometric properties of the compactification procedure.

In this note we extend an example of such a dictionary and discuss some of the physical implications that can be derived from it. We consider compactifications of the rank $Q$ E-string theory. This is a 6d $(1,0)$ SCFT obtained for example by considering $Q$ M5 branes probing the "end-of-the-world" M9 brane in M-theory. Compactifications of these models on two punctured spheres with particular value of flux and on tori were studied in [8,18]. The relevant 4d models turn out to be rather interesting. For example these $\mathcal{N}=1$ models in 4d satisfy certain self-duality and emergence of symmetry properties which upon dimensional reduction to 3d lead to [18,19] some of the canonical instances of $\mathcal{N}=4$ mirror symmetry [20]. Here we extend the results to compactifications on a two-sphere. We also turn on a non-trivial flux supported on the sphere for various abelian subgroups of the $E_8 \times \mathrm{SU}(2)$ global symmetry of the 6d SCFT.

A two-sphere is a special Riemann surface. First, as it does not have complex structure moduli and does not have non trivial cycles around which we can define non-trivial holonomies for background gauge fields, we do not expect the 4d theories resulting in compactifications on a two sphere (with generic values of fluxes) to have any exactly marginal deformation. Second, the two-sphere has a non-trivial compact isometry group $\mathrm{SU}(2)_{\mathrm{ISO}}$[1]. An interesting question is whether this geometric symmetry can be seen as a symmetry of the 4d theory[2]. By directly studying the compactifications of the rank $Q$ E-string theory we will indeed see that this is the case.

The derivation of the theories corresponding to compactifications on a two sphere proceeds first by understanding what is the theory obtained by compactifying on a disc (*a.k.a* one punctured sphere or a cap) with some value of flux and then gluing such discs together with the tubes of [18] to form a closed surface, see Figure 1. We derive the model corresponding to a disc by starting with a two punctured sphere and closing one of the two punctures. Field

---

[1]The details of the global structure of the group will be not important to us so we will be cavalier with it.

[2]Of course one can think about other symmetries of 6d SCFTs geometrically by imbedding the SCFTs in string theoretic constructions.

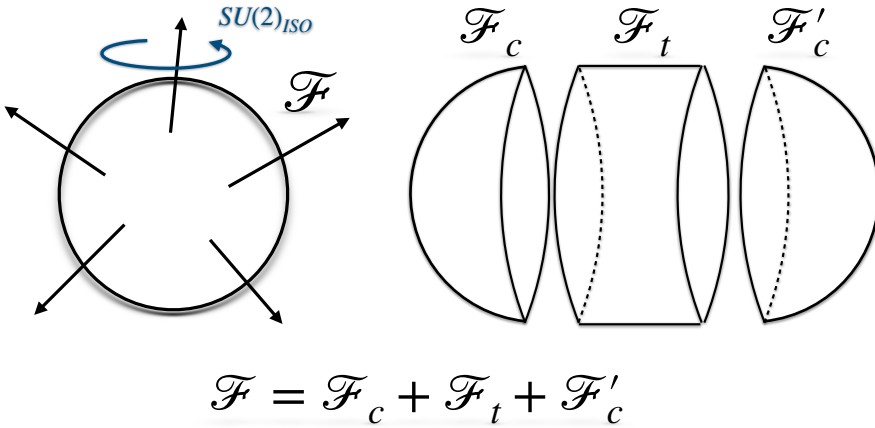

$$\mathscr{F} = \mathscr{F}_c + \mathscr{F}_t + \mathscr{F}'_c$$

Figure 1: The theory corresponding to a sphere with flux can be constructed by combining together theories corresponding to one punctured spheres (caps) and two punctured spheres (tubes). The flux associated to the resulting theory is the sum of fluxes of the components. In particular different ways to split the flux into components might lead to different looking Lagrangians which have to be dual to each other. Moreover different values of flux related by the Weyl group of the 6d symmetry lead again to equivalent, IR dual, theories. The flux of the components can break the symmetry of the 6d theory to smaller sub-group than the combined theory, leading to Lagrangians with emergent symmetry in the IR. The compactification surface has an $SU(2)_{\text{ISO}}$ isometry symmetry which becomes a global symmetry of the 4d model.

theoretically closing a puncture corresponds to turning on a sequence of vacuum expectation values for operators charged under the symmetry of the puncture and adding certain gauge invariant flipping fields. The exact procedure is deduced by matching the anomalies of the resulting theory to the expectations from 6d. We perform this analysis in Section 2. Moreover, the correct identification of the cap theory will entail IR emergence of various non-abelian structures of the global symmetry once we glue the caps and tubes together and fine tune the flux to get interesting symmetries.

We then construct several examples of theories corresponding to spheres with a variety of fluxes in Section 3. The models we obtain are expected to have symmetries determined by the fluxes which are bigger than the symmetry of the ingredients. One example we will discuss is a sphere with $E_6 \times SU(2) \times U(1)^2$ preserving flux. The Lagrangian for this theory is a $USp(2Q)$ gauge theory with a traceless antisymmetric chiral field, an octet of fundemantal fields, and a collection of gauge singlet fields. The emergence of the $E_6$ symmetry is related to the self-duality of this model, as it was previously discussed in [21, 22]. The construction of this paper is thus a geometric interpretation of the pertinent self-duality. In this example, and in others, we will also observe the emergence of the geometric $SU(2)_{\text{ISO}}$ symmetry: in fact this symmetry appears explicitly in the UV Lagrangian. We compute the various 't Hooft anomalies of $SU(2)_{\text{ISO}}$ from our 4d field theoretic description and (using the Bott-Cattaneo formula [23] in Appendix B) also from the 6d anomaly polynomial showing that these two computations agree.

We will also consider a sphere model preserving $SO(14) \times U(1)^2$ symmetry with a particular value of flux for the two abelian factors. It will turn out that this model can be engineered in two different ways which are equivalent from the 6d perspective. The two ways correspond to fluxes which are related by the action of the Weyl group of the $E_8$ global symmetry of the 6d theory and are thus equivalent. In one of the two ways the Lagrangian turns out to be a $USp(2Q)$ theory with a traceless antisymmetric chiral field, a sextet of fundamentals, and gauge

singlets. In the second way the theory is a WZ model. The IR equivalence of the two models follows from a known example of S-confinement due to Csaki, Schmaltz, and Skiba [24]. Thus our derivation is a geometric reinterpretation of this instance of S-confinement. In this degenerate case although the Cartan generator of the geometric $SU(2)_{ISO}$ symmetry can be identified in the UV Lagrangian, it does not act faithfully in the IR. This is consistent with all the anomalies involving $SU(2)_{ISO}$ vanishing in the 6d computation.

Several appendices detail some of the background material and additional computations. In particular in Appendix C we discuss the $SU(2)_{ISO}$ global symmetry in $\mathbb{S}^2$ $\mathcal{N} = 1$ $A_1$ class $\mathcal{S}$ compactifications. These are very simple 4d Lagrangian theories which exhibit the appearance of this geometric symmetry.

Let us comment here on various ways in which our results can be used and/or extended. It will be interesting for example to understand the emergence of the $SU(2)_{ISO}$ symmetry in other examples of compactifications with known tube theories, such as class $(G, G)$ ($G \in ADE$) conformal matter [12]. Moreover, as we will see the Cartan of $SU(2)_{ISO}$ is a symmetry of the two punctured sphere. This symmetry is broken by anomalies/superpotentials to a discrete symmetry on a torus, the order of which depends on the flux. It will be very interesting to understand the 6d origin of this discrete symmetry. It will be also interesting to study the enhancement of global symmetry for sphere theories in class $(G, G)$ ($G \in ADE$) conformal matter [12]. Studying various geometric constructions of the sort we discuss here often Seiberg dualities (and their generalizations) with special unitary and symplectic groups naturally appear. It would be interesting to understand in what scenarios dualities with orthogonal groups[3] and generalizations of the special unitary dualities with adjoint fields [26–29] make their appearance. Yet another question is to understand the 3d reductions of the sphere theories and in particular whether they have mirror symmetry duals (See for example [30]).

## 2 Derivation of the basic cap model

In this section we are going to construct the theory corresponding to the compactification of the rank $Q$ E-string theory on a one punctured sphere with some value of flux for the $E_8 \times SU(2)_L$ global symmetry,[4] which we shall call the *cap model*. This will be, together with the tube theory of [18], one of the fundamental building blocks using which we will construct theories corresponding to compactifications on spheres with fluxes in the next section.

The key idea to derive the cap model is to start from the theory obtained by compactification on a tube and completely close one of the two punctures. The way the closure of the puncture is implemented in field theory is analogous to what is usually done in class $\mathcal{S}$ theories [1]. In that context, each puncture carries a flavor symmetry and thanks to $\mathcal{N} = 2$ supersymmetry we have moment map operators that contain the conserved currents for such symmetries. The puncture can then be completely or partially closed by giving a nilpotent vev to the moment map operator, which breaks the global symmetry of the puncture to some subgroup [31, 32]. In our case, since we only have $\mathcal{N} = 1$ supersymmetry we don't have true moment map operators for the symmetry of the puncture, but we still have some operators that transform non-trivially under this symmetry and to which we can give a vev to break it. These operators can be identified with the 5d matter fields assigned Neumann boundary conditions at the puncture in the 5d effective description of the puncture, see *e.g.* [8]. The relevant 5d description of the rank $Q$ E-string is in terms of $USp(2Q)$ $\mathcal{N} = 1$ gauge theory with eight fundamental hypermultiplets as well as a hypermultiplet in traceless antisymmetric representation [33]. This matter content leads to a $USp(2Q)$ global symmetry associated to the

---

[3]Some special dualities with *Spin* groups have appeared in such constructions in [21, 25].

[4]For $Q = 1$ the global symmetry is just $E_8$ and the additional $SU(2)_L$ factor appears for $Q > 1$.

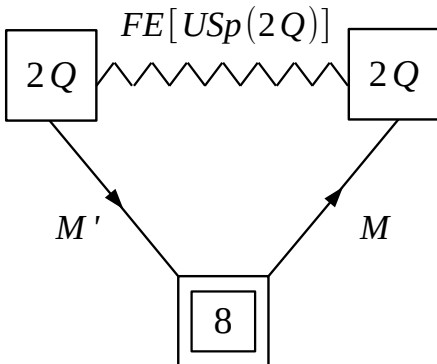

Figure 2: The basic tube theory. It is built from a quiver theory we denote by $FE[USp(2Q)]$ and two octets of fundamental fields charged under the two $USp(2Q)$ puncture symmetries. In this drawing simple square boxes represent $USp(2Q)$ flavor symmetries and the double square box represents an $SU(8)$ flavor symmetry. Moreover, straight lines denote standard 4d $\mathcal{N} = 1$ chiral fields $M^a$, $M'_a$ transforming under the symmetries of the nodes they are connecting, while the wiggle line denotes a more exotic type of matter which corresponds to the $FE[USp(2Q)]$ SCFT first introduced in [18] (see Appendix A for a review).

puncture in 4d and to an octet of $\mathcal{N} = 1$ chiral operators in the fundamental representation of this symmetry as well as an $\mathcal{N} = 1$ chiral operator in the traceless antisymmetric representation. With a little abuse of terminology, we will still refer to these operators as "moment maps".

Our starting point is thus the model obtained from the compactification of the rank $Q$ E-string theory on a tube with fluxes. This has been identified in [8] for $Q = 1$ and in [18] for arbitrary $Q$ in those cases in which we have fluxes only for the $E_8$ part of the 6d global symmetry. The simplest model is the one corresponding to flux $\mathcal{F}_{tube} = (0; \frac{1}{2}; 0, \cdots, 0)$[5] and more general tubes associated to different choices of flux can be derived by various gluings of several copies of this basic one [18]. This tube model is described by the quiver theory of Figure 2. The superpotential of the tube model is

$$\mathcal{W}_{tube} = \mathcal{W}_{FE[USp(2Q)]} + \sum_{a=1}^{8} \text{Tr}_x \, \text{Tr}_y \, M^a \Pi M'_a. \tag{1}$$

In this expression $\mathcal{W}_{FE[USp(2Q)]}$ is the superpotential of the $FE[USp(2Q)]$ theory defined in (A.1), $\Pi$ is an operator of $FE[USp(2Q)]$ transforming in the bifundamental representation

---

[5]Here we are using the convention in which the vector of fluxes takes the form $\mathcal{F} = (n_t; n_c; n_1, \cdots, n_8)$, where $n_t$ is the flux for the Cartan generator $U(1)_t \subset SU(2)_L$ while $n_c$, $n_a$ are the fluxes in the Cartan of $U(1)_c \times SU(8) \subset E_8$ with the constraint $\sum_{a=1}^{8} n_a = 0$. This is also the same basis for the flux vector in terms of which we write the anomalies predicted from 6d by integrating the 8-form anomaly polynomial over the Riemann surface in Appendix B. In the next section we will introduce a different basis which is defined considering a different subgroup, $SO(16) \subset E_8$, and which will be more suitable for studying the gluings of the fundamental building blocks we consider in this section. In the $SO(16)$ basis the fluxes are given by $n_a^{SO(16)} = n^{U(1)} + 2n_a^{SU(8)}$ for $a = 1 \cdots 8$. We normalize the fluxes so that legal fluxes on closed surfaces have $n_a^{SO(16)}$ integer. Although we will not discuss this here, certain fractional fluxes are also allowed due to the possibility of turning on *center fluxes*. The center fluxes break part of the rank of the global symmetry. See Appendix C of [8] for details. The basic tube of Figure 2 has half a unit of flux so that tori constructed from even number of these tubes have integer fluxes, while odd number of tubes will involve also center flux. This center flux breaks $E_7$ to a rank four subgroup, *e.g.* $F_4$ [8].

of the $\mathrm{USp}(2Q)_x \times \mathrm{USp}(2Q)_y$ symmetry and $\mathrm{Tr}_{x,y}$ denote traces over the $\mathrm{USp}(2Q)_{x,y}$ flavor symmetries[6].

The manifest non-anomalous global symmetry of the tube model is

$$\mathrm{USp}(2Q)_x \times \mathrm{USp}(2Q)_y \times \mathrm{U}(1)_c \times \mathrm{U}(1)_t \times \mathrm{U}(1)_f \times \mathrm{SU}(8)_u. \tag{2}$$

The two $\mathrm{USp}(2Q)$ symmetries are associated to the two punctures, while the rest, except for the $\mathrm{U}(1)_f$ to be discussed momentarily, is the residual 6d symmetry that is preserved by the compactification. In particular $\mathrm{U}(1)_t \subset \mathrm{SU}(2)_L$, while $\mathrm{U}(1)_c \times \mathrm{SU}(8)_u \subset \mathrm{E}_8$. In the next section we will see that upon the construction of theories corresponding to $\mathbb{S}^2$ compactifications the $\mathrm{U}(1)_f$ symmetry is to be identified, upon properly mixing it with other $U(1)$ symmetries, with the Cartan of the $\mathrm{SU}(2)_{\mathrm{ISO}}$ isometry of $\mathbb{S}^2$. In particular it is natural then to think about the $\mathrm{U}(1)_f$ symmetry of the tube to be associated with its isometry, namely to be related to the KK symmetry of the comapctification of 6d SCFT on a circle[7].

It is convenient to encode the matter content and charges under various symmetries in the expressions of the superconformal index for the theory, which is written as

$$\mathcal{I}(p,q) = Tr(-1)^F p^{j_1 + j_2 + \frac{1}{2}r} q^{j_2 - j_1 + \frac{1}{2}r} t_i^{\mathcal{Q}_i}, \tag{3}$$

where $j_1$ and $j_2$ are the Cartan generators of the $\mathrm{SU}(2)_1 \times \mathrm{SU}(2)_2$ isometry group of $\mathbb{S}^3$, $r$ is the generator of the $\mathrm{U}(1)_r$ R-symmetry, and $Q_i$'s are the Cartan generators of the global symmetry. More detailed reviews of the index are given in Appendix D. The supersymmetric index of the tube theory is then,

$$
\begin{aligned}
\mathcal{I}_{tube}(x_n; y_n; c, t, f; u_a) &= \prod_{n=1}^{Q} \prod_{a=1}^{8} \Gamma_e \left( (pq)^{\frac{1}{2}} c^{-\frac{1}{2}} f^{-\frac{1}{4}} u_a x_n^{\pm 1} \right) \Gamma_e \left( (pq)^{\frac{1}{2}} c^{-\frac{1}{2}} f^{\frac{1}{4}} u_a^{-1} y_n^{\pm 1} \right) \times \\
&\times \ \mathcal{I}_{FE[\mathrm{USp}(2Q)]}(x_n; y_n; c, t),
\end{aligned}
\tag{4}
$$

where $\mathcal{I}_{FE[\mathrm{USp}(2Q)]}$ is the supersymmetric index of $FE[\mathrm{USp}(2Q)]$ defined in (A.8) and $x_n$, $y_n$, $c$, $t$, $f$, $u_a$ are fugacities in the Cartan of $\mathrm{USp}(2Q)_x$, $\mathrm{USp}(2Q)_y$, $\mathrm{U}(1)_c$, $\mathrm{U}(1)_t$, $\mathrm{U}(1)_f$, $\mathrm{SU}(8)_u$, respectively, with the constraint $\prod_{a=1}^{8} u_a = 1$. Moreover, we used the parametrization of the abelian symmetries and of the R-symmetry specified in Appendix A for $FE[\mathrm{USp}(2Q)]$. Note that the R-symmetry assignment we use is not necessarily the superconformal one. In particular, the two octets of chirals $M$, $M'$ have $\mathrm{U}(1)_c$ charge $-\frac{1}{2}$, $\mathrm{U}(1)_f$ charge charge $\pm\frac{1}{4}$ and R-charge 1.

The tube theory possesses two types of moment map operators that transform under the symmetries of the punctures:

- The chiral fields contained in the two octets $M$ and $M'$, which transform in the fundamental representation of $\mathrm{USp}(2Q)_x$ and $\mathrm{USp}(2Q)_y$ respectively;

- The operators $\mathsf{O}_H$ and $\mathsf{C}$ of $FE[\mathrm{USp}(2Q)]$ (see Appendix A), which transform in the traceless antisymmetric representation of $\mathrm{USp}(2Q)_x$ and $\mathrm{USp}(2Q)_y$ respectively.

We would like to give vev or linear superpotential interaction to some of these operators so to completely break the symmetry of one of the two punctures, say $\mathrm{USp}(2Q)_x$. We will do so by

---

[6]Contractions of indices of $\mathrm{USp}(2n)$ representations are performed with the two-index totally antisymmetric tensor $J = \mathbb{I}_n \otimes i\,\sigma_2$, so all the traces are defined with the insertion of such a tensor.

[7]Let us also mention here that the $\mathrm{U}(1)_f$ symmetry is broken by anomalies/superpotentials when one constructs theories corresponding to a torus, see *e.g.* [8]. This breaking leaves behind a discrete group, whose order depends on the flux. It would be very interesting to understand precisely the relation between $U(1)_f$ to the KK symmetry and the nature of this discrete subgroup. We leave this for future investigations.

first giving a vev to the traceless antisymmetric operator $O_H$. Such a vev can at most break the $USp(2Q)_x$ symmetry down to $SU(2)_v$. Hence, we will then need a vev for one of the octet fields to further break $SU(2)_v$.

We first deform the tube model by turning on linearly in the superpotential the operator $O_H$ as follows:

$$\delta \mathcal{W} = J_Q O_H \,, \tag{5}$$

where

$$J_Q = \frac{i\sigma_2}{2} \otimes \left( J_Q + J_Q^T \right) \tag{6}$$

and $J_Q$ is the Jordan matrix of dimension $Q$. The operator $O_H$ of $FE[USp(2Q)]$ is actually a matrix of singlets flipping the operator H (see Appendix A). Hence, this deformation implies a vev for H that can be understood by looking at the equations of motion of $O_H$[8]

$$\langle H \rangle = J_Q \,. \tag{7}$$

The effect of this vev can be more easily understood at the level of the supersymmetric index. In particular, the deformation imposes the following constraints on the fugacities:

$$x_n = t^{n-1} v, \quad n = 1, \cdots, Q \,. \tag{8}$$

Using the identity (A.13), which was proven in Corollary 2.8 of [34], we find that the supersymmetric index after such a specification of the fugacities reduces to

$$
\begin{aligned}
\mathcal{I}_{tube}(v, t\, v, \cdots, t^{Q-1} v; y_n; c, t, f; u_a) &= \prod_{n=1}^{Q} \prod_{a=1}^{8} \Gamma_e \left( (pq)^{\frac{1}{2}} c^{-\frac{1}{2}} u_a (t^{n-1} v)^{\pm 1} \right) \times \\
&\times \; \Gamma_e \left( (pq)^{\frac{1}{2}} c^{-\frac{1}{2}} u_a^{-1} y_n^{\pm 1} \right) \times \\
&\times \; \prod_{n=2}^{Q} \frac{1}{\Gamma_e(t^n)} \prod_{n=1}^{Q} \frac{\Gamma_e \left( v\, c\, y_n^{\pm 1} \right) \Gamma_e \left( v^{-1} c\, t^{1-Q} y_n^{\pm 1} \right)}{\Gamma_e(c^2 t^{1-n})} \,. 
\end{aligned} \tag{9}
$$

In order to make the residual $SU(2)_v$ symmetry manifest we have to perform the redefinition $v \to t^{\frac{1-Q}{2}} v$

$$
\begin{aligned}
\mathcal{I}_{tube}(t^{\frac{1-Q}{2}} v, t^{\frac{3-Q}{2}} v, \cdots, t^{\frac{Q-1}{2}} v; y_n; c, t, f; u_a) &= \\
= \prod_{n=1}^{Q} \prod_{a=1}^{8} \Gamma_e &\left( (pq)^{\frac{1}{2}} c^{-\frac{1}{2}} t^{\frac{Q-2n+1}{2}} v^{\pm 1} f^{-\frac{1}{4}} u_a \right) \Gamma_e \left( (pq)^{\frac{1}{2}} c^{-\frac{1}{2}} f^{\frac{1}{4}} u_a^{-1} y_n^{\pm 1} \right) \times \\
\times \prod_{n=2}^{Q} \Gamma_e &\left( pq\, t^{-n} \right) \prod_{n=1}^{Q} \Gamma_e \left( pq\, c^{-2} t^{n-1} \right) \Gamma_e \left( c\, t^{\frac{1-Q}{2}} v^{\pm 1} y_n^{\pm 1} \right) \,.
\end{aligned} \tag{10}
$$

The next step consists of breaking also the $SU(2)_v$ symmetry by giving a vev to one of the octet fields represented now in the index by $\prod_{n=1}^{Q} \prod_{a=1}^{8} \Gamma_e \left( (pq)^{\frac{1}{2}} c^{-\frac{1}{2}} t^{\frac{Q-2n+1}{2}} v^{\pm 1} f^{-\frac{1}{4}} u_a \right)$. It turns out that the correct choice to reproduce the anomalies of the cap model predicted from 6d is to

---

[8]A more general class of vevs for the antisymmetric operators of $FE[USp(2Q)]$ has been studied in [19]. The vevs studied in that reference are labeled by partitions of $N$ and preserve a subgroup of $USp(2Q)$ of the form $\prod_k USp(2n_k)$, where the integers $n_k$ can be determined by the data of the partition. Such more generic choices of vev should lead to different types of punctures with global symmetry being a subgroup of $USp(2Q)$. We will be only interested here in completely breaking the puncture symmetry, that is producing *no puncture*. We thus focus on the vev preserving the minimal $SU(2)_v$ symmetry which we then completely break with an octet vev.

give vev to the field corresponding to $n = 1$ and any $a = 1, \cdots, 8$. For definiteness we shall choose $a = 8$. Then, the vev implies the following constraint on the fugacities of the index:

$$v = (pq)^{\frac{1}{2}} c^{-\frac{1}{2}} t^{\frac{Q-1}{2}} f^{-\frac{1}{4}} u_8 \,. \tag{11}$$

The index (10) after giving such a vev becomes[9]

$$
\begin{aligned}
\mathcal{I} \;=\; & \prod_{n=2}^{Q} \Gamma_e\left(t^{1-n}\right) \Gamma_e\left(pq\, t^{-n}\right) \prod_{n=1}^{Q} \Gamma_e\left(pq\, c^{-2} t^{n-1}\right) \Gamma_e\left(pq\, c^{-1} t^{Q-n} f^{-\frac{1}{2}} u_8^2\right) \times \\
& \times\; \Gamma_e\left((pq)^{\frac{1}{2}} c^{-\frac{1}{2}} f^{\frac{1}{4}} u_8^{-1} y_n^{\pm 1}\right) \Gamma_e\left((pq)^{\frac{1}{2}} c^{\frac{1}{2}} f^{-\frac{1}{4}} u_8 y_n^{\pm 1}\right) \Gamma_e\left((pq)^{-\frac{1}{2}} c^{\frac{3}{2}} t^{1-Q} f^{\frac{1}{4}} u_8^{-1} y_n^{\pm 1}\right) \times \\
& \times\; \prod_{a=1}^{7} \Gamma_e\left(pq\, c^{-1} t^{Q-n} f^{-\frac{1}{2}} u_8 u_a\right) \Gamma_e\left(t^{1-n} u_8^{-1} u_a\right) \Gamma_e\left((pq)^{\frac{1}{2}} c^{-\frac{1}{2}} f^{\frac{1}{4}} u_a^{-1} y_n^{\pm 1}\right) \,. \tag{12}
\end{aligned}
$$

This is not the index of the cap model that we are looking for yet. Indeed, in general one may need to introduce additional gauge singlet chiral fields that flip some of the operators of the theory in order to get the correct model corresponding to the compactification of the 6d theory. The singlets that need to be added can be worked out by requiring that the anomalies of the resulting model match those predicted from 6d for a compactification on a one-punctured sphere with some value of the flux (see Appendix B). This was already noticed in Section 6 of [8] for the rank $Q = 1$ case, where the symmetry carried by each puncture is just SU(2) so only the octet vev was needed in order to close it. It turns out that in our higher rank case some of the operators that we need to flip are just straightforward generalizations of those worked out for rank 1, while the others only appear for $Q > 1$. The complete list of singlets that we have to add is the following (again encoded in their contributions to the index):

$$\prod_{n=1}^{8} \prod_{a=1}^{7} \Gamma_e\left(pq\, t^{n-1} u_8 u_a^{-1}\right)$$

$$\prod_{n=1}^{Q} \Gamma_e\left(c\, t^{n-Q} f^{\frac{1}{2}} u_8^{-2}\right)$$

$$\prod_{n=2}^{Q} \Gamma_e\left(pq\, t^{n-1}\right) \Gamma_e\left(t^n\right)$$

$$\Gamma_e(t)^{Q-1} \prod_{n<m}^{Q} \Gamma_e\left(t\, y_n^{\pm 1} y_m^{\pm 1}\right) \,. \tag{13}$$

In addition, we make a shift of fugacity $c \to c f^{\frac{1}{2}}$ for later convenience. The index of the resulting cap model is thus

$$
\begin{aligned}
& \mathcal{I}_{cap}(y_n; c, t, f; u_a; u_8) \\
=\; & \Gamma_e(t)^{Q-1} \prod_{n<m}^{Q} \Gamma_e\left(t\, y_n^{\pm 1} y_m^{\pm 1}\right) \prod_{n=1}^{Q} \underbrace{\Gamma_e\left(pq\, c^{-2} t^{n-1} f^{-1}\right)}_{b_n} \underbrace{\Gamma_e\left((pq)^{-\frac{1}{2}} c^{\frac{3}{2}} t^{1-Q} f u_8^{-1} y_n^{\pm 1}\right)}_{P} \times \\
& \times\; \underbrace{\Gamma_e\left((pq)^{\frac{1}{2}} c^{-\frac{1}{2}} u_8^{-1} y_n^{\pm 1}\right)}_{L_8} \underbrace{\Gamma_e\left((pq)^{\frac{1}{2}} c^{\frac{1}{2}} u_8 y_n^{\pm 1}\right)}_{K} \times \\
& \times\; \prod_{a=1}^{7} \underbrace{\Gamma_e\left(pq\, c^{-1} t^{Q-n} f^{-1} u_8 u_a\right)}_{R^{(n)a}} \underbrace{\Gamma_e\left((pq)^{\frac{1}{2}} c^{-\frac{1}{2}} u_a^{-1} y_n^{\pm 1}\right)}_{L_a} \,. \tag{14}
\end{aligned}
$$

---

[9]For future convenience we are not simplifying in this expressions the contributions of some of the massive fields.

We can check that the model we obtained after the vevs and the addition of the singlets indeed corresponds to the compactification of the rank $Q$ E-string theory on a sphere with one puncture for some value of flux by computing its anomalies and comparing with those that can be predicted from $6d$, which we review in Appendix B. This anomaly matching is a necessary condition for the duality between the 4d model we propose and the 6d theory compactified on the 2d surface. In addition, we also provide further evidence in the next section that one can construct various 4d models corresponding to the E-string theory on a sphere with different fluxes by gluing the cap models we propose here. Those 4d models exhibit the expected (enhanced) symmetries and the spectrums of operators perfectly consistent with the 6d theory compactified on a sphere with a given flux, which is strong evidence of our conjecture that the proposed cap model indeed corresponds to the E-string theory compactified on a punctured sphere, or a cap.

Remembering the constraint $u_8 = \prod_{a=1}^{7} u_a^{-1}$, we first find the following linear anomalies for the U(1)'s in the Cartan of the original 6d $E_8 \times SU(2)_L$ symmetry:

$$\begin{aligned}
\operatorname{Tr} U(1)_R &= 2Q(Q+1), & \operatorname{Tr} U(1)_t &= (Q-1)(4Q+1), \\
\operatorname{Tr} U(1)_c &= -13Q, & \operatorname{Tr} U(1)_{u_a} &= -6Q \quad a = 1, \cdots, 7.
\end{aligned} \tag{15}$$

For the cubic non-mixed anomalies we have

$$\begin{aligned}
\operatorname{Tr} U(1)_R^3 &= -2Q(Q+1)^2, & \operatorname{Tr} U(1)_t^3 &= (Q-1)(2Q^2+1), \\
\operatorname{Tr} U(1)_c^3 &= -10Q, & \operatorname{Tr} U(1)_{u_a}^3 &= -6Q.
\end{aligned} \tag{16}$$

Next, we list all the cubic mixed anomalies for the abelian symmetries:

$$\begin{aligned}
&\operatorname{Tr} U(1)_R^2 U(1)_t = -\frac{2}{3} Q(Q^2-1), \ \operatorname{Tr} U(1)_R^2 U(1)_c = \frac{3}{2} Q(Q+1), \ \operatorname{Tr} U(1)_R^2 U(1)_{u_a} = Q(Q+1), \\
&\operatorname{Tr} U(1)_t^2 U(1)_R = \frac{2}{3} Q(Q^2-1), \ \operatorname{Tr} U(1)_t^2 U(1)_c = -\frac{3}{2} Q(Q-1), \ \operatorname{Tr} U(1)_t^2 U(1)_{u_a} = -Q(Q-1), \\
&\operatorname{Tr} U(1)_c^2 U(1)_R = Q(Q+1), \ \operatorname{Tr} U(1)_c^2 U(1)_t = Q(Q-1), \ \operatorname{Tr} U(1)_c^2 U(1)_{u_a} = -2Q, \\
&\operatorname{Tr} U(1)_{u_a}^2 U(1)_R = Q(Q+1), \ \operatorname{Tr} U(1)_{u_a}^2 U(1)_t = Q(Q-1), \ \operatorname{Tr} U(1)_{u_a}^2 U(1)_c = -4Q, \\
&\operatorname{Tr} U(1)_{u_a} U(1)_{u_b}^2 = -3Q, \ \operatorname{Tr} U(1)_R U(1)_{u_a} U(1)_{u_b} = \frac{1}{2} Q(Q+1), \\
&\operatorname{Tr} U(1)_t U(1)_{u_a} U(1)_{u_b} = \frac{1}{2} Q(Q-1), \ \operatorname{Tr} U(1)_c U(1)_{u_a} U(1)_{u_b} = -2Q, \qquad a \neq b, \\
&\operatorname{Tr} U(1)_{u_a} U(1)_{u_b} U(1)_{u_d} = -2Q, \qquad a \neq b \neq d \neq a.
\end{aligned} \tag{17}$$

Finally, we have the anomalies between these U(1) symmetries and the USp($2Q$) symmetry of the puncture

$$\begin{aligned}
\operatorname{Tr} U(1)_R \operatorname{USp}(2Q)^2 &= -\frac{Q+1}{2}, & \operatorname{Tr} U(1)_t \operatorname{USp}(2Q)^2 &= -\frac{Q-1}{2}, \\
\operatorname{Tr} U(1)_c \operatorname{USp}(2Q)^2 &= -1, & \operatorname{Tr} U(1)_{u_a} \operatorname{USp}(2Q)^2 &= 0.
\end{aligned} \tag{18}$$

All of these anomalies perfectly match those that one can compute from 6d using equations (B.16)-(B.17) for a sphere with one puncture and flux $\mathcal{F}_{cap} = \left(-\frac{1}{2}; \frac{3}{4}; \frac{1}{8}, \cdots, \frac{1}{8}, -\frac{7}{8}\right)$. Comparing with the flux of the original tube model $\mathcal{F}_{tube} = (0; \frac{1}{2}; 0, \cdots, 0)$, we notice that the effect of the vevs has been to shift the entries of the flux vector. In particular, $n_c \to n_c + \frac{1}{4}$, $n_8 \to n_8 - \frac{7}{8}$ and $n_a \to n_a + \frac{1}{8}$ for $a = 1, \cdots, 7$. This is the same shift for the $E_8$ fluxes found in [8] for the rank $Q = 1$ case, so we interpret it as the effect of the octet vev. Moreover, we also have that the flux for $SU(2)_L$ has been shifted by $n_t \to n_t - \frac{1}{2}$, which we instead interpret as the effect of the antisymmetric vev.

In order to interpret the model we obtained, it is useful to redefine the fugacities in the index (14) in such a way that they conform to the new manifest global symmetry, which is

$$\text{USp}(2Q) \times \text{SU}(7)_u \times \text{U}(1)_x \times \text{U}(1)_c \times \text{U}(1)_t \times \text{U}(1)_f. \qquad (19)$$

This is achieved by the shifts

$$
\begin{aligned}
u_{a=1,\dots,7} &\longrightarrow x^{-1} u_a, \\
u_8 &\longrightarrow x^7,
\end{aligned} \qquad (20)
$$

with the new $u_a$ on the right hand side satisfying $\prod_{a=1}^{7} u_a = 1$. The index thus reads

$$
\begin{aligned}
&\mathcal{I}_{cap}(y_n; c, t, f, x^{-1} u_a; x) \\
&= \Gamma_e(t)^{Q-1} \underbrace{\prod_{n<m}^{Q} \Gamma_e\left(t\, y_n^{\pm 1} y_m^{\pm 1}\right)}_{A} \prod_{n=1}^{Q} \underbrace{\Gamma_e\left(pq\, c^{-2} t^{n-1} f^{-1}\right)}_{b_n} \underbrace{\Gamma_e\left((pq)^{-\frac{1}{2}} c^{\frac{3}{2}} t^{1-Q} f x^{-7} y_n^{\pm 1}\right)}_{P} \times \\
&\times \underbrace{\Gamma_e\left((pq)^{\frac{1}{2}} c^{-\frac{1}{2}} x^{-7} y_n^{\pm 1}\right)}_{L_8} \underbrace{\Gamma_e\left((pq)^{\frac{1}{2}} c^{\frac{1}{2}} x^7 y_n^{\pm 1}\right)}_{K} \times \\
&\times \prod_{a=1}^{7} \underbrace{\Gamma_e\left(pq\, c^{-1} t^{Q-n} f^{-1} x^{-6} u_a\right)}_{R^{(n)a}} \underbrace{\Gamma_e\left((pq)^{\frac{1}{2}} c^{-\frac{1}{2}} x\, u_a^{-1} y_n^{\pm 1}\right)}_{L_a}. \qquad (21)
\end{aligned}
$$

Note that the cap model is thus a simple Wess–Zumino model. Of course this flows to a collection of free fields in the IR. However, the various superpotentials of the model are constraining the global symmetry of the theory. When one glues the cap model to other theories by gauging the USp(2$Q$) symmetry these superpotentials become important. The massless fields of the WZ model can be schematically represented with the following quiver diagram:

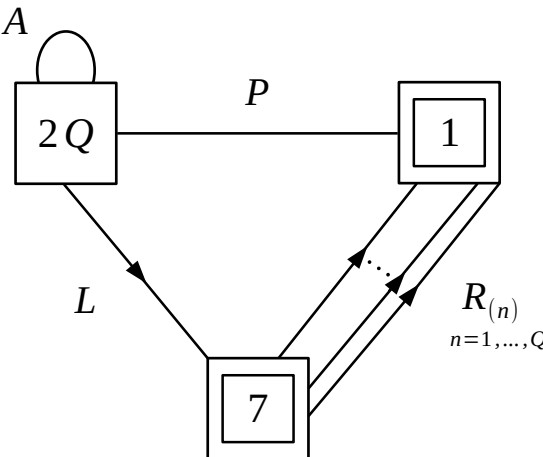

Figure 3: The cap theory. This is just a WZ model with the superpotentials detailed in (22).

Here as before the simple square box represents the USp(2$Q$) flavor symmetry of the remaining puncture, while double square boxes represent unitary flavor symmetries out of which we have to mod out an overall U(1). In particular the square box with the 1 inside represents the U(1) flavor symmetry associated to the fugacity $x$ in the index, while the square box with the 7

inside represents the SU(7) flavor symmetry associated to the fugacities $u_a$ in the index. On top of the fields represented in the quiver, we also have the singlets under the non-abelian symmetries $b_n$ and the two massive fields $L_8$ and $K$, which we prefer to keep and not integrate out in order to write the superpotential in a more pleasant way:

$$\mathcal{W}_{cap} = \sum_{n=1}^{Q} \sum_{a=1}^{7} \text{Tr}_y \left( R^{(n)a} A^{Q-n} P L_a \right) + \sum_{n=1}^{Q} \text{Tr}_y \left( b_n A^{Q-n} P K \right) + K L_8 \,. \tag{22}$$

Integrating $L_8$ and $K$ out we obtain the field content of the previous quiver and the interaction superpotential between the remaining massless fields. We keep here the massive fields as using those it will be more convenient to define the procedure of gluing the cap to other models.

## 3 Examples of $\mathbb{S}^2$ models

In the previous section we have obtained the model corresponding to the one punctured sphere and here we will use it to construct several examples of theories which we will associate to compactifications on $\mathbb{S}^2$. In particular we will see how various dualities and emergence of symmetry phenomena naturally arise in this construction. Moreover we will directly identify the $U(1)_f$ symmetry with the Cartan generator of the $SU(2)_{\text{ISO}}$ geometric symmetry of $\mathbb{S}^2$[10].

### 3.1 $SU(8) \times U(1)^2$ and $SU(8) \times SU(2) \times U(1)$ sphere models

Our first example is the 4d model corresponding to the compactification of the rank $Q$ E-string theory on a sphere with flux

$$\mathcal{F} = \left( -1; \frac{3}{2}; \frac{1}{4}, \frac{1}{4}, \frac{1}{4}, \frac{1}{4}, \frac{1}{4}, \frac{1}{4}, \frac{1}{4}, -\frac{7}{4} \right). \tag{23}$$

The flux is given as before in the basis of $U(1)_t \times U(1) \times SU(8) \subset SU(2)_L \times E_8$, where $E_8 \times SU(2)_L$ is the global symmetry of the E-string theory. However, the gluing rules we consider will be written in a simpler way if we use the SO(16) basis rather than that of $U(1) \times SU(8)$. Thus, using the transformation rule between the bases

$$n_a^{SO(16)} = n^{U(1)} + 2n_a^{SU(8)}, \qquad a = 1, \dots, 8, \tag{24}$$

the flux (23) can be written in the SO(16) basis as follows:

$$\mathcal{F} = (-1; 2, 2, 2, 2, 2, 2, 2, -2). \tag{25}$$

The first component of $\mathcal{F}$ corresponds to the $U(1)_t$ flux, while the other 8 components correspond to the flux of $U(1)^8 \subset SO(16) \subset E_8$. One can easily check that this flux preserves $SU(8) \times U(1)_b \times U(1)_t \subset E_8 \times SU(2)_L$. To find the preserved symmetry we need to find the roots of SO(16) which are orthogonal to the flux as well as the orthogonal spinorial weights [18]. Here we have that only 56 of the roots are orthogonal to $\mathcal{F}$, which together with the Cartans form $SU(8) \times U(1)_b$ preserved symmetry. A sphere with such flux can be constructed from two basic caps, whose flux we recall is given by

$$\mathcal{F}^{(1)} = \mathcal{F}^{(2)} = \left( -\frac{1}{2}; \frac{3}{4}; \frac{1}{8}, \frac{1}{8}, \frac{1}{8}, \frac{1}{8}, \frac{1}{8}, \frac{1}{8}, \frac{1}{8}, -\frac{7}{8} \right), \tag{26}$$

---

[10]For simplicity we will only consider gluing pair of cap theories together in various ways without introducing tube theories. Technically this makes it simpler to perform explicitly various index checks. As the tube theory has one of the USp(2Q) symmetries emergent in the IR, adding it would entail gauging of IR emergent symmetries: this is not an issue conceptually, but as we refrain from doing so the dualities we will arrive at are standard dualities between completely Lagrangian theories.

in the U(1) × SU(8) basis or

$$\mathcal{F}^{(1)} = \mathcal{F}^{(2)} = \left(-\frac{1}{2}; 1, 1, 1, 1, 1, 1, 1, -1\right), \tag{27}$$

in the SO(16) basis. We remind the reader that the geometric action of gluing has the following field theoretic interpretation, see *e.g.* [18]. We take two theories with punctures with symmetry USp(2Q) and associated "moment map" operators $M_a$ and $M'_a$ respectively. Then for each component of the moment map we have the choice whether to Φ-glue or S-glue. The former choice amounts to adding a chiral field $\Phi_a$ in the fundamental/traceless antisymmetric representation of USp(2Q) and turning on the superpotential

$$\Delta W_a = \Phi_a \cdot (M_a - M'_a). \tag{28}$$

The latter choice amounts only to turning on a superpotential without adding any additional fields

$$\Delta W_a = M_a \cdot M'_a. \tag{29}$$

The fundamental moment maps are charged under the Cartan of SO(16) while the traceless antisymmetric moment map is charged under the Cartan of $SU(2)_L$. To obtain the fluxes of the glued theory we join the fluxes of the two theories depending on the gluing. If a moment map component is S-glued we subtract the fluxes of the symmetry it is charged under while we add them for Φ-gluing:

$$n'_a = \begin{cases} n_a^{(1)} + n_a^{(2)}, & a \in \Phi, \\ n_a^{(1)} - n_a^{(2)}, & a \in S, \end{cases} \tag{30}$$

where Φ and S denotes the Φ-gluing and the S-gluing respectively. The SO(16) basis is thus more convenient to discuss the gluings.

Hence, the two basic caps glued with the Φ-gluing for all the moment map components simply leads to a sphere with the flux

$$\mathcal{F} = \mathcal{F}^{(1)} + \mathcal{F}^{(2)} = (-1; 2, 2, 2, 2, 2, 2, 2, -2), \tag{31}$$

which is exactly (25).

The resulting 4d model is thus expected to preserve $SU(8)_v \times U(1)_b \times U(1)_t$, which stems from $E_8 \times SU(2)_t$ in 6d. Moreover, on top of $SU(8)_v \times U(1)_b \times U(1)_t$, this theory also exhibits the $SU(2)_f$ symmetry which we will argue comes from the isometry of the two-sphere on which we compactified the 6d theory. Therefore, the total global symmetry of the model is given by

$$SU(8)_v \times SU(2)_f \times U(1)_b \times U(1)_t, \tag{32}$$

which we will confirm using the superconformal index.

To construct the model corresponding to the flux (25), we first recall that the basic cap is given by the WZ model with the superpotential

$$\mathcal{W}_{cap} = \sum_{n=1}^{Q} \sum_{a=1}^{7} \text{Tr}_z \left( R^{(n)a} A^{Q-n} P L_a \right) + \sum_{n=1}^{Q} \text{Tr}_z \left( b_n A^{Q-n} P K \right) + K L_8, \tag{33}$$

which preserves

$$USp(2Q) \times SU(7)_u \times U(1)_x \times U(1)_c \times U(1)_t \times U(1)_f. \tag{34}$$

The index of the basic cap is given by

$$
\begin{aligned}
&\mathcal{I}_{cap}(y_n; c, t, f; u_1, \ldots, u_7; x)\\
&= \Gamma_e(t)^{Q-1} \underbrace{\prod_{n<m}^{Q} \Gamma_e\left(t y_n^{\pm 1} y_m^{\pm 1}\right)}_{A} \underbrace{\prod_{n=1}^{Q} \Gamma_e\left(pq\, t^{n-1} c^{-2} f^{-1}\right)}_{b_n} \underbrace{\prod_{n=1}^{Q} \Gamma_e\left((pq)^{-\frac{1}{2}} t^{1-Q} c^{\frac{3}{2}} f x^{-7} y_n^{\pm 1}\right)}_{P}\\
&\times \underbrace{\prod_{a=1}^{7}\prod_{n=1}^{Q} \Gamma_e\left((pq)^{\frac{1}{2}} c^{-\frac{1}{2}} x u_a^{-1} y_n^{\pm 1}\right)}_{L_a} \underbrace{\prod_{n=1}^{Q}\prod_{a=1}^{7} \Gamma_e\left(pq\, t^{n-1} c^{-1} f^{-1} x^6 u_a\right)}_{R^{(n)a}}\\
&\times \underbrace{\prod_{n=1}^{Q} \Gamma_e\left((pq)^{\frac{1}{2}} c^{-\frac{1}{2}} x^{-7} y_n^{\pm 1}\right)}_{L_8} \underbrace{\prod_{n=1}^{Q} \Gamma_e\left((pq)^{\frac{1}{2}} c^{\frac{1}{2}} x^7 y_n^{\pm 1}\right)}_{K},
\end{aligned}
\tag{35}
$$

where recall that we have redefined

$$
\begin{aligned}
u_{a=1,\ldots,7} &\longrightarrow x^{-1} u_a,\\
u_8 &\longrightarrow x^7
\end{aligned}
\tag{36}
$$

is taken with new $u_a$ on the right hand side satisfying $\prod_{a=1}^{7} u_a = 1$.

We then glue two caps using the $\Phi$-gluing; namely, we introduce $\hat{A}$, $\Phi^{1,\ldots,8}$ and the superpotential

$$
\mathcal{W}_{glue} = \text{Tr}_z\left[\hat{A} \cdot (A - \tilde{A})\right] + \sum_{b=1}^{8} \Phi^b\left(L_b - \tilde{L}_b\right)
\tag{37}
$$

and gauge the puncture symmetry $USp(2Q)$. The entire superpotential of the glued theory is thus given by

$$
\begin{aligned}
\mathcal{W} &= \mathcal{W}_{cap} + \tilde{\mathcal{W}}_{cap} + \mathcal{W}_{glue}\\
&= \sum_{n=1}^{Q}\sum_{a=1}^{7} \text{Tr}_z\left(R^{(n)a} A^{Q-n} P L_a\right) + \sum_{n=1}^{Q} \text{Tr}_z\left(b_n A^{Q-n} P K\right) + K L_8\\
&\quad + \sum_{n=1}^{Q}\sum_{a=1}^{7} \text{Tr}_z\left(\tilde{R}^{(n)a} \tilde{A}^{Q-n} \tilde{P} \tilde{L}_a\right) + \sum_{n=1}^{Q} \text{Tr}_z\left(\tilde{b}_n \tilde{A}^{Q-n} \tilde{P} \tilde{K}\right) + \tilde{K} \tilde{L}_8\\
&\quad + \text{Tr}_z\left[\hat{A} \cdot (A - \tilde{A})\right] + \sum_{b=1}^{8} \Phi^b\left(L_b - \tilde{L}_b\right),
\end{aligned}
\tag{38}
$$

where $\text{Tr}_z$ is the trace over the gauged puncture symmetry $USp(2Q)$. Note that the superpotential contains some massive fields. Once we integrate them out, the superpotential becomes

$$
\begin{aligned}
\mathcal{W} &= \sum_{n=1}^{Q}\sum_{b=1}^{7} \text{Tr}_z\left(R^{(n)b} A^{Q-n} P L_b\right) + \sum_{n=1}^{Q}\sum_{b=1}^{7} \text{Tr}_z\left(\tilde{R}^{(n)b} A^{Q-n} \tilde{P} L_b\right)\\
&\quad + \frac{1}{3}\sum_{n=1}^{Q} \text{Tr}_z\left(b_n A^{Q-n} P(K - \tilde{K} - \Phi)\right) + \frac{1}{3}\sum_{n=1}^{Q} \text{Tr}_z\left(\tilde{b}_n \tilde{A}^{Q-n} \tilde{P}(\tilde{K} - K + \Phi)\right).
\end{aligned}
\tag{39}
$$

One can check that the superpotential (39) actually preserves

$$
SU(8)_v \times SU(2)_f \times U(1)_b \times U(1)_t,
\tag{40}
$$

Table 1: The matter contents of the $SU(8) \times SU(2) \times U(1)^2$ model and their charges.

| | $USp(2Q)$ | $SU(8)_v$ | $SU(2)_f$ | $U(1)_b$ | $U(1)_t$ | $U(1)_R$ |
|---|---|---|---|---|---|---|
| $S^{(n)}$ | **1** | **8** | **2** | $-3$ | $n-1$ | $2$ |
| $Q$ | **2Q** | $\overline{\bf 8}$ | **1** | $-1$ | $0$ | $1$ |
| $P$ | **2Q** | **1** | **2** | $4$ | $1-Q$ | $-1$ |
| $A$ | **antisym** | **1** | **1** | $0$ | $1$ | $0$ |

which is consistent with the 6d prediction. The symmetry (40) can be made manifest by rewriting the superpotential as follows:

$$\mathcal{W} = \sum_{n=1}^{Q} \sum_{b=1}^{8} \sum_{\alpha=\pm} \mathrm{Tr}_z \left( S^{(n)b}{}_{\alpha} A^{Q-n} P^{\alpha} Q_b \right), \tag{41}$$

where we have defined

$$S^{(n)b}_{+} = \begin{cases} R^{(n)b}, & b=1,\dots,7, \\ b_n, & b=8, \end{cases} \tag{42}$$

$$S^{(n)b}_{-} = \begin{cases} \tilde{R}^{(n)b}, & b=1,\dots,7, \\ -\tilde{b}_n, & b=8, \end{cases} \tag{43}$$

$$Q_b = \begin{cases} L_b, & b=1,\dots,7, \\ \frac{K-\tilde{K}-\Phi}{3}, & b=8, \end{cases} \tag{44}$$

$$P^+ = P, \tag{45}$$

$$P^- = \tilde{P}. \tag{46}$$

The resulting model is the $USp(2Q)$ gauge theory with one traceless antisymmetric $A$, 10 fundamentals $(Q_b; P^{\pm})$ and $16Q$ gauge singlets $S^{(n)b}_{\pm}$. The charges of each chiral multiplet are presented in Table 1. As we mentioned, $SU(2)_f$ doesn't come from the symmetry of 6d E-string but originates from the isometry of the compactifying two-sphere, which we will show using the anomaly shortly.

Now let us evaluate the superconformal index. The index of this model is given by

$$\mathcal{I}_{(-1;2^7,-2)} = \oint dz_Q \, \Gamma_e \left( pqt^{-1} \right)^{Q-1} \prod_{n<m}^{Q} \Gamma_e \left( pqt^{-1} z_n^{\pm 1} z_m^{\pm 1} \right) \prod_{n=1}^{Q} \prod_{a=1}^{8} \Gamma_e \left( (pq)^{\frac{1}{2}} c^{\frac{1}{2}} u_a z_n^{\pm 1} \right)$$

$$\times \mathcal{I}_{cap}(z_n; c, t, f; u_1, \dots, u_7; x) \times \mathcal{I}_{cap}(z_n; c, t, f^{-1}; u_1, \dots, u_7; x)$$

$$= \prod_{n=1}^{Q} \Gamma_e \left( pqt^{n-1} c^{-2} f^{\pm 1} \right) \prod_{n=1}^{Q} \prod_{a=1}^{7} \Gamma_e \left( pqt^{n-1} c^{-1} f^{\pm 1} x^6 u_a \right)$$

$$\times \oint dz_Q \, \Gamma_e (t)^{Q-1} \prod_{n<m}^{Q} \Gamma_e \left( t z_n^{\pm 1} z_m^{\pm 1} \right) \prod_{n=1}^{Q} \Gamma_e \left( (pq)^{-\frac{1}{2}} t^{1-Q} c^{\frac{3}{2}} f^{\pm 1} x^{-7} z_n^{\pm 1} \right)$$

$$\times \prod_{a=1}^{7} \prod_{n=1}^{Q} \Gamma_e \left( (pq)^{\frac{1}{2}} c^{-\frac{1}{2}} x u_a^{-1} z_n^{\pm 1} \right) \prod_{n=1}^{Q} \Gamma_e \left( (pq)^{\frac{1}{2}} c^{\frac{1}{2}} x^7 z_n^{\pm 1} \right), \tag{47}$$

where all the massive contributions are canceled out. As we have just observed, while the basic cap preserves $SU(7)_u \times U(1)_x \times U(1)_c$, the sphere model preserves not only $SU(7)_u \times U(1)_x \times U(1)_c$ but also $SU(8)_v \times U(1)_b$. Therefore, in terms of the $SU(8)_v \times U(1)_b$ fugacities which are defined

by

$$v_{a=1,\dots,7} = c^{\frac{1}{8}} x^{\frac{3}{4}} u_a \,,$$
$$v_8 = c^{-\frac{7}{8}} x^{-\frac{21}{4}} \,,$$
$$b = c^{\frac{3}{8}} x^{-\frac{7}{4}} \,, \tag{48}$$

the index is written as

$$\mathcal{I}_{(-1;2^7,-2)} = \prod_{n=1}^{Q}\prod_{a=1}^{8} \underbrace{\Gamma_e\left(pq\, t^{n-1} b^{-3} f^{\pm 1} v_a\right)}_{S^{(n)a}_{\pm}} \oint \mathrm{d}z_Q\; \underbrace{\Gamma_e(t)^{Q-1} \prod_{n<m}^{Q} \Gamma_e\left(t z_n^{\pm 1} z_m^{\pm 1}\right)}_{A}$$

$$\times \underbrace{\prod_{n=1}^{Q} \Gamma_e\left((pq)^{-\frac{1}{2}} t^{1-Q} b^4 f^{\pm 1} z_n^{\pm 1}\right)}_{P^{\pm}} \underbrace{\prod_{a=1}^{8}\prod_{n=1}^{Q} \Gamma_e\left((pq)^{\frac{1}{2}} b^{-1} v_a^{-1} z_n^{\pm 1}\right)}_{Q_a}, \tag{49}$$

which preserves the symmetry

$$\mathrm{SU}(8)_v \times \mathrm{SU}(2)_f \times \mathrm{U}(1)_b \times \mathrm{U}(1)_t \,. \tag{50}$$

As we already anticipated, we claim that the $\mathrm{SU}(2)_f$ symmetry descends from the isometry of the two-sphere on which we compactified the 6d theory, which manifests itself in 4d as a flavor symmetry. To support this claim, we can compute the anomalies for this symmetry in our four-dimensional model and check that they match those of the $\mathbb{S}^2$ isometry which can be computed from the 8-form anomaly polynomial of the original 6d theory. The anomalies for the flavor symmetries in arbitrary even dimensions that come from the isometries of the compactification manifold can be computed following the strategy of [35]. In Appendix B we apply it to the compactification of the rank $Q$ E-string theory on a two-sphere, so to find the mixed anomalies between its $\mathrm{SU}(2)_{\mathrm{ISO}}$ isometry and the $\mathrm{U}(1)$ symmetries in the Cartan of the $E_8 \times \mathrm{SU}(2)_L$ global symmetry of the 6d theory. In particular, in equation (B.21) we give the anomalies in the basis for the $\mathrm{U}(1)$ symmetries that correspond to the Cartan of the subgroup $\mathrm{U}(1)_c \times \mathrm{SU}(8)_u \subset E_8$. Hence, in order to compare (B.21) with the anomalies of the 4d model we are considering in this section we first need to translate back the fugacities appearing in the index (49) in terms of the original $\mathrm{U}(1)_c \times \mathrm{SU}(8)_u$ fugacities using (36)-(48). By doing so, we find the following anomalies for $\mathrm{SU}(2)_f$:

$$\mathrm{Tr}\left(\mathrm{SU}(2)_f^2 \mathrm{U}(1)_R\right) = Q(Q+1), \quad \mathrm{Tr}\left(\mathrm{SU}(2)_f^2 \mathrm{U}(1)_t\right) = Q(Q-1),$$
$$\mathrm{Tr}\left(\mathrm{SU}(2)_f^2 \mathrm{U}(1)_c\right) = -3Q, \qquad \mathrm{Tr}\left(\mathrm{SU}(2)_f^2 \mathrm{U}(1)_{u_a}\right) = -2Q \,. \tag{51}$$

These perfectly match the anomalies (B.21) that we can compute from 6d for the value of the flux (31). In addition, one can see that the mixing of the original $U(1)_f$ from (4) with $U(1)_c$ that we did above (14) and that gives the correct anomalies matches the mixing prediction given in (B.22).

While the model we have considered is obtained from the $\Phi$-gluing for the octet moment maps as well as the antisymmetric moment maps, one can also consider the $S$-gluing for the antisymmetric moment maps[11]. The corresponding flux is given by

$$\mathcal{F} = (-1/2 - (-1/2); 1+1, 1+1, 1+1, 1+1, 1+1, 1+1, 1+1, -1+(-1))$$
$$= (0; 2, 2, 2, 2, 2, 2, 2, -2), \tag{52}$$

---

[11]The $S$-gluing for the octet moment maps will be considered in the subsequent subsections.

which now has the vanishing $\mathrm{U}(1)_t$ flux. As a result, this model is supposed to inherit the full $\mathrm{SU}(2)_L$ symmetry of E-string rather than just its $\mathrm{U}(1)_t$ subgroup. In addition, this model also has a geometric $\mathrm{SU}(2)_f$ symmetry coming from the isometry of the two-sphere. However, those $\mathrm{SU}(2)_L$ and $\mathrm{SU}(2)_f$ symmetries are not manifest; only $\mathrm{U}(1)_t \times \mathrm{U}(1)_f \subset \mathrm{SU}(2)_L \times \mathrm{SU}(2)_f$ is visible in the Lagrangian description.

Since we now take the $S$-gluing for the antisymmetric moment maps, we introduce the superpotential[12]

$$\mathcal{W}_{glue} = \mathrm{Tr}_z\left(A \cdot \tilde{A}\right) + \sum_{b=1}^{8} \Phi^b\left(L_b - \tilde{L}_b\right),\tag{53}$$

without the additional chiral superfield $\hat{A}$. The total superpotential is then

$$\mathcal{W} = \sum_{n=1}^{Q}\sum_{b=1}^{8}\mathrm{Tr}_z\left(S^{(n)b}A^{Q-n}PQ_b\right) + \sum_{n=1}^{Q}\sum_{b=1}^{8}\mathrm{Tr}_{\tilde{z}}\left(\tilde{S}^{(n)b}\tilde{A}^{Q-n}\tilde{P}Q_b\right) + \mathrm{Tr}_{\tilde{z}}\left(A \cdot \tilde{A}\right),\tag{54}$$

where we have defined as before

$$S^{(n)b} = \begin{cases} R^{(n)b}, & b=1,\ldots,7, \\ b_n, & b=8, \end{cases}\tag{55}$$

$$\tilde{S}^{(n)b} = \begin{cases} \tilde{R}^{(n)b}, & b=1,\ldots,7, \\ -\tilde{b}_n, & b=8, \end{cases}\tag{56}$$

$$Q_b = \begin{cases} L_b, & b=1,\ldots,7, \\ \frac{K-\tilde{K}-\Phi}{3}, & b=8. \end{cases}\tag{57}$$

Note that we don't integrate out $A$ and $\tilde{A}$ for simplicity of the superpotential. Unlike (41), $(S^{(n)b}, \tilde{S}^{(n)b})$ and $(P, \tilde{P})$ do not form doublets of any $\mathrm{SU}(2)$ because $A \neq \tilde{A}$. Thus, the manifest symmetry is only

$$\mathrm{SU}(8)_v \times \mathrm{U}(1)_f \times \mathrm{U}(1)_b \times \mathrm{U}(1)_t.\tag{58}$$

Nevertheless, using the superconformal index, we find that $\mathrm{U}(1)_f \times \mathrm{U}(1)_t$ is actually enhanced to $\mathrm{SU}(2)_f \times \mathrm{SU}(2)_L$ as we will see by computing the superconformal index. The full global symmetry is thus

$$\mathrm{SU}(8)_v \times \mathrm{SU}(2)_f \times \mathrm{U}(1)_b \times \mathrm{SU}(2)_L.\tag{59}$$

The symmetry charges of each chiral multiplet are presented in Table 2.

Let us evaluate the index of the model corresponding to the flux (52), which is given by the following matrix integral:

$$\mathcal{I}_{(0;2^7,-2)} = \oint \mathrm{d}z_Q \prod_{n=1}^{Q}\prod_{a=1}^{8}\Gamma_e\left((pq)^{\frac{1}{2}}c^{\frac{1}{2}}u_a z_n^{\pm 1}\right)$$
$$\times \mathcal{I}_{cap}(z_n;c,tf,f;u_1,\ldots,u_7;x) \times \mathcal{I}_{cap}(z_n;c,pqt^{-1}f^{-1},f^{-1};u_1,\ldots,u_7;x),\tag{60}$$

---

[12]Notice that since the antisymmetric chirals are massive in this case, the theory is just $\mathrm{USp}(2Q)$ with some fundamental chirals, so the rank $Q$ can't be too large. Specifically, since we have 10 fundamental chirals, the theory is free for $Q=3$, as it can be seen from the fact that it's Intriligator–Pouliot dual [36] is just a WZ model, and it is SUSY breaking for $Q > 3$ (notice that in this case the dual would have negative rank). In the following we will study in more details the cases $Q = 1, 2$ by computing the superconformal index.

Table 2: The matter contents of the $SU(8) \times SU(2)^2 \times U(1)$ model and their charges.

| | USp(2Q) | SU(8)$_v$ | U(1)$_f$ | U(1)$_b$ | U(1)$_t$ | U(1)$_R$ |
|---|---|---|---|---|---|---|
| $S^{(n)}$ | **1** | **8** | $2-n$ | $-3$ | $n-1$ | 2 |
| $\tilde{S}^{(n)}$ | **1** | **8** | $n-2$ | $-3$ | $1-n$ | $2n$ |
| $Q$ | **2Q** | $\overline{\textbf{8}}$ | 0 | $-1$ | 0 | 1 |
| $P$ | **2Q** | **1** | $Q-2$ | 4 | $1-Q$ | $-1$ |
| $\tilde{P}$ | **2Q** | **1** | $2-Q$ | 4 | $Q-1$ | $1-2Q$ |
| $A$ | **antisym** | **1** | $-1$ | 0 | 1 | 0 |
| $\tilde{A}$ | **antisym** | **1** | 1 | 0 | $-1$ | 2 |

where we have made a shift of fugacity $t \to tf$. If we use the $SU(8)_v \times U(1)_b \supset SU(7)_u \times U(1)_x \times U(1)_c$ fugacities defined in (48), the index is written as

$$
\mathcal{I}_{(0;2^7,-2)} = \prod_{n=1}^{Q} \prod_{a=1}^{8} \underbrace{\Gamma_e \left( pq t^{n-1} b^{-3} f^{2-n} v_a \right)}_{S^{(n)a}} \prod_{n=1}^{Q} \prod_{a=1}^{8} \underbrace{\Gamma_e \left( pq \left( pqt^{-1} \right)^{n-1} b^{-3} f^{n-2} v_a \right)}_{\tilde{S}^{(n)a}}
$$

$$
\times \oint dz_Q \underbrace{\Gamma_e \left( tf^{-1} \right)^{Q-1} \prod_{n<m}^{Q} \Gamma_e \left( tf^{-1} z_n^{\pm 1} z_m^{\pm 1} \right)}_{A} \times \underbrace{\Gamma_e \left( pqt^{-1}f \right)^{Q-1} \prod_{n<m}^{Q} \Gamma_e \left( pqt^{-1}f z_n^{\pm 1} z_m^{\pm 1} \right)}_{\tilde{A}}
$$

$$
\times \prod_{n=1}^{Q} \underbrace{\Gamma_e \left( (pq)^{-\frac{1}{2}} t^{1-Q} b^4 f^{Q-2} z_n^{\pm 1} \right)}_{P} \prod_{n=1}^{Q} \underbrace{\Gamma_e \left( (pq)^{-\frac{1}{2}} \left( pqt^{-1} \right)^{1-Q} b^4 f^{2-Q} z_n^{\pm 1} \right)}_{\tilde{P}}
$$

$$
\times \prod_{n=1}^{Q} \prod_{a=1}^{8} \underbrace{\Gamma_e \left( (pq)^{\frac{1}{2}} b^{-1} v_a^{-1} z_n^{\pm 1} \right)}_{Q_a} . \tag{61}
$$

The corresponding theory is the USp(2Q) theory with 10 fundamental chirals $(Q_a; P, \tilde{P})$, two massive traceless antisymmetric chirals $(A, \tilde{A})$, $16Q$ gauge singlets $(S^{(n)a}; \tilde{S}^{(n)a})$ and the superpotential (54).

As we mentioned before, we will see using the superconformal index that $U(1)_f$ and $U(1)_t$ are enhanced in the IR to $SU(2)_f$ and $SU(2)_L$ respectively. Moreover, as in the previous example, we claim that $SU(2)_f$ descends from the isometry of the two-sphere on which we performed the compactification. This can be checked by computing the anomalies for $U(1)_f$ in 4d and comparing with those predicted from 6d (B.21). From our 4d Lagrangian description we find

$$
\text{Tr} \left( U(1)_f^2 U(1)_R \right) = -\frac{4}{3} Q(Q+1)(Q-4), \qquad \text{Tr} \left( U(1)_f^2 U(1)_t \right) = 0,
$$
$$
\text{Tr} \left( U(1)_f^2 U(1)_c \right) = 3Q(Q-5), \qquad \text{Tr} \left( U(1)_f^2 U(1)_{u_a} \right) = 2Q(Q-5), \tag{62}
$$

with all the other anomalies non-quadratic in $U(1)_f$ being zero, in agreement with the claim that this symmetry is enhanced to $SU(2)_f$ in the IR. Notice that the mixed anomaly with $U(1)_t$ is also zero, again in agreement with it being enhanced to $SU(2)_L$. Remembering that the anomalies for an SU(2) symmetry are related to those for its U(1) Cartan by

$$
\text{Tr} \left( U(1)^2 U(1)_i \right) = 4 \text{Tr} \left( SU(2)^2 U(1)_i \right), \tag{63}
$$

since 4 is the embedding index of U(1) inside SU(2), we can perfectly match the anomalies (62) computed in 4d with those predicted from 6d (B.21) for the value of the flux (52). In

addition, we find that the mixing of the original $U(1)_f$ from (4) with the other $U(1)$'s giving the correct anomalies matches the mixing prediction in (B.22).

We will now compute the index for small values of $Q$ to see explicitly the various enhancements of symmetry.

**Rank 1**

Now we compute the indices (47) and (60) for $Q = 1$. Note that there is no distinction between (47) and (60) for $Q = 1$ because the traceless antisymmetric representation of USp(2Q) is trivial in this case. Also U(1)$_t$ decouples for $Q = 1$ because no fields are charged under it. The remaining abelian symmetry is U(1)$_b$, whose mixing coefficient with the R-symmetry can be determined by the $a$-maximization [37]. In general, once the mixing coefficients $R_a$ are determined for a set of abelian symmetries $\prod_a U(1)_a$, the R-charge we use for the expansion of the index is given by

$$R = R_0 + \sum_a R_a Q_a \,, \tag{64}$$

where $Q_a$ is the $U(1)_a$ charge and $R_0$ is the trial R-charge we have used to define the index formula. For example, the index (47) is defined with the trial R-charge given in Table 1. Given the $U(1)_a$ fugacity $t_a$, the mixing of R-symmetry with $U(1)_a$ is realized by a shift of the fugacity: $t_a \to t_a(pq)^{R_a/2}$.

In the current example, the mixing coefficient of $U(1)_b$ is given by

$$R_b \approx 0.4269 \,. \tag{65}$$

To avoid clutter due to the irrational value of the mixing coefficient, we will use the following approximate rational value

$$R_b = \frac{3}{7} \tag{66}$$

to evaluate the index, which shouldn't affect the contribution of the conserved current we are interested in because it is in the adjoint representation of the symmetry group and hence independent of $U(1)$ mixing coefficients. In addition, the exact value of the R-charge can be easily implemented if necessary by shifting $U(1)$ fugacities as explained above. With this choice of the mixing coefficient, the index for $Q = 1$ is given by

$$
\begin{aligned}
\mathcal{I}^{Q=1}_{(-1;2^7,-2)} &= \\
&= 1 + b^{-3} \chi^{\mathrm{SU}(2)}_{\mathbf{2}} \chi^{\mathrm{SU}(8)}_{\mathbf{8}} (pq)^{\frac{5}{14}} + b^{-2} \chi^{\mathrm{SU}(8)}_{\overline{\mathbf{28}}} (pq)^{\frac{4}{7}} + \left( b^8 + b^{-6} \left( \chi^{\mathrm{SU}(2)}_{\mathbf{3}} \chi^{\mathrm{SU}(8)}_{\mathbf{36}} + \chi^{\mathrm{SU}(8)}_{\mathbf{28}} \right) \right) (pq)^{\frac{5}{7}} \\
&+ b^{-3} \chi^{\mathrm{SU}(2)}_{\mathbf{2}} \chi^{\mathrm{SU}(8)}_{\mathbf{8}} (pq)^{\frac{5}{14}} (p+q) + b^{-5} \chi^{\mathrm{SU}(2)}_{\mathbf{2}} \chi^{\mathrm{SU}(8)}_{\overline{\mathbf{216}}} (pq)^{\frac{13}{14}} + \left( -\chi^{\mathrm{SU}(8)}_{\mathbf{63}} - \chi^{\mathrm{SU}(2)}_{\mathbf{3}} - 1 \right) pq + \dots \,,
\end{aligned}
\tag{67}
$$

where $\chi^{\mathrm{SU}(2)}_{\mathbf{m}}$ is the character of representation $\mathbf{m}$ of SU(2)$_f$ and $\chi^{\mathrm{SU}(8)}_{\mathbf{n}}$ is the character of representation $\mathbf{n}$ of SU(8)$_v$. Those characters are written in terms of $f$ and $v_a$ respectively. Especially, we highlight the negative contributions at order $pq$, which correspond to the conserved current multiplet [38]. We find that they are in the adjoint representation of

$$\mathrm{SU}(8)_v \times \mathrm{SU}(2)_f \times \mathrm{U}(1)_b \,. \tag{68}$$

In addition to checking that the global symmetry of the theory is at least the one preserved by the flux, there is another important test that we can make for our proposal that this model

corresponds to the compactification of the 6d E-string theory on a sphere with flux. We expect indeed that the 4d theory possesses some gauge invariant operators that descend from the stress-energy tensor and the conserved current multiplets of the parent 6d theory [39]. We refer the reader to Appendix E of [8], in particular equation (E.2), for a review on how these operators should contribute to the 4d index. Here we shall just briefly state the result and apply it to our case.

Suppose that the flux used in the compactification preserves a subgroup $G$ of the 6d global symmetry. Then, from the 6d conserved currents we get operators with R-charge 2 under the 6d R-symmetry and in representations of $G$ resulting from the decomposition of the adjoint representation of the 6d global symmetry under the subgroup $G$. Moreover, these operators appear in the index with multiplicity $g - 1$ if they are in the adjoint representation of $G$ and $g - 1 - qF$ otherwise, where $g$ is the genus of the Riemann surface, $q$ is the charge of the operator under the U(1) for which we turned on a flux and $F$ is the value of the flux for such U(1). In our case of sphere compactification we have $g - 1 = -1$ and $g - 1 - qF = -1 - qF$, where the sign of these numbers determines whether the corresponding operator is bosonic or fermionic.

For the case at hand, we need to consider the branching rule for the adjoint representation of $E_8$ with respect to its $SU(8) \times U(1)$ subgroup:

$$\mathbf{248} \to \mathbf{1}^0 \oplus \mathbf{63}^0 \oplus \overline{\mathbf{56}}^1 \oplus \mathbf{56}^{-1} \oplus \mathbf{28}^2 \oplus \overline{\mathbf{28}}^{-2} \oplus \overline{\mathbf{8}}^3 \oplus \mathbf{8}^{-3}. \tag{69}$$

The 6d R-symmetry assigns R-charge 1 to the octet fields $Q_a$ and so it is related to the R-symmetry we used for computing the index (67) by the shift $b \to b(pq)^{-\frac{3}{14}}$. Moreover, the U(1) inside the 6d global symmetry for which we turned on a unit of flux $F = 1$[13] is related to the $U(1)_b$ symmetry of our 4d model. With this dictionary, we can immediately identify all the states appearing in (69) that contribute to the index (67) up to the order we evaluated it:

$$
\begin{aligned}
\mathbf{8}^{-3} &\to 2b^{-3}\chi_{\mathbf{8}}^{SU(8)}(pq)^{\frac{5}{14}} \\
\overline{\mathbf{28}}^{-2} &\to b^{-2}\chi_{\overline{\mathbf{28}}}^{SU(8)}(pq)^{\frac{4}{7}} \\
\mathbf{1}^0 \oplus \mathbf{63}^0 &\to -(\chi_{\mathbf{63}}^{SU(8)} + 1)pq.
\end{aligned}
\tag{70}
$$

Notice that the state $\mathbf{56}^{-1}$ doesn't contribute with any operator since in this case $-1 - qF = 0$. Moreover, the state $\mathbf{8}^{-3}$ contributes with two operators, which are distinguished in our index (109) by the quantum number for the geometric $SU(2)_f$ symmetry.

**Rank 2**

Next let us compute the indices for $Q = 2$. Now the abelian symmetry is $U(1)_b \times U(1)_t$. We use the mixing coefficients:

$$R_t \approx 0.2892 \approx \frac{2}{7}, \qquad R_b \approx 0.4707 \approx \frac{7}{15}, \tag{71}$$

---

[13]We choose to work in a normalization for this U(1) such that the minimal flux is 1. Our model is the one with the minimal value of flux that preserves $SU(8) \times U(1)$, since it was constructed by gluing two caps together without inserting any tube in the middle, which would have increased the flux.

which give rise to the following expansion of the index:

$$
\begin{aligned}
\mathcal{I}^{Q=2}_{(-1;2^7,-2)} = {} & 1 + t^2(pq)^{\frac{2}{7}} + b^{-3}\chi_{\mathbf{2}}^{\mathrm{SU(2)}}\chi_{\mathbf{8}}^{\mathrm{SU(8)}}(pq)^{\frac{3}{10}} + tb^{-3}\chi_{\mathbf{2}}^{\mathrm{SU(2)}}\chi_{\mathbf{8}}^{\mathrm{SU(8)}}(pq)^{\frac{31}{70}} \\
& + b^{-2}\chi_{\overline{\mathbf{28}}}^{\mathrm{SU(8)}}(pq)^{\frac{8}{15}} + t^4(pq)^{\frac{4}{7}} + t^{-2}b^8(pq)^{\frac{61}{105}} + t^2 b^{-3}\chi_{\mathbf{2}}^{\mathrm{SU(2)}}\chi_{\mathbf{8}}(pq)^{\frac{41}{70}} \\
& + b^{-6}\left(\chi_{\mathbf{28}}^{\mathrm{SU(8)}} + \chi_{\mathbf{3}}^{\mathrm{SU(2)}}\chi_{\mathbf{36}}^{\mathrm{SU(8)}}\right)(pq)^{\frac{3}{5}} + tb^{-2}\chi_{\overline{\mathbf{28}}}^{\mathrm{SU(8)}}(pq)^{\frac{71}{105}} + t^{-1}b^8(pq)^{\frac{76}{105}} \\
& + t^3 b^{-3}\chi_{\mathbf{2}}^{\mathrm{SU(2)}}\chi_{\mathbf{8}}^{\mathrm{SU(8)}}(pq)^{\frac{51}{70}} + tb^{-6}\left(1+\chi_{\mathbf{3}}^{\mathrm{SU(2)}}\right)\left(\chi_{\mathbf{28}}^{\mathrm{SU(8)}} + \chi_{\mathbf{36}}^{\mathrm{SU(8)}}\right)(pq)^{\frac{26}{35}} \\
& + t^2(pq)^{\frac{2}{7}}(p+q) + \cdots + \left(-\chi_{\mathbf{63}}^{\mathrm{SU(8)}} - \chi_{\mathbf{3}}^{\mathrm{SU(2)}} - 2\right)pq + \dots .
\end{aligned}
\tag{72}
$$

We find that the contribution of the conserved current, which is highlighted in blue, is in the adjoint representation of

$$
\mathrm{SU(8)}_v \times \mathrm{SU(2)}_f \times \mathrm{U(1)}_t \times \mathrm{U(1)}_b .
\tag{73}
$$

This is the same as the manifest symmetry of the Lagrangian description we found.

Again, on top of checking that the global symmetry of the model is the one preserved by the flux, we can also check the presence of the operators coming from the 6d conserved currents. In this case we also have operators charged under $\mathrm{U(1)}_t$ coming from the conserved current for $\mathrm{SU(2)}_L$, according to the branching rule

$$
\mathbf{3}_{\mathrm{SU(2)}_L} \to \mathbf{1}^0 \oplus \mathbf{1}^{\pm 2} .
\tag{74}
$$

The 6d R-symmetry assigns R-charge 1 to the fundamentals $Q_a$ and also to the antisymmetric field $A$. Hence, it is related to the R-symmetry we used for computing the index (72) by the shifts $b \to b(pq)^{-\frac{7}{30}}$ and $t \to t(pq)^{\frac{5}{14}}$. Moreover, the $\mathrm{U(1)} \subset \mathrm{SU(2)}_L$ for which we turned on flux $F_t = -1$ is related to the $\mathrm{U(1)}_t$ symmetry of our 4d model, while the $\mathrm{U(1)} \subset \mathrm{E}_8$ for which we turned flux $F = 1$ is related to $\mathrm{U(1)}_b$. With this dictionary, we can immediately identify all the states appearing in (69) and (74) that contribute to the index (72) up to the order we evaluated it:

$$
\begin{aligned}
\mathbf{8}^{(-3,0)} \quad &\to \quad 2b^{-3}\chi_{\mathbf{8}}^{\mathrm{SU(8)}}(pq)^{\frac{3}{10}} \\
\overline{\mathbf{28}}^{(-2,0)} \quad &\to \quad b^{-2}\chi_{\overline{\mathbf{28}}}^{\mathrm{SU(8)}}(pq)^{\frac{8}{15}} \\
\mathbf{1}^{(0,2)} \quad &\to \quad t^2(pq)^{\frac{2}{7}} \\
2 \times \mathbf{1}^{(0,0)} \oplus \mathbf{63}^{(0,0)} \quad &\to \quad -(\chi_{\mathbf{63}}^{\mathrm{SU(8)}} + 2)pq ,
\end{aligned}
\tag{75}
$$

where at the exponent of the states on the left hand side we reported, in order, the charges under $\mathrm{U(1)}_b$ and $\mathrm{U(1)}_t$. Notice again that the state $\mathbf{8}^{(-3,0)}$ contributes with two operators, which transform as a doublet under the geometric $\mathrm{SU(2)}_f$ symmetry.

In fact, one can see that there are operators violating the unitarity bound $R = \frac{2}{3}$, which corresponds to $(pq)^{\frac{1}{3}}$. For instance, the first nontrivial term in (72), which corresponds to $\mathrm{Tr}A^2$, is below this bound. Repeating the $a$-maximization after flipping a unitarity violating operator whenever it appears, we have found that the decoupled operators are $\mathrm{Tr}A^2$ and $S^{(1)}$, which correspond to the first two nontrivial terms of the index (72) respectively. We have also found that the remaining interacting sector still exhibits the same symmetry as (73).

On the other hand, the index (60) with the $S$-gluing is evaluated with different mixing coefficients:

$$
R_t = 1 , \qquad R_b \approx 0.6321 ,
\tag{76}
$$

where the latter is approximated by

$$R_b = \frac{7}{13}.$$ (77)

The resulting index is

$$
\begin{aligned}
\mathcal{I}^{Q=2}_{(0;2^7,-2)} = {} & 1 + b^8(pq)^{\frac{2}{13}} + b^{-3}\chi^{SU(2)}_2 \chi^{SU(8)}_8 (pq)^{\frac{5}{26}} + b^{16}(pq)^{\frac{4}{13}} + b^5\chi^{SU(2)}_2\chi^{SU(8)}_8(pq)^{\frac{9}{26}} \\
& + b^{-6}\left(\chi^{SU(8)}_{28} + \chi^{SU(2)}_3\chi^{SU(8)}_{36}\right)(pq)^{\frac{5}{13}} + \left(b^{24} + b^{-2}\chi^{SU(8)}_{\overline{28}}\right)(pq)^{\frac{6}{13}} \\
& + b^{13}\chi^{SU(2)}_2\chi^{SU(8)}_8(pq)^{\frac{1}{2}} + b^2\left(\chi^{SU(8)}_{28} + \chi^{SU(2)}_3\chi^{SU(8)}_{36}\right)(pq)^{\frac{7}{13}} \\
& + b^{-9}\left(\chi^{SU(2)}_2\chi^{SU(8)}_{168} + \chi^{SU(2)}_4\chi^{SU(8)}_{120}\right)(pq)^{\frac{15}{26}} + \left(b^{32} + b^6\chi^{SU(8)}_{\overline{28}}\right)(pq)^{\frac{8}{13}} \\
& + b^8(pq)^{\frac{2}{13}}(p+q) + \left(b^{21}\chi^{SU(2)}_2\chi^{SU(8)}_8 + b^{-5}\chi^{SU(2)}_2\left(\chi^{SU(8)}_{\overline{8}} + \chi^{SU(8)}_{\overline{216}}\right)\right)(pq)^{\frac{17}{26}} \\
& + \cdots + 2b^{13}\chi^{SU(2)}_2\chi^{SU(8)}_8(pq)^{\frac{1}{2}}(p+q) + \left(-\chi^{SU(8)}_{63} - \chi^{SU(2)}_3 - \chi^{SU(2)_t}_3 - 1\right. \\
& \left. + b^{26}\left(\chi^{SU(8)}_{28} + \chi^{SU(2)}_3\chi^{SU(8)}_{36}\right) + \chi^{SU(8)}_{720} + \chi^{SU(2)}_3\chi^{SU(8)}_{945}\right)pq + \ldots,
\end{aligned}
$$ (78)

where one can see that the manifest symmetry $SU(8)_v \times U(1)_f \times U(1)_t \times U(1)_b$ is indeed enhanced to

$$SU(8)_v \times SU(2)_f \times SU(2)_t \times U(1)_b.$$ (79)

Again, the operators appearing in the index confirm our expectation from $6d$. With respect to the case of $\Phi$-gluing, the flux preserves the $SU(2)_L$ $6d$ symmetry this time, which is related to $SU(2)_t$ of our 4d model. Moreover, in this case the 6d R-symmetry is related to the one we used for computing the index (78) by the shift $b \to b(pq)^{-\frac{7}{26}}$. With this dictionary, we can immediately identify all the states that are expected from the 6d conserved currents according to the branching rule (69) in the index (78) up to the order we evaluated it:

$$
\begin{aligned}
(\mathbf{8},\mathbf{1})^{-3} &\to 2b^{-3}\chi^{SU(8)}_8(pq)^{\frac{5}{26}} \\
(\overline{\mathbf{28}},\mathbf{1})^{-2} &\to b^{-2}\chi^{SU(8)}_{\overline{28}}(pq)^{\frac{6}{13}} \\
(\mathbf{1},\mathbf{1})^0 \oplus (\mathbf{63},\mathbf{1})^0 \oplus (\mathbf{1},\mathbf{3})^0 &\to -\left(\chi^{SU(8)}_{63} + \chi^{SU(2)_t}_3 + 1\right)pq,
\end{aligned}
$$ (80)

where in the states on the left hand side we reported, in order, the representations under $SU(8)$ and $SU(2)_L$. Notice again that the state $(\mathbf{8},\mathbf{1})^{-3}$ contributes with two operators, which transform as a doublet under the geometric $SU(2)_f$ symmetry.

We have some operators violating the unitarity bound $R = \frac{2}{3}$:

$$\mathrm{Tr}P\tilde{P}, \qquad (S^{(1)}, \tilde{S}^{(1)}),$$ (81)

which correspond to the first two nontrivial terms of the index (78). Once those are flipped, the interacting sector turns out to be independent of $SU(2)_f$ and only exhibits

$$SU(8) \times SU(2)_t \times U(1)_b.$$ (82)

Note that we still have nontrivial enhancement from $U(1)_t$ to $SU(2)_t$. On the other hand, the geometric $SU(2)_f$ symmetry is realized in the decoupled sector because $(S^{(1)}, \tilde{S}^{(1)})$ form a doublet of $SU(2)_f$.

## 3.2 $E_6 \times SU(2) \times U(1)^2$ **sphere model**

The next example is the model corresponding to the compactification of the E-string theory on a sphere with flux

$$\mathcal{F} = (-1; 0, 0, 2, 2, 2, 2, 2, -2), \tag{83}$$

which preserves $E_6 \times SU(2)_v \times U(1)_b \times U(1)_t \subset E_8 \times SU(2)_L$. Such flux can be achieved by gluing two basic caps, taking the $S$-gluing for the first two octet moment maps[14] and the $\Phi$-gluing for the other octet moment maps and the antisymmetric moment maps[15]. As this model also turns out to have the geometric $SU(2)_f$ symmetry, the total global symmetry of the theory is given by

$$E_6 \times SU(2)_v \times SU(2)_f \times U(1)_b \times U(1)_t. \tag{84}$$

First recall that the basic cap is given by the WZ model with the superpotential

$$\mathcal{W}_{cap} = \sum_{n=1}^{Q} \sum_{a=1}^{7} \mathrm{Tr}_z \left( R^{(n)a} A^{Q-n} P L_a \right) + \sum_{n=1}^{Q} \mathrm{Tr}_z \left( b_n A^{Q-n} P K \right) + K L_8, \tag{85}$$

which preserves

$$USp(2Q) \times SU(7)_u \times U(1)_x \times U(1)_c \times U(1)_t \times U(1)_f. \tag{86}$$

As we mentioned, we take the $S$-gluing for $L_{1,2}$ and the $\Phi$-gluing for $L_{3,\dots,8}$ and $A$. Namely, we introduce $\hat{A}$, $\Phi^{3,\dots,8}$ and the superpotential

$$\mathcal{W}_{glue} = \mathrm{Tr}_z \left[ \hat{A} \cdot (A - \tilde{A}) \right] + \sum_{a=1}^{2} L_a \tilde{L}^a + \sum_{b=3}^{8} \Phi^b \left( L_b - \tilde{L}_b \right) \tag{87}$$

and gauge the puncture symmetry $USp(2Q)$. The entire superpotential of the glued theory is given by

$$\begin{aligned}
\mathcal{W} &= \mathcal{W}_{cap} + \tilde{\mathcal{W}}_{cap} + \mathcal{W}_{glue} \\
&= \sum_{n=1}^{Q} \sum_{a=1}^{7} \mathrm{Tr}_z \left( R^{(n)a} A^{Q-n} P L_a \right) + \sum_{n=1}^{Q} \mathrm{Tr}_z \left( b_n A^{Q-n} P K \right) + K L_8 \\
&\quad + \sum_{n=1}^{Q} \sum_{a=1}^{2} \mathrm{Tr}_z \left( \tilde{R}^{(n)}{}_a \tilde{A}^{Q-n} \tilde{P} \tilde{L}^a \right) + \sum_{n=1}^{Q} \sum_{a=3}^{7} \mathrm{Tr}_z \left( \tilde{R}^{(n)a} \tilde{A}^{Q-n} \tilde{P} \tilde{L}_a \right) + \sum_{n=1}^{Q} \mathrm{Tr}_z \left( \tilde{b}_n \tilde{A}^{Q-n} \tilde{P} \tilde{K} \right) \\
&\quad + \tilde{K} \tilde{L}_8 + \mathrm{Tr}_z \left[ \hat{A} \cdot (A - \tilde{A}) \right] + \sum_{a=1}^{2} L_a \tilde{L}^a + \sum_{b=3}^{8} \Phi^b \left( L_b - \tilde{L}_b \right),
\end{aligned} \tag{88}$$

---

[14]In this and the next example, we choose an even number of octet moment maps to be $S$-glued. This is because if an odd number of octet moment maps are $S$-glued, the resulting theory has the $USp(2Q)$ gauge group with an odd number of fundamental chirals, which lead to an inconsistency of the theory due to the Witten anomaly [40]. In general, one has to take the gluing rule in such a way that the resulting theory is a consistent anomaly-free theory.

[15]One may wonder about the possibility for $Q > 1$ of performing an $S$-gluing for the antisymmetric moment maps, which would give flux $n_t = 0$ that preserves the $SU(2)_L$ symmetry. It turns out that the resulting model doesn't flow to an SCFT. In order to avoid this problem one should consider a sphere compactification with higher values of the fluxes $n_c$, $n_a$, which can be achieved by connecting the two caps with an even number of tubes. The resulting model is too complicated to analyze with the superconformal index, so we will neglect this possibility in what follows.

Table 3: The matter contents of the $E_6 \times SU(2)^2 \times U(1)^2$ model and their charges.

|  | USp(2Q) | SU(2)$_v$ | SU(6)$_w$ | SU(2)$_d$ | SU(2)$_f$ | U(1)$_b$ | U(1)$_t$ | U(1)$_R$ |
|---|---|---|---|---|---|---|---|---|
| $S^{(n)}$ | 1 | 1 | 6 | 2 | 1 | −2 | $n-1$ | 2 |
| $(R^{(n)}, \tilde{R}^{(n)})$ | 1 | 2 | 1 | 1 | 2 | −3 | $n-1$ | 2 |
| $Q$ | 2Q | 1 | $\bar{6}$ | 1 | 1 | −1 | 0 | 1 |
| $P$ | 2Q | 1 | 1 | 2 | 1 | 3 | $1-Q$ | −1 |
| $A$ | antisym | 1 | 1 | 1 | 1 | 0 | 1 | 0 |

where $\mathrm{Tr}_z$ is the trace over the gauged puncture symmetry USp(2Q). After integrating out the massive fields, we obtain the USp(2Q) theory with one traceless antisymmetric $A$, 8 fundamentals $(Q_b; P^{\pm})$, 16Q gauge singlets $(R^{(n)a}, \tilde{R}^{(n)}{}_a; S^{(n)b}{}_{\pm})$ and the superpotential

$$\mathcal{W} = \sum_{n=1}^{Q} \sum_{b=3}^{8} \sum_{\alpha=\pm} \mathrm{Tr}_z \left( S^{(n)b}{}_{\alpha} A^{Q-n} P^{\alpha} Q_b \right) - \sum_{m,n=1}^{Q} \sum_{a=1}^{2} \mathrm{Tr}_z \left( A^{2Q-m-n} P^+ P^- R^{(m)a} \tilde{R}^{(n)}{}_a \right), \quad (89)$$

where we have defined

$$S^{(n)b}{}_{+} = \begin{cases} R^{(n)b}, & b = 3, \ldots, 7, \\ b_n, & b = 8, \end{cases} \quad (90)$$

$$S^{(n)b}{}_{-} = \begin{cases} \tilde{R}^{(n)b}, & b = 3, \ldots, 7, \\ -\tilde{b}_n, & b = 8, \end{cases} \quad (91)$$

$$Q_b = \begin{cases} L_b, & b = 3, \ldots, 7, \\ \frac{K - \check{K} - \Phi}{3}, & b = 8, \end{cases} \quad (92)$$

$$P^+ = P, \quad (93)$$

$$P^- = \tilde{P}. \quad (94)$$

The superpotential (89) preserves

$$SU(2)_v \times SU(6)_w \times SU(2)_d \times SU(2)_f \times U(1)_b \times U(1)_t. \quad (95)$$

The charges of each chiral multiplet are presented in Table 3. In particular, using the superconformal index, we will show that $SU(6)_w \times SU(2)_d$ is enhanced to $E_6$. Thus, the enhanced global symmetry of the theory is given by

$$E_6 \times SU(2)_v \times SU(2)_f \times U(1)_b \times U(1)_t. \quad (96)$$

To evaluate the index, we first note that the gluing we take breaks $SU(7)_u \times U(1)_x$ of the basic cap into $SU(2)_v \times SU(5)_u \times U(1)_y \times U(1)_x$. Thus, we need to redefine the fugacities as follows:

$$\begin{aligned} u_1 &\longrightarrow x^{-1} y^{-5} v, \\ u_2 &\longrightarrow x^{-1} y^{-5} v^{-1}, \\ u_{b=3,\ldots,7} &\longrightarrow x^{-1} y^2 u_b, \\ u_8 &\longrightarrow x^7, \end{aligned} \quad (97)$$

where $u_b$ on the right hand side satisfies $\prod_{b=3}^{7} u_b = 1$. The octet fugacities of the caps are then given by

$$a_i = \left( c^{\frac{1}{2}} x^{-1} y^{-5} v, c^{\frac{1}{2}} x^{-1} y^{-5} v^{-1}; c^{\frac{1}{2}} x^{-1} y^2 u_b; c^{\frac{1}{2}} x^7 \right), \quad (98)$$

for the left cap and

$$b_i = \left( c'^{\frac{1}{2}} x'^{-1} y'^{-5} v', c'^{\frac{1}{2}} x'^{-1} y'^{-5} v'^{-1}; c'^{\frac{1}{2}} x'^{-1} y'^2 u'_b; c'^{\frac{1}{2}} x'^7 \right), \tag{99}$$

for the right cap. Since we take the $S$-gluing for the first two components and the $\Phi$-gluing for the others, we impose the conditions $a_i = b_i^{-1}$ for $i = 1, 2$ and $a_i = b_i$ otherwise, which are solved by

$$\begin{aligned}
c' &= c^{\frac{1}{2}} x y^5, \\
x' &= c^{\frac{1}{28}} x^{\frac{13}{14}} y^{-\frac{5}{14}}, \\
y' &= c^{\frac{1}{7}} x^{-\frac{2}{7}} y^{-\frac{3}{7}}, \\
v' &= v^{-1}, \\
u'_b &= u_b.
\end{aligned} \tag{100}$$

With this identification, we obtain the index of the theory on a sphere with the flux (83), which is given by

$$\begin{aligned}
\mathcal{I}_{(-1;0^2,2^5,-2)} &= \\
&= \oint dz_Q \, \Gamma_e \left( pq t^{-1} \right)^{Q-1} \prod_{n<m}^{Q} \Gamma_e \left( pq t^{-1} z_n^{\pm 1} z_m^{\pm 1} \right) \prod_{n=1}^{Q} \prod_{a=3}^{8} \Gamma_e \left( (pq)^{\frac{1}{2}} c^{\frac{1}{2}} u_a z_n^{\pm 1} \right) \\
&\quad \times \mathcal{I}_{cap}(z_n; c, t, f; u_1, \ldots, u_7; x) \\
&\quad \times \left( \mathcal{I}_{cap}(z_n; c, t, f^{-1}; u_1, \ldots, u_7; x) \Big|_{c \to c^{\frac{1}{2}} x y^5, x \to c^{\frac{1}{28}} x^{\frac{13}{14}} y^{-\frac{5}{14}}, y \to c^{\frac{1}{7}} x^{-\frac{2}{7}} y^{-\frac{3}{7}}, v \to v^{-1}} \right), \tag{101}
\end{aligned}$$

where (97) is understood for $u_a$. Furthermore, introducing the following redefinition of the fugacities:

$$\begin{aligned}
w_{a=1,\ldots,5} &= c^{\frac{1}{6}} x y^{\frac{1}{3}} u_{a+2}, \\
w_6 &= c^{-\frac{5}{6}} x^{-5} y^{-\frac{5}{3}}, \\
b &= c^{\frac{1}{3}} x^{-2} y^{\frac{5}{3}}, \\
d &= c^{-\frac{1}{2}} x y^5 f^{-1},
\end{aligned} \tag{102}$$

one can see that the index has the manifest $SU(2)_v \times SU(6)_w \times SU(2)_d \times SU(2)_f \times U(1)_b \times U(1)_t$ symmetry as follows:

$$\begin{aligned}
\mathcal{I}_{(-1;0^2,2^5,-2)} \\
&= \prod_{n=1}^{Q} \underbrace{\Gamma_e \left( pq t^{n-1} b^{-3} f^{\pm 1} v^{\pm 1} \right)}_{(R^{(n)\pm}, \pm \tilde{R}^{(n)}_{\mp})} \prod_{n=1}^{Q} \prod_{a=1}^{6} \underbrace{\Gamma_e \left( pq t^{n-1} b^{-2} d^{\pm 1} w_a \right)}_{S_{\pm}^{(n)a}} \\
&\quad \times \oint dz_Q \, \underbrace{\Gamma_e(t)^{Q-1} \prod_{n<m} \Gamma_e \left( t z_n^{\pm 1} z_m^{\pm 1} \right)}_{A} \prod_{n=1}^{Q} \prod_{a=1}^{6} \underbrace{\Gamma_e \left( (pq)^{\frac{1}{2}} b^{-1} w_a^{-1} z_n^{\pm 1} \right)}_{Q_a} \\
&\quad \times \prod_{n=1}^{Q} \underbrace{\Gamma_e \left( (pq)^{-\frac{1}{2}} t^{1-Q} b^3 d^{\pm 1} z_n^{\pm 1} \right)}_{P^{\pm}}. \tag{103}
\end{aligned}$$

We comment again that the $SU(2)_f$ symmetry descends from the isometry of the $\mathbb{S}^2$ on which we performed the compactification. The anomalies for this symmetry computed using our 4d Lagrangian description are

$$\text{Tr}\left(SU(2)_f^2 U(1)_R\right) = \frac{Q(Q+1)}{2}, \quad \text{Tr}\left(SU(2)_f^2 U(1)_t\right) = \frac{Q(Q-1)}{2},$$

$$\text{Tr}\left(SU(2)_f^2 U(1)_c\right) = -Q, \text{Tr}\left(SU(2)_f^2 U(1)_{u_a}\right) = -\frac{Q}{2} \quad a = 1, 2,$$

$$\text{Tr}\left(SU(2)_f^2 U(1)_{u_a}\right) = -Q \quad a = 3, \cdots, 7, \tag{104}$$

where we went back from the U(1)'s used to write the index (103) to those parametrizing the Cartan of $U(1)_c \times SU(8)_u \subset E_8$ using (97)-(102). These anomalies perfectly match those that we can compute from 6d (B.21) for the value of the flux (83). In this case as well, the mixing of the original $U(1)_f$ from (4) with the other $U(1)$'s that gives the correct anomalies matches the mixing prediction in (B.22).

While the manifest UV symmetry visible in the integral expression is

$$SU(2)_v \times SU(6)_w \times SU(2)_d \times SU(2)_f \times U(1)_b \times U(1)_t, \tag{105}$$

the expanded index will show that the global symmetry is enhanced to

$$E_6 \times SU(2)_v \times SU(2)_f \times U(1)_b \times U(1)_t. \tag{106}$$

Notice also that the $SU(2)_v$ and the $SU(2)_f$ symmetries only act on the singlets $R^{(n)\pm}$ and $\tilde{R}^{(n)}{}_\pm$. Removing these fields we get a model with $E_6 \times U(1)_b \times U(1)_t$ symmetry. This turns out to be dual under the Intriligator–Pouliot duality [36] to the model which was observed to have $E_6$ enhancement in [41] for $Q = 1$ and in [22] for arbitrary $Q$ up to some extra flips of operators neutral under $SU(2)_v \times SU(2)_f$. Indeed, applying the $a$-maximization, we will see that the flipped operators will decouple in the IR as their R-charges fall below the unitarity bound. Then the interacting sector corresponds exactly to the $E_6$ models in [41] and [22].

Let us again analyze the index of low values of $Q$ in more detail.

**Rank 1**

We first compute the index (103) for $Q = 1$ with the $U(1)_b$ mixing coefficient:

$$R_b \approx 0.5117, \tag{107}$$

which is approximated by the following rational value

$$R_b = \frac{1}{2}. \tag{108}$$

The resulting index is

$$\mathcal{I}_{(-1;0^2,2^5,-2)}^{Q=1} =$$
$$= 1 + b^{-3} \chi_2^{SU(2)_v} \chi_2^{SU(2)_f} (pq)^{\frac{1}{4}} + \left(b^6 + b^{-6}\left(1 + \chi_3^{SU(2)_v} \chi_3^{SU(2)_f}\right) + b^{-2} \chi_{\mathbf{27}}^{E_6}\right)(pq)^{\frac{1}{2}}$$
$$+ b^{-3} \chi_2^{SU(2)_v} \chi_2^{SU(2)_f} (pq)^{\frac{1}{4}} (p+q)$$
$$+ \left(b^{-9}\left(\chi_2^{SU(2)_v} \chi_2^{SU(2)_f} + \chi_4^{SU(2)_v} \chi_4^{SU(2)_f}\right) + b^{-5} \chi_2^{SU(2)_v} \chi_2^{SU(2)_f} \chi_{\mathbf{27}}^{E_6}\right)(pq)^{\frac{3}{4}}$$
$$+ \left(b^6 + b^{-6}\left(1 + \chi_2^{SU(3)_v}\right)\left(1 + \chi_3^{SU(2)_f}\right) + b^{-2} \chi_{\mathbf{27}}^{E_6}\right)(pq)^{\frac{1}{2}}(p+q)$$
$$+ \left(-\chi_{\mathbf{78}}^{E_6} - \chi_3^{SU(2)_v} - \chi_3^{SU(2)_f} - 1 + b^{12} + b^{-12}\left(1 + \chi_3^{SU(2)_v} \chi_3^{SU(2)_f} + \chi_5^{SU(2)_v} \chi_5^{SU(2)_f}\right)\right.$$
$$\left. + b^{-8}\left(1 + \chi_3^{SU(2)_v} \chi_3^{SU(2)_f}\right) \chi_{\mathbf{27}}^{E_6} + b^{-4} \chi_{\mathbf{351'}}^{E_6}\right) pq + \cdots, \tag{109}$$

where $\chi_{\mathbf{n}}^{\mathrm{SU}(2)_v}$ denotes the character of the representation $\mathbf{n}$ of SU(2)$_v$ written in terms of the fugacity $v$, $\chi_{\mathbf{n}}^{\mathrm{SU}(2)_f}$ denotes the character of the representation $\mathbf{n}$ of SU(2)$_f$ written in terms of the fugacity $f$, and $\chi_{\mathbf{m}}^{\mathrm{E}_6}$ denotes the character of the representation $\mathbf{m}$ of E$_6$ written in terms of the fugacities $w_a$ and $d$. The negative terms at order $pq$ highlighted in blue represent the conserved currents for the global symmetry of the theory. We can see that the IR global symmetry is indeed

$$\mathrm{E}_6 \times \mathrm{SU}(2)_v \times \mathrm{SU}(2)_f \times \mathrm{U}(1)_b. \tag{110}$$

Also in this case we can check the presence of gauge invariant operators that descend from the 6d conserved currents. For rank 1 we focus on the branching rule of the adjoint representation of E$_8$ under the E$_6 \times$ SU(2) $\times$ U(1) subgroup:

$$\mathbf{248} \rightarrow (\mathbf{1},\mathbf{1})^0 \oplus (\mathbf{1},\mathbf{3})^0 \oplus (\mathbf{78},\mathbf{1})^0 \oplus (\overline{\mathbf{27}},\mathbf{2})^1 \oplus (\mathbf{27},\mathbf{2})^{-1} \oplus (\mathbf{27},\mathbf{1})^2 \oplus (\overline{\mathbf{27}},\mathbf{1})^{-2} \oplus (\mathbf{1},\mathbf{2})^{\pm 2}. \tag{111}$$

The 6d R-symmetry is related to the one we used to compute the index (109) by the shift $b \rightarrow b(pq)^{\frac{1}{4}}$. Moreover, the U(1) inside E$_8$ for which we turned on a unit of flux is related to U(1)$_b$. With this dictionary, we can identify the states appearing in (111) in the index (109) up to the order we evaluated it:

$$\begin{aligned}
(\mathbf{1},\mathbf{2})^{-3} &\rightarrow & 2b^{-3}\chi_{\mathbf{2}}^{\mathrm{SU}(2)_v}(pq)^{\frac{1}{4}} \\
(\overline{\mathbf{27}},\mathbf{1})^{-2} &\rightarrow & b^{-2}\chi_{\overline{\mathbf{27}}}^{\mathrm{E}_6}(pq)^{\frac{1}{2}} \\
(\mathbf{1},\mathbf{1})^0 \oplus (\mathbf{1},\mathbf{3})^0 \oplus (\mathbf{78},\mathbf{1})^0 &\rightarrow & -\left(\chi_{\mathbf{78}}^{\mathrm{E}_6} + \chi_{\mathbf{3}}^{\mathrm{SU}(2)_v} + 1\right)pq.
\end{aligned} \tag{112}$$

Notice again that, similarly to the example of the previous subsection, the state $(\mathbf{1},\mathbf{2})^{-3}$ contributes with two operators, which transform as a doublet under the geometric SU(2)$_f$.

The $a$-maximization tells us that the following operators decouple in the IR:

$$(R^{(1)\pm}, \tilde{R}^{(1)\pm}), \tag{113}$$

which are in $(\mathbf{2},\mathbf{2})$ of SU(2)$_v \times$ SU(2)$_f$. In fact, as we mentioned, the singlets $(R^{(1)\pm}, \tilde{R}^{(1)\pm})$ are the only chiral multiplets charged under SU(2)$_v \times$ SU(2)$_f$. Thus, the remaining interacting sector is independent of SU(2)$_v \times$ SU(2)$_f$. Indeed, the interacting sector is the Intriligator–Pouliot dual theory [36] of the $E_6$ model in [41], which was observed to have the enhanced IR symmetry

$$\mathrm{E}_6 \times \mathrm{U}(1)_b. \tag{114}$$

**Rank 2**

For rank $Q = 2$ we compute the index (103) with the following mixing coefficients:

$$R_t \approx 0.2221 \approx \frac{1}{4}, \qquad R_b \approx 0.5539 \approx \frac{4}{7}. \tag{115}$$

The expansion of the index up to order $pq$ is much more complicated than for the rank 1 case. For simplicity we only show the first few orders and the order $pq$ containing the conserved currents, but we checked that also the other terms organize into characters of the expected

global symmetry

$$\mathcal{I}^{Q=2}_{(-1;0^2,2^5,-2)} =$$

$$= 1 + b^{-3}\chi_{\mathbf{2}}^{\text{SU(2)}_v}\chi_{\mathbf{2}}^{\text{SU(2)}_f}(pq)^{\frac{1}{7}} + t^2(pq)^{\frac{1}{4}} + b^{-3}t\chi_{\mathbf{2}}^{\text{SU(2)}_v}\chi_{\mathbf{2}}^{\text{SU(2)}_f}(pq)^{\frac{15}{56}}$$

$$+ b^{-6}\left(1 + \chi_{\mathbf{3}}^{\text{SU(2)}_v}\chi_{\mathbf{3}}^{\text{SU(2)}_f}\right)(pq)^{\frac{2}{7}} + b^{-3}t^2\chi_{\mathbf{2}}^{\text{SU(2)}_v}\chi_{\mathbf{2}}^{\text{SU(2)}_f}(pq)^{\frac{11}{28}}$$

$$+ b^{-6}t\left(1 + \chi_{\mathbf{3}}^{\text{SU(2)}_v}\right)\left(1 + \chi_{\mathbf{3}}^{\text{SU(2)}_f}\right)(pq)^{\frac{23}{56}}$$

$$+ \left(b^{-9}\left(\chi_{\mathbf{2}}^{\text{SU(2)}_v}\chi_{\mathbf{2}}^{\text{SU(2)}_f} + \chi_{\mathbf{4}}^{\text{SU(2)}_v}\chi_{\mathbf{4}}^{\text{SU(2)}_f}\right) + b^{-2}\chi_{\mathbf{27}}^{\text{E}_6}\right)(pq)^{\frac{3}{7}} + b^6 t^{-2}(pq)^{\frac{13}{28}} + t^4(pq)^{\frac{1}{2}}$$

$$+ \cdots + t^4(pq)^{\frac{1}{2}}(p+q) + \left(-\chi_{\mathbf{78}}^{\text{E}_6} - \chi_{\mathbf{3}}^{\text{SU(2)}_v} - \chi_{\mathbf{3}}^{\text{SU(2)}_f} - 2 + 1 + t^8 + \chi_{\mathbf{3}}^{\text{SU(2)}_v}\chi_{\mathbf{3}}^{\text{SU(2)}_f}\right.$$

$$+ b^{-21}\left(\chi_{\mathbf{2}}^{\text{SU(2)}_v}\chi_{\mathbf{2}}^{\text{SU(2)}_f} + \chi_{\mathbf{4}}^{\text{SU(2)}_v}\chi_{\mathbf{4}}^{\text{SU(2)}_f} + \chi_{\mathbf{6}}^{\text{SU(2)}_v}\chi_{\mathbf{6}}^{\text{SU(2)}_f} + \chi_{\mathbf{8}}^{\text{SU(2)}_v}\chi_{\mathbf{8}}^{\text{SU(2)}_f}\right)$$

$$\left. + b^{-14}\left(1 + \chi_{\mathbf{3}}^{\text{SU(2)}_v}\chi_{\mathbf{3}}^{\text{SU(2)}_f} + \chi_{\mathbf{5}}^{\text{SU(2)}_v}\chi_{\mathbf{5}}^{\text{SU(2)}_f}\right)\chi_{\mathbf{27}}^{\text{E}_6} + b^{-7}\chi_{\mathbf{2}}^{\text{SU(2)}_v}\chi_{\mathbf{2}}^{\text{SU(2)}_f}\left(\chi_{\mathbf{351'}}^{\text{E}_6} + \chi_{\mathbf{27}}^{\text{E}_6}\right)\right)pq + \cdots.$$

$$(116)$$

Again we can see characters $\chi_{\mathbf{m}}^{\text{E}_6}$ for the $\text{E}_6$ symmetry that is enhanced from manifest $\text{SU(6)}_w \times \text{SU(2)}_d$. Moreover, from the negative terms at order $pq$ highlighted in blue we can see the conserved currents for

$$\text{E}_6 \times \text{SU(2)}_v \times \text{SU(2)}_f \times \text{U(1)}_b \times \text{U(1)}_t, \tag{117}$$

as expected. Note that there should be at least two U(1) currents because the theory already exhibits $\text{U(1)}_b$ and $\text{U(1)}_t$.

Also in this case we can check the presence of gauge invariant operators that descend from the 6d conserved currents. On top of those that we found for rank 1 coming from the $\text{E}_8$ conserved currents, we now expect also operators coming from the $\text{SU(2)}_L$ conserved currents, according to the branching rule of its adjoint representation (74). This time the 6d R-symmetry is related to the one we used to compute the index (116) by the shifts $b \to b(pq)^{-\frac{2}{7}}$ and $t \to t(pq)^{\frac{3}{8}}$. Moreover, the U(1) inside $\text{SU(2)}_L$ for which we turned on a flux $-1$ coincides with the $\text{U(1)}_t$ of our 4d model. With this dictionary, we can identify the states appearing in (111) and (74) in the index (116) up to the order we evaluated it:

$$\begin{aligned}
(\mathbf{1},\mathbf{2})^{(-3,0)} &\to & 2b^{-3}\chi_{\mathbf{2}}^{\text{SU(2)}_v}(pq)^{\frac{1}{7}} \\
(\overline{\mathbf{27}},\mathbf{1})^{-2} &\to & b^{-2}\chi_{\mathbf{27}}^{\text{E}_6}(pq)^{\frac{3}{7}} \\
(\mathbf{1},\mathbf{1})^{(0,2)} &\to & t^2(pq)^{\frac{1}{4}} \\
2 \times (\mathbf{1},\mathbf{1})^{(0,0)} \oplus (\mathbf{1},\mathbf{3})^{(0,0)} \oplus (\mathbf{78},\mathbf{1})^{(0,0)} &\to & -\left(\chi_{\mathbf{78}}^{\text{E}_6} + \chi_{\mathbf{3}}^{\text{SU(2)}_v} + 2\right)pq,
\end{aligned} \tag{118}$$

where at the exponent of the states on the left hand side we reported, in order, the charges under $\text{U(1)}_b$ and $\text{U(1)}_t$. Notice again that the state $(\mathbf{1},\mathbf{2})^{(-3,1)}$ contributes with two operators, which transform as a doublet under the geometric $\text{SU(2)}_f$.

We have found that the decoupled operators for $Q = 2$ are given by

$$(R^{(1)\pm}, \tilde{R}^{(1)\pm}), \qquad \text{Tr}A^2, \qquad (R^{(2)\pm}, \tilde{R}^{(2)\pm}), \tag{119}$$

which correspond to the first three nontrivial terms of the index (116) respectively. Note that $(R^{(n)\pm}, \tilde{R}^{(n)\pm})$ for $n = 1, 2$ are in $(\mathbf{2},\mathbf{2})$ of $\text{SU(2)}_v \times \text{SU(2)}_f$. The remaining interacting sector is dual to the $E_6$ model in [22], which was shown to have the enhanced IR symmetry

$$\text{E}_6 \times \text{U(1)}_b \times \text{U(1)}_t. \tag{120}$$

### 3.3  $SO(14) \times U(1)^2$ **sphere model I**

In this subsection, we consider the model corresponding to the flux

$$\mathcal{F} = (-1; 0, 0, 0, 0, 2, 2, 2, -2), \tag{121}$$

which preserves $SO(14) \times U(1)_b \times U(1)_t \subset E_8 \times SU(2)_t$. This model can be obtained by gluing two basic caps with the $S$-gluing for the first four octet moment maps and the $\Phi$-gluing for the other octet moment maps and the antisymmetric moment maps. From the flux (121) we expect that the theory has the global symmetry

$$SO(14) \times U(1)_b \times U(1)_t. \tag{122}$$

Again we start from the basic cap, the WZ model with the superpotential

$$\mathcal{W}_{cap} = \sum_{n=1}^{Q} \sum_{a=1}^{7} \text{Tr}_z \left( R^{(n)a} A^{Q-n} P L_a \right) + \sum_{n=1}^{Q} \text{Tr}_z \left( b_n A^{Q-n} P K \right) + K L_8, \tag{123}$$

which preserves

$$USp(2Q) \times SU(7)_u \times U(1)_x \times U(1)_c \times U(1)_t \times U(1)_f. \tag{124}$$

Since we want to take the $S$-gluing for $L_{1,\ldots,4}$ and the $\Phi$-gluing for $L_{5,\ldots,8}$ and $A$, we introduce $\hat{A}$, $\Phi^{5,\ldots,8}$ and the superpotential

$$\mathcal{W}_{glue} = \text{Tr}_z \left[ \hat{A} \cdot (A - \tilde{A}) \right] + \sum_{a=1}^{4} L_a \tilde{L}^a + \sum_{b=5}^{8} \Phi^b \left( L_b - \tilde{L}_b \right) \tag{125}$$

and gauge the puncture symmetry $USp(2Q)$. The total superpotential of the glued theory is given by

$$
\begin{aligned}
\mathcal{W} &= \mathcal{W}_{cap} + \tilde{\mathcal{W}}_{cap} + \mathcal{W}_{glue} \\
&= \sum_{n=1}^{Q} \sum_{a=1}^{7} \text{Tr}_z \left( R^{(n)a} A^{Q-n} P L_a \right) + \sum_{n=1}^{Q} \text{Tr}_z \left( b_n A^{Q-n} P K \right) + K L_8 \\
&\quad + \sum_{n=1}^{Q} \sum_{a=1}^{4} \text{Tr}_z \left( \tilde{R}^{(n)}{}_a \tilde{A}^{Q-n} \tilde{P} \tilde{L}^a \right) + \sum_{n=1}^{Q} \sum_{a=5}^{7} \text{Tr}_z \left( \tilde{R}^{(n)a} \tilde{A}^{Q-n} \tilde{P} \tilde{L}_a \right) + \sum_{n=1}^{Q} \text{Tr}_z \left( \tilde{b}_n \tilde{A}^{Q-n} \tilde{P} \tilde{K} \right) \\
&\quad + \tilde{K} \tilde{L}_8 + \text{Tr}_z \left[ \hat{A} \cdot (A - \tilde{A}) \right] + \sum_{a=1}^{4} L_a \tilde{L}^a + \sum_{b=5}^{8} \Phi^b \left( L_b - \tilde{L}_b \right),
\end{aligned}
\tag{126}
$$

which becomes

$$\mathcal{W} = \sum_{n=1}^{Q} \sum_{b=5}^{8} \sum_{\alpha=\pm} \text{Tr}_z \left( S^{(n)b}{}_\alpha A^{Q-n} P^\alpha Q_b \right) - \sum_{m,n=1}^{Q} \sum_{a=1}^{4} \text{Tr}_z \left( A^{2Q-m-n} P^+ P^- R^{(m)a} \tilde{R}^{(n)}{}_a \right), \tag{127}$$

once we integrate out the massive fields. We have defined

$$S^{(n)b}_+ = \begin{cases} R^{(n)b}, & b = 5, \ldots, 7, \\ b_n, & b = 8, \end{cases} \tag{128}$$

$$S^{(n)b}_- = \begin{cases} \tilde{R}^{(n)b}, & b = 5, \ldots, 7, \\ -\tilde{b}_n, & b = 8, \end{cases} \tag{129}$$

$$Q_b = \begin{cases} L_b, & b = 5, \ldots, 7, \\ \frac{K - \tilde{K} - \Phi}{3}, & b = 8, \end{cases} \tag{130}$$

$$P^+ = P, \tag{131}$$

$$P^- = \tilde{P}. \tag{132}$$

Table 4: The matter contents of the $SO(14) \times U(1)^2$ model I and their charges.

|  | USp(2Q) | SU(4)$_v$ | SU(4)$_w$ | U(1)$_d$ | U(1)$_f$ | U(1)$_b$ | U(1)$_t$ | U(1)$_R$ |
|---|---|---|---|---|---|---|---|---|
| $S_\pm^{(n)}$ | **1** | **1** | **4** | $\pm 2$ | $\pm 1$ | $-1$ | $n-1$ | $2$ |
| $R^{(n)}$ | **1** | **4** | **1** | $1$ | $0$ | $-2$ | $n-1$ | $2$ |
| $\tilde{R}^{(n)}$ | **1** | $\overline{\textbf{4}}$ | **1** | $-1$ | $0$ | $-2$ | $n-1$ | $2$ |
| $Q$ | **2Q** | **1** | $\overline{\textbf{4}}$ | $0$ | $0$ | $-1$ | $0$ | $1$ |
| $P^\pm$ | **2Q** | **1** | **1** | $\mp 2$ | $\mp 1$ | $2$ | $1-Q$ | $-1$ |
| $A$ | **antisym** | **1** | **1** | $0$ | $0$ | $0$ | $1$ | $0$ |

Note that the superpotential (127) preserves[16]

$$SU(4)_v \times SU(4)_w \times U(1)_d \times U(1)_f \times U(1)_b \times U(1)_t\,, \tag{135}$$

with the charges of the chiral multiplets given in Table 4.

Interestingly, although the $S^{(n)}$ and $P$ are charged under $U(1)_f$, we will see that it is not a faithful symmetry because no gauge invariant operator is charged under this $U(1)_f$. In addition, we will also see that $SU(4)_v \times SU(4)_w \times U(1)_d$ is enhanced to $SO(14)$. Thus, the global symmetry of the theory is given by

$$SO(14) \times U(1)_b \times U(1)_t\,. \tag{136}$$

We should comment that, as we will see later on, this theory flows to a WZ model [24]. The symmetry (136) is the one preserved by the superpotential of the dual WZ model:

$$\mathcal{W} = \sum_{k+l+m+n=2Q+1} \sum_{a=1}^{14} A^{(k)} P^{(l)} Q_a^{(m)} Q_a^{(n)}\,, \tag{137}$$

where the dual field can be identified with that of the current model as follows:

$$\begin{aligned}
A^{(k)} &\longleftrightarrow \mathrm{Tr} A^k\,, \\
P^{(l)} &\longleftrightarrow \mathrm{Tr}\left[A^{l-1}P^+P^-\right]\,, \\
Q^{(m)} &\longleftrightarrow R^{(n)}, \tilde{R}^{(n)}, \mathrm{Tr}\left[A^{m-1}Q_{[a}Q_{b]}\right]\,.
\end{aligned} \tag{138}$$

Since the theory is a WZ model, the superpotential (137) is irrelevant and all the fields in (138) become free in the IR. Accordingly, neglecting the superpotential the global symmetry becomes much larger. We will nevertheless keep the superpotential (137) because we want to focus on the smaller symmetry (136) so to show explicitly that our model conforms to the expectations from 6d. The true larger symmetry is not in contrast with the 6d prediction and it is accidental from this point of view.

---

[16]While we focus on the global symmetry (135) for simplicity, the full symmetry preserved by the superpotential (127) is

$$SO(8)_s \times SU(4)_w \times SU(2)_g \times U(1)_b \times U(1)_t\,, \tag{133}$$

where $SO(8)_s \times SU(2)_g$ is enhanced from $SU(4)_v \times U(1)_d \times U(1)_f$. The fugacity map between the two groups is given by

$$\begin{aligned}
s_a &= dv_a\,, \\
g &= d^2 f\,.
\end{aligned} \tag{134}$$

Before evaluating the index, we should note that the gluing breaks $SU(7)_u \times U(1)_x$ of the basic cap into $SU(4)_v \times SU(3)_u \times U(1)_y \times U(1)_x$. Thus, we need to redefine the fugacities as follows:

$$
\begin{aligned}
u_{a=1,\dots,4} &\longrightarrow x^{-1}y^{-3}v_a, \\
u_{b=5,\dots,7} &\longrightarrow x^{-1}y^{4}u_b, \\
u_8 &\longrightarrow x^{7},
\end{aligned}
\tag{139}
$$

where $v_a$ and $u_b$ on the right hand side satisfy $\prod_{a=1}^{4} v_a = 1$ and $\prod_{b=5}^{7} u_b = 1$ respectively. The octet fugacities of the caps are then given by

$$
a_i = \left( c^{\frac{1}{2}} x^{-1} y^{-3} v_a; c^{\frac{1}{2}} x^{-1} y^{4} u_b; c^{\frac{1}{2}} x^{7} \right),
\tag{140}
$$

for the left cap and

$$
b_i = \left( c'^{\frac{1}{2}} x'^{-1} y'^{-3} v_a; c'^{\frac{1}{2}} x'^{-1} y'^{4} u'_b; c'^{\frac{1}{2}} x'^{7} \right),
\tag{141}
$$

for the right cap. Since we take the $S$-gluing for the first four and the $\Phi$-gluing for the others, we impose the conditions $a_i = b_i^{-1}$ for $i = 1, \dots, 4$ and $a_i = b_i$ otherwise, which are solved by

$$
\begin{aligned}
c' &= x^2 y^6, \\
x' &= c^{\frac{1}{14}} x^{\frac{6}{7}} y^{-\frac{3}{7}}, \\
y' &= c^{\frac{1}{7}} x^{-\frac{2}{7}} y^{\frac{1}{7}}, \\
v'_a &= v_a^{-1}, \\
u'_b &= u_b.
\end{aligned}
\tag{142}
$$

With this identification, we obtain the index of the theory with the flux (121), which is given by

$$
\begin{aligned}
\mathcal{I}_{(-1;0^4,2^3,-2)} &= \\
&= \oint dz_Q\, \Gamma_e\left(pqt^{-1}\right)^{Q-1} \prod_{n<m}^{Q} \Gamma_e\left(pqt^{-1} z_n^{\pm 1} z_m^{\pm 1}\right) \prod_{n=1}^{Q} \prod_{a=5}^{8} \Gamma_e\left((pq)^{\frac{1}{2}} c^{\frac{1}{2}} u_a z_n^{\pm 1}\right) \\
&\quad \times \mathcal{I}_{cap}(z_n; c, t, f; u_1, \dots, u_7; x) \\
&\quad \times \left( \mathcal{I}_{cap}(z_n; c, t, f^{-1}; u_1, \dots, u_7; x)\Big|_{c \to x^2 y^6,\, x \to c^{\frac{1}{14}} x^{\frac{6}{7}} y^{-\frac{3}{7}},\, y \to c^{-\frac{1}{7}} x^{\frac{2}{7}} y^{-\frac{1}{7}},\, v_a \to v_a^{-1}} \right),
\end{aligned}
\tag{143}
$$

where (139) is understood for $u_a$. Furthermore, if we introduce the following redefinition of the fugacities:

$$
\begin{aligned}
w_{a=1,2,3} &= c^{\frac{1}{4}} x^{\frac{3}{2}} y u_a, \\
w_4 &= c^{-\frac{3}{4}} x^{-\frac{9}{2}} y^{-3}, \\
b &= c^{\frac{1}{4}} x^{-\frac{5}{2}} y^3, \\
d &= c^{-\frac{1}{2}} x y^3 f^{-1},
\end{aligned}
\tag{144}
$$

the index is written with the manifest $SU(4)_v \times SU(4)_w \times U(1)_d \times U(1)_f \times U(1)_b \times U(1)_t$ symmetry

as follows:

$$\mathcal{I}_{(-1;0^4,2^3,-2)} =$$

$$= \prod_{n=1}^{Q}\prod_{a=1}^{4}\underbrace{\Gamma_e\left(pqt^{n-1}b^{-2}(dv_a)^{\pm 1}\right)}_{R^{(n)},\tilde{R}^{(n)}}\prod_{n=1}^{Q}\prod_{a=1}^{4}\underbrace{\Gamma_e\left(pqt^{n-1}b^{-1}(d^2f)^{\pm 1}w_a\right)}_{S_{\pm}^{(n)a}}$$

$$\times \oint dz_Q\ \underbrace{\Gamma_e(t)^{Q-1}\prod_{n<m}\Gamma_e\left(tz_n^{\pm 1}z_m^{\pm 1}\right)}_{A}\prod_{n=1}^{Q}\prod_{a=1}^{4}\underbrace{\Gamma_e\left((pq)^{\frac{1}{2}}b^{-1}w_a^{-1}z_n^{\pm 1}\right)}_{Q_a}$$

$$\times \prod_{n=1}^{Q}\underbrace{\Gamma_e\left((pq)^{-\frac{1}{2}}t^{1-Q}b^2(d^2f)^{\pm 1}z_n^{\pm 1}\right)}_{P^{\mp}}. \tag{145}$$

Although the integrand depends on $f$, by explicitly expanding the index, we are able to see that $U(1)_f$ is not a faithful symmetry because the gauge invariant operators are all invariant under $U(1)_f$. The faithful symmetry of the theory is thus

$$SO(14) \times U(1)_b \times U(1)_t. \tag{146}$$

The fact that the $U(1)_f$ symmetry is not faithful manifests itself also in the vanishing of all the anomalies involving it. Moreover, we again want to interpret $U(1)_f$ as the symmetry descending from the isometry of the two-sphere. The two statements are indeed compatible, since we find that all of the anomalies for this symmetry computed from 6d (B.21) accidentally vanish for the value of the flux (121). As before, the mixing of the original $U(1)_f$ from (4) with the other U(1)'s that gives the correct anomalies matches the predicted mixing given in (B.22).

We discuss the index for the low values of $Q$ in some detail.

**Rank 1**

To see the enhanced global symmetry, let us expand the the index (145) for some low values of rank $Q$. Firstly, for $Q = 1$, we take the mixing coefficient of $U(1)_b$ with the R-symmetry as

$$R_b = \frac{2}{3}, \tag{147}$$

which is the value determined by the $a$-maximization. Note that this is the exact value rather than the approximate one. With this mixing coefficient, the expansion of the index is given by

$$\mathcal{I}_{(-1;0^4,2^3,-2)}^{Q=1} = 1 + \left(b^4 + b^{-2}\chi_{14}\right)(pq)^{\frac{1}{3}} + \left(b^8 + b^{-4}\chi_{104}\right)(pq)^{\frac{2}{3}} + \left(b^4 + b^{-2}\chi_{14}\right)(pq)^{\frac{1}{3}}(p+q)$$
$$+ \left(-\chi_{91} - 1 + b^{12} + b^{-6}\chi_{546}\right)pq + \dots, \tag{148}$$

where $\chi_{\mathbf{m}}$ is the $\mathbf{m}$-dimensional character of $SO(14)$ written in terms of $v_a$, $w_a$ and $d$. For example, the $\mathbf{14}$ of $SO(14)$ is decomposed into

$$\mathbf{14} \longrightarrow (\mathbf{4},\mathbf{1})^1 \oplus (\overline{\mathbf{4}},\mathbf{1})^{-1} \oplus (\mathbf{1},\mathbf{6})^0 \tag{149}$$

under $SU(4)_v \times SU(4)_w \times U(1)_d$ and the character is written accordingly. Furthermore, we see that the contribution of the current multiplet, which is highlighted in blue, is in the adjoint representation of

$$SO(14) \times U(1)_b, \tag{150}$$

as expected.

Also in this case we can check the presence of gauge invariant operators that descend from the 6d conserved currents. For rank 1 we focus on the branching rule of the adjoint representation of $E_8$ under the $SO(14) \times U(1)$ subgroup:

$$\mathbf{248} \to \mathbf{1}^0 \oplus \mathbf{91}^0 \oplus \mathbf{64}^1 \oplus \overline{\mathbf{64}}^{-1} \oplus \mathbf{14}^{\pm 2}. \tag{151}$$

The 6d R-symmetry is related to the one we used to compute the index (148) by the shift $b \to b(pq)^{-\frac{1}{3}}$. Moreover, the $U(1)$ inside $E_8$ for which we turned on a unit of flux is related to $U(1)_b$. With this dictionary, we can identify the states appearing in (151) in the index (148) up to the order we evaluated it:

$$
\begin{aligned}
\mathbf{14}^{-2} &\to b^{-2}\chi_{14}(pq)^{\frac{1}{3}} \\
\mathbf{1}^0 \oplus \mathbf{91}^0 &\to -(\chi_{91}+1)pq.
\end{aligned}
\tag{152}
$$

We notice that in this case there is no state contributing with more than one operator and which can form representations of $U(1)_f$. This is compatible with our finding that $U(1)_f$ is not a symmetry of the IR theory.

Note that in this model the chiral operators given by the first non trivial term of the index (148) hit the unitarity bound. In fact these operators become free fields in the IR and are the full content of the theory in the IR, as we remarked below eq. (138). This model flows indeed to a WZ model with the superpotential

$$\mathcal{W}^{Q=1} = \sum_{a=1}^{14} P^{(1)} Q_a^{(1)} Q_a^{(1)}, \tag{153}$$

which preserves the symmetry (150).

**Rank 2**

Next we compute the index (145) for $Q = 2$. For $Q = 2$, we take the mixing coefficients as

$$R_t = \frac{1}{7}, \qquad R_b = \frac{7}{10}, \tag{154}$$

which approximate the irrational values determined by the $a$-maximization:

$$R_t \approx 0.1442, \qquad R_b \approx 0.7045. \tag{155}$$

With those mixing coefficients, the index is given by

$$
\begin{aligned}
\mathcal{I}^{Q=2}_{(-1;0^4,2^3,-2)} = {}& 1 + t^2(pq)^{\frac{1}{7}} + b^4 t^{-2}(pq)^{\frac{9}{35}} + t^4(pq)^{\frac{2}{7}} + b^{-2}\chi_{14}(pq)^{\frac{3}{10}} + b^4 t^{-1}(pq)^{\frac{23}{70}} \\
& + b^{-2}t\chi_{14}(pq)^{\frac{13}{35}} + b^4(pq)^{\frac{2}{5}} + t^6(pq)^{\frac{3}{7}} + b^{-2}t^2\chi_{14}(pq)^{\frac{31}{70}} + b^4 t(pq)^{\frac{33}{70}} \\
& + \left(b^8 t^{-4} + b^{-2}t^3\chi_{14}\right)(pq)^{\frac{18}{35}} + b^4 t^2(pq)^{\frac{19}{35}} + b^2 t^{-2}\chi_{14}(pq)^{\frac{39}{70}} + t^8(pq)^{\frac{4}{7}} \\
& + \left(b^8 t^{-3} + b^{-2}t^4\chi_{14}\right)(pq)^{\frac{41}{70}} + b^{-4}(1+\chi_{104})(pq)^{\frac{3}{5}} + b^4 t^3(pq)^{\frac{43}{70}} \\
& + b^2 t^{-1}\chi_{14}(pq)^{\frac{22}{35}} + t^2(pq)^{\frac{1}{7}}(p+q) + \cdots + \left(-\chi_{91}-2+t^{14}+\chi_{104}\right)pq + \dots,
\end{aligned}
\tag{156}
$$

where we see the contribution of the conserved current for

$$SO(14) \times U(1)_b \times U(1)_t. \tag{157}$$

Also in this case we can check the presence of gauge invariant operators that descend from the 6d conserved currents. On top of those that we found for rank 1 coming from the $E_8$ conserved currents, we now expect also operators coming from the $SU(2)_L$ conserved currents, according to the branching rule of its adjoint representation (74). This time the 6d R-symmetry is related to the one we used to compute the index (156) by the shifts $b \to b(pq)^{-\frac{7}{20}}$ and $t \to t(pq)^{\frac{3}{7}}$. Moreover, the $U(1)$ inside $SU(2)_L$ for which we turned on a flux $-1$ coincides with the $U(1)_t$ of our 4d model. With this dictionary, we can identify the states appearing in (151) and (74) in the index (156) up to the order we evaluated it:

$$
\begin{aligned}
\mathbf{14}^{(-2,0)} \quad &\to \quad b^{-2}\chi_{\mathbf{14}}(pq)^{\frac{3}{10}} \\
\mathbf{1}^{(0,2)} \quad &\to \quad t^2(pq)^{\frac{1}{7}} \\
2 \times \mathbf{1}^{(0,0)} \oplus \mathbf{91}^{(0,0)} \quad &\to \quad -\left(\chi_{\mathbf{91}} + 2\right)pq,
\end{aligned}
\tag{158}
$$

where at the exponent of the states on the left hand side we reported, in order, the charges under $U(1)_b$ and $U(1)_t$. Again there is no state contributing with more than one operator and which can form representations of $U(1)_f$. This is compatible with our finding that $U(1)_f$ is not a symmetry of the IR theory.

Also for $Q = 2$ the theory flows to a free theory. One can see that the first nontrivial term of the index (156), which corresponds to $\mathrm{Tr}A^2$, is below the unitarity bound. Once we flip this operator, the resulting theory only includes chiral operators hitting the unitarity bound $R = \frac{2}{3}$. Therefore, this models flows to a free theory in the IR where all the chiral operators become free fields. The symmetry (157) captured by the index should be understood, like the $Q = 1$ case, as the symmetry preserved during the flow to the free model.

## 3.4  $SO(14) \times U(1)^2$ sphere model II and Csaki–Skiba–Schmaltz duality

Lastly, we analyse an example corresponding to the flux

$$
\mathcal{F} = (-1; 1, 1, 1, 1, 1, 1, 3, -1),
\tag{159}
$$

which also preserves $SO(14) \times U(1)_b \times U(1)_t \subset E_8 \times SU(2)_t$. Most importantly, the flux (159) is equivalent to (121) up to Weyl reflections of $E_8$. Therefore, from the 6d perspective, they stem from the same 6d theory compactified on a sphere with the same flux, which thus leads to a duality between the resulting 4d theories. Indeed, we will see that the theory with the flux (159) is the WZ model dual to the $USp(2Q)$ theory in the previous subsection.

This model can be simply obtained from the cap theory by closing the remaining puncture. First recall that the index of the cap is given by

$$
\begin{aligned}
&\mathcal{I}_{cap}(y_n; c, t; u_1, \dots, u_7; x) = \\
&= \Gamma_e(t)^{Q-1} \prod_{n<m}^{Q} \Gamma_e\left(t y_n^{\pm 1} y_m^{\pm 1}\right) \prod_{n=1}^{Q} \Gamma_e\left(pq t^{n-1} c^{-2} f^{-1}\right) \prod_{n=1}^{Q} \Gamma_e\left((pq)^{-\frac{1}{2}} t^{1-Q} c^{\frac{3}{2}} f u_8^{-1} y_n^{\pm 1}\right) \\
&\times \prod_{n=1}^{Q}\prod_{a=1}^{7} \Gamma_e\left(pq t^{n-1} c^{-1} f^{-1} u_8 u_a\right) \prod_{n=1}^{Q}\prod_{a=1}^{7} \Gamma_e\left((pq)^{\frac{1}{2}} c^{-\frac{1}{2}} u_a^{-1} y_n^{\pm 1}\right).
\end{aligned}
\tag{160}
$$

One can repeat the procedure in Section 2 now specialising $y_n = w t^{-\frac{Q-2n+1}{2}}$ and $w = (pq)^{\frac{1}{2}} c^{-\frac{1}{2}} t^{\frac{Q-1}{2}} u_7^{-1}$ with extra singlets provided to ensure the anomaly matching. The

resulting sphere model has the index:

$$\mathcal{I}_{(-1;1^6,3,-1)} = \prod_{n=2}^{Q} \Gamma_e(t^n) \prod_{n=1}^{Q} \Gamma_e\left(pqt^{n-1}c^{-2}f^{-1}\right) \prod_{n=1}^{Q} \Gamma_e\left(pqt^{n-1}y^{-2}\right)$$

$$\times \prod_{n=1}^{Q} \Gamma_e\left((pq)^{-1}t^{-2Q+1+n}c^2fy^2\right) \prod_{n=1}^{Q}\prod_{a=1}^{6} \Gamma_e\left(pqt^{n-1}c^{-1}f^{-1}y^{-1}xu_a\right)$$

$$\times \prod_{n=1}^{Q}\prod_{a=1}^{6} \Gamma_e\left(pqt^{n-1}c^{-1}y^{-1}x^{-1}u_a^{-1}\right), \tag{161}$$

where we have made the following redefinition of the fugacities:

$$\begin{aligned} u_{a=1,\dots,6} &\longrightarrow x^{-\frac{1}{2}}u_a\,, \\ u_7 &\longrightarrow x^{\frac{3}{2}}y\,, \\ u_8 &\longrightarrow x^{\frac{3}{2}}y^{-1}\,, \end{aligned} \tag{162}$$

with new $u_a$ on the right hand side satisfying $\prod_{a=1}^{6} u_a = 1$. This choice of the fugacities makes only $SU(6)_u \times U(1)_x \times U(1)_y \subset SU(8)$ manifest. However, as we mentioned, the full symmetry of the theory is supposed to be $SO(14) \times U(1)_b \times U(1)_t$. Indeed, one can introduce the $SO(14) \times U(1)_b$ fugacities defined by

$$\begin{aligned} v_{a=1,\dots,6} &= xu_a f^{-\frac{1}{2}}\,, \\ v_7 &= cy^{-1}f^{\frac{1}{2}}\,, \\ b &= c^{\frac{1}{2}}y^{\frac{1}{2}}f^{\frac{1}{4}}\,, \end{aligned} \tag{163}$$

so that the index is written in the $SO(14) \times U(1)_b \times U(1)_t$ symmetrc way. The mixing of $U(1)_f$ with the other $U(1)$'s reflected in (163) exactly matches the predicted mixing given in (B.22). After mixing the dependence of $f$ completely disappears, which is consistent with the fact that we expect this $U(1)_f$ descends from the isometry of the two-sphere, whose anomalies computed from 6d (B.21) vanish for the value of the flux (159). With those fugacities in (163), the index is now written as

$$\mathcal{I}_{(-1;1^6,3,-1)} = \prod_{n=2}^{Q} \underbrace{\Gamma_e(t^n)}_{A^{(n)}} \prod_{n=1}^{Q} \underbrace{\Gamma_e\left((pq)^{-1}t^{-2Q+1+n}b^4\right)}_{P^{(n)}} \prod_{n=1}^{Q}\prod_{a=1}^{7} \underbrace{\Gamma_e\left(pqt^{n-1}b^{-2}v_a^{\pm1}\right)}_{Q^{(n)}}. \tag{164}$$

From the index, one can read that the corresponding 4d theory is a Wess–Zumino model with the superpotential

$$\mathcal{W} = \sum_{k+l+m+n=2Q+1} \sum_{a=1}^{14} A^{(k)}P^{(l)}Q_a^{(m)}Q_a^{(n)}\,, \tag{165}$$

with $k = 2,\dots,Q$ and $l,m,n = 1,\dots,Q$. The superpotential (165) preserves

$$SO(14)_v \times U(1)_b \times U(1)_t\,, \tag{166}$$

as expected from the 6d perspective. The symmetry charges of each chiral multiplet are presented in Table 5. Note as this is a WZ model it flows to a collection of free chiral fields with the above symmetry being a subgroup of the symmetry of the free theory in the IR.

Table 5: The matter contents of the $SO(14) \times U(1)^2$ model II and their charges.

|  | SO(14) | $U(1)_b$ | $U(1)_t$ | $U(1)_R$ |
|---|---|---|---|---|
| $Q^{(n)}$ | **14** | $-2$ | $n-1$ | $2$ |
| $P^{(n)}$ | **1** | $4$ | $-2Q+1+n$ | $-2$ |
| $A^{(n)}$ | **1** | $0$ | $n$ | $0$ |

One can also explicitly expand the index (164). With the R-symmetry mixing coefficient of $U(1)_b$

$$R_b = \frac{2}{3}, \tag{167}$$

the index (164) for $Q = 1$ is evaluated as

$$\mathcal{I}^{Q=1}_{(-1;1^6,3,-1)} = 1 + \left(b^4 + b^{-2}\chi_{\mathbf{14}}\right)(pq)^{\frac{1}{3}} + \left(b^8 + b^{-4}\chi_{\mathbf{104}}\right)(pq)^{\frac{2}{3}} + \left(b^4 + b^{-2}\chi_{\mathbf{14}}\right)(pq)^{\frac{1}{3}}(p+q)$$
$$+ \left(-\chi_{\mathbf{91}} - 1 + b^{12} + b^{-6}\chi_{\mathbf{546}}\right)pq + \dots, \tag{168}$$

which is exactly the same as the index (148) of the $SO(14) \times U(1)_b$ model in the previous subsection. Indeed, those two models for general $Q$ have been described as dual theories [24], whose index agreement was also shown in [42]. Our approach then provides a novel way to understand this duality from the 6d perspective. We have shown that both theories are obtained by compactifying the same E-string theory on a sphere with the same flux because the flux (159) is equivalent to (121) up to Weyl reflections of the $E_8$ symmetry of the 6d theory. This leads to a duality between the resulting 4d theories, both of which should exhibit the same global symmetry $SO(14) \times U(1)_b \times U(1)_t$.

Also in this case, as for the dual model corresponding to the flux (121), all of the anomalies for the $U(1)_f$ symmetry trivially vanish because all the chiral multiplets are neutral under $U(1)_f$. This is again compatible with the interpretation of this $U(1)_f$ as the symmetry descending from the isometry of the two-sphere, since we find that all of the anomalies for this symmetry computed from 6d (B.21) accidentally vanish also for the value of the flux (159).

# Acknowledgements

We would like to thank Sara Pasquetti for collaboration at the early stages of this project. We are also grateful to Ibrahima Bah, Hee-Cheol Kim, Sara Pasquetti and Gabi Zafrir for interesting discussions. CH is partially supported by STFC consolidated grants ST/P000681/1, ST/T000694/1. This research of SSR and ES is supported in part by Israel Science Foundation under grant no. 2289/18, by I-CORE Program of the Planning and Budgeting Committee, by a Grant No. I-1515-303./2019 from the GIF, the German-Israeli Foundation for Scientific Research and Development, and by BSF grant no. 2018204. The research of SSR is also supported by the IBM Einstein fellowship of the Institute of Advnaced Study. MS is partially supported by the ERC-STG grant 637844-HBQFTNCER, by the University of Milano-Bicocca grant 2016-ATESP0586, by the MIUR-PRIN contract 2017CC72MK003, and by the INFN.

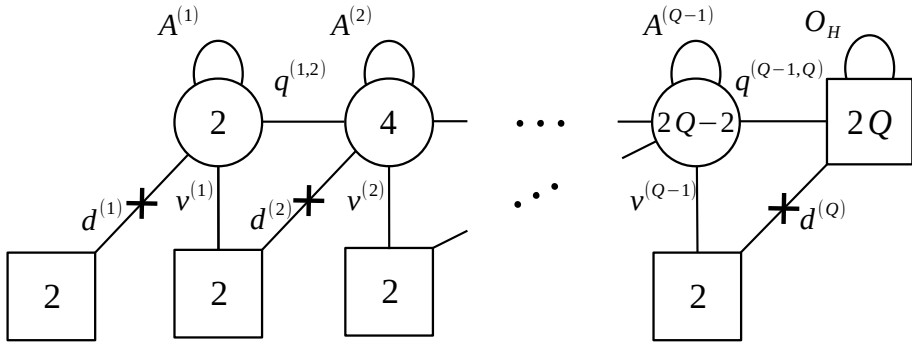

Figure 4: The quiver description for the $FE[\mathrm{USp}(2Q)]$ theory. Each node with a label $2n$ inside denotes a $\mathrm{USp}(2n)$ group, with circle nodes being gauge groups and square nodes being flavor groups. Lines connecting adjacent nodes represent chiral fields transforming under the associated symmetries. In particular, we have an antisymmetric (with the trace part) chiral field $A^{(n)}$ at each gauge node, bifundamental chiral fields $q^{(n,n+1)}$ and fundamental chiral fields $d^{(n)}$, $v^{(n)}$. We also have additional gauge singlet chiral fields $b_n$ represented by crosses in the above quiver and a set of singlets collected in a matrix $\mathsf{O_H}$ transforming in the traceless antisymmetric representation of the $\mathrm{USp}(2Q)_x$ flavor symmetry associated to the node at the end of the tail.

## A  Review of the $FE[\mathrm{USp}(2Q)]$ theory

In this appendix we review some of the properties of the $FE[\mathrm{USp}(2Q)]$ theory[17] introduced in [18] and further studied in [19][18]. For more details we refer the reader to those references. The $FE[\mathrm{USp}(2Q)]$ theory is a 4d $\mathcal{N} = 1$ SCFT that admits the quiver description shown in Figure 4. The superpotential of the theory is[19]

$$
\begin{aligned}
W_{FE[\mathrm{USp}(2Q)]} \;\;=\;\; & \sum_{n=1}^{Q-1} \mathrm{Tr}_n \left[ A^{(n)} \left( \mathrm{Tr}_{n+1}\, q^{(n,n+1)} q^{(n,n+1)} - \mathrm{Tr}_{n-1}\, q^{(n-1,n)} q^{(n-1,n)} \right) \right] + \\
& -\;\; \mathrm{Tr}_x \left( \mathsf{O_H}\, \mathrm{Tr}_{Q-1}\, q^{(Q-1,Q)} q^{(Q-1,Q)} \right) + \sum_{n=1}^{Q-1} \mathrm{Tr}_n \left[ v^{(n)}\, \mathrm{Tr}_{n+1} \left( q^{(n,n+1)} d^{(n+1)} \right) \right] + \\
& +\;\; \sum_{n=1}^{Q} b_n\, \mathrm{Tr}_n \left( d^{(n)} d^{(n)} \right),
\end{aligned}
\tag{A.1}
$$

where $\mathrm{Tr}_n$ denotes the trace over the $n$-th $\mathrm{USp}(2n)$ gauge group, $\mathrm{Tr}_x$ denotes the trace over the $\mathrm{USp}(2Q)_x$ flavor symmetry associated to the last node of the quiver and we are setting $q^{(0,1)} = 0$.

---

[17]In this paper we follow the nomenclature of [19] and call this theory $FE[\mathrm{USp}(2Q)]$ , where the "$F$" stands for the presence of the gauge singlet $\mathsf{O_H}$ that flips the antisymmetric operator $\mathsf{H}$.

[18]See also [43] for a different construction than the one discussed in this paper of a theory involving $FE[\mathrm{USp}(2Q)]$ as a building block and enjoying IR global symmetry enhancement.

[19]In the appendix of [22] it was shown that for $Q = 2$ this superpotential is *unstable* in the sense of the chiral ring stability criterion of [44]. This simply means that turning on the relevant deformations in the superpotential sequentially and analyzing the dynamics after each deformation, some of the deformations become irrelevant. Consequently, the symmetry broken by these irrelevant deformations is restored in the IR. The stabilized theory enjoys a larger IR global symmetry enhancement to $\mathrm{SO}(10) \times \mathrm{U}(1)_c \times \mathrm{U}(1)_t$ than the one of $FE[\mathrm{USp}(2Q)]$ for generic $Q$ that we are going to review.

The non-anomalous global symmetry that is manifest in the Lagrangian description is

$$\mathrm{USp}(2Q)_x \times \mathrm{SU}(2)^Q \times \mathrm{U}(1)_c \times \mathrm{U}(1)_t . \tag{A.2}$$

This symmetry is actually enhanced in the IR to

$$\mathrm{USp}(2Q)_x \times \mathrm{USp}(2Q)_y \times \mathrm{U}(1)_c \times \mathrm{U}(1)_t . \tag{A.3}$$

This enhancement was argued in [18] by showing that the gauge invariant operators reorganize into representations of the enhanced $\mathrm{USp}(2Q)_y$ symmetry (*e.g.* using the supersymmetric index) and by means of a duality where in the dual frame $\mathrm{USp}(2Q)_y$ is manifest while $\mathrm{USp}(2Q)_x$ is enhanced at low energies.

The charges of all the chiral fields under the two independent non-anomalous abelian symmetries $\mathrm{U}(1)_c$ and $\mathrm{U}(1)_t$, as well as a possible choice of UV trial R-symmetry, are summarized in Figure 5.

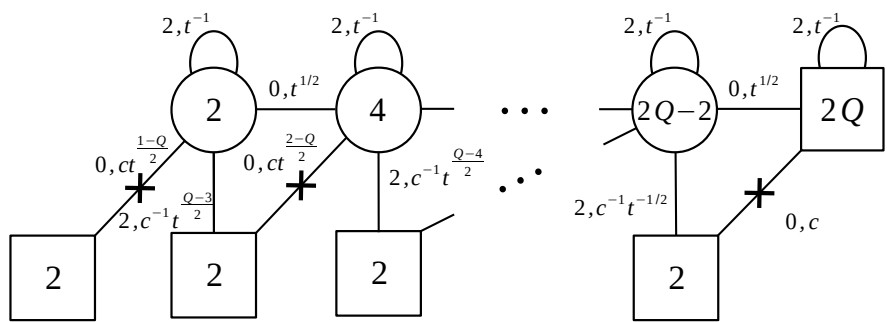

Figure 5: The $U(1)$ charges of the chiral fields of the $FE[\mathrm{USp}(2Q)]$ theory. Each number before the comma denotes the trial R-charges, while the exponents of $c$ and $t$ represent the charges under $\mathrm{U}(1)_c$ and $\mathrm{U}(1)_t$ respectively.

Among the gauge invariant operators of $FE[\mathrm{USp}(2Q)]$, which have been studied in detail in [18, 19], some that play an important role in the constructions of the E-string models discussed in the main text are the following:

- The matrix of singlets $\mathsf{O_H}$ which flips the meson matrix $\mathsf{H} = \mathrm{Tr}_Q\, q^{(Q-1,Q)} q^{(Q-1,Q)}$ and which transforms in the traceless antisymmetric representation of $\mathrm{USp}(2Q)_x$.

- The matrix $\mathsf{C}$ transforming in the traceless antisymmetric representation of the enhanced $\mathrm{USp}(2Q)_y$. Since in the Lagrangian description only the subgroup $\mathrm{SU}(2)^Q \subset \mathrm{USp}(2Q)_y$ is manifest, $\mathsf{C}$ should be constructed by using the branching rule of the antisymmetric representation of $\mathrm{USp}(2Q)_y$ under the $\mathrm{SU}(2)^Q$ subgroup:

$$\mathbf{Q(2Q-1)-1} \to (Q-1) \times (\mathbf{1}, \cdots, \mathbf{1}) \oplus [(\mathbf{2}, \mathbf{2}, \mathbf{1}, \cdots, \mathbf{1}) \oplus \text{permutations}] . \tag{A.4}$$

The $Q-1$ singlets $(\mathbf{1}, \cdots, \mathbf{1})$ correspond to the traces $\mathrm{Tr}_n A^{(n)}$ of the antisymmetric chiral fields at each gauge node, with $n = 1, \cdots, Q-1$. Instead, the bifundamentals $[(\mathbf{2}, \mathbf{2}, \mathbf{1}, \cdots, \mathbf{1}) \oplus \text{permutations}]$ correspond to the operators of the form

$$d^{(n)} \left( \prod_{k=i}^{m-1} q^{(k,k+1)} \right) v^{(m)} , \tag{A.5}$$

with $n = 1, \cdots, Q-1$ and $m = n+1, \cdots, Q-1$.

- The operator $\Pi$ in the bifundamental representation of $USp(2Q)_x \times USp(2Q)_y$, which is constructed collecting operators charged under the manifest $SU(2)^Q \subset USp(2Q)_y$ according to the branching rule

$$\mathbf{2Q} \rightarrow (\mathbf{2}, \mathbf{1}, \cdots, \mathbf{1}) \oplus (\mathbf{1}, \mathbf{2}, \mathbf{1}, \cdots, \mathbf{1}) \oplus \cdots \oplus (\mathbf{1}, \cdots, \mathbf{1}, \mathbf{2}). \tag{A.6}$$

These $Q$ $USp(2Q)_x \times SU(2)$ bifundamentals are operators of the form $d^{(n)} \prod_{k=n}^{Q-1} q^{(k,k+1)}$ with $n = 1, \cdots, Q$.

These operators transform under the enhanced global symmetry (A.3) according to the following table:

| | $USp(2Q)_x$ | $USp(2Q)_y$ | $U(1)_c$ | $U(1)_t$ | $U(1)_{R_0}$ |
|---|---|---|---|---|---|
| $O_H$ | $\mathbf{Q(2Q-1)-1}$ | $\mathbf{1}$ | $0$ | $-1$ | $2$ |
| $C$ | $\mathbf{1}$ | $\mathbf{Q(2Q-1)-1}$ | $0$ | $-1$ | $2$ |
| $\Pi$ | $\mathbf{2Q}$ | $\mathbf{2Q}$ | $1$ | $0$ | $0$ |

$$\tag{A.7}$$

It is also useful to write explicitly the expression for the supersymmetric index of $FE[USp(2Q)]$. Using the conventions reviewed in Appendix D, this reads

$$
\begin{aligned}
\mathcal{I}_{FE[USp(2Q)]}(x_n; y_n; c, t) = {} & \Gamma_e\left(pqc^{-2}\right)\Gamma_e\left(pqt^{-1}\right)^{Q-1}\prod_{n<m}^{Q}\Gamma_e\left(pqt^{-1}x_n^{\pm 1}x_m^{\pm 1}\right)\prod_{n=1}^{Q}\Gamma_e\left(c\,y_Q^{\pm 1}x_n^{\pm 1}\right) \times \\
& \times \oint d\vec{u}_{Q-1} \frac{\mathcal{I}_{FE[USp(2(Q-1))]}(u_1, \cdots, u_{Q-1}; y_1, \cdots, y_{Q-1}; t^{-1/2}c, t)}{\Gamma_e(t)\prod_{a<b}^{Q-1}\Gamma_e\left(u_a^{\pm 1}u_b^{\pm 1}\right)\prod_{a=1}^{Q-1}\Gamma_e\left(u_a^{\pm 2}\right)} \times \\
& \times \prod_{a=1}^{Q-1}\prod_{n=1}^{Q}\Gamma_e\left(t^{1/2}u_a^{\pm 1}x_n^{\pm 1}\right)\prod_{a=1}^{Q-1}\Gamma_e\left(pqt^{-1/2}c^{-1}y_Q^{\pm 1}u_a^{\pm 1}\right),
\end{aligned}
\tag{A.8}
$$

where the integration measure is

$$d\vec{z}_n = \frac{[(p;p)(q;q)]^n}{2^n n!}\prod_{i=1}^{n}\frac{dz_i}{2\pi i\, z_i}. \tag{A.9}$$

To define the index we use the assignment of R-charges and the parametrization of the abelian global symmetries as summarized in Figure 5. A crucial observation made in [18] is that this supersymmetric index, as an integral function, coincides with the *interpolation kernel* of [34]. Thanks to this connection, many of the non-trivial IR properties of $FE[USp(2Q)]$ can be strongly validated using the various integral identities proved in [34] for this interpolation kernel function.

One of such properties that we used in the main text to construct cap and sphere models starting from the tube model is the dynamics triggered by a linear superpotential term for the antisymmetric operators $O_H$ and $C$. A large class of these deformations have been studied in [19]. In this paper, we are interested in the one that breaks one of the two $USp(2Q)$ symmetries to the smallest possible subgroup, namely $SU(2)$. For example, the following linear superpotential involving the operator $O_H$

$$\delta\mathcal{W} = J_Q O_H, \tag{A.10}$$

where

$$J_Q = \frac{i\sigma_2}{2} \otimes \left(J_Q + J_Q^T\right) \tag{A.11}$$

and $J_Q$ is the Jordan matrix of dimension $Q$, breaks the flavor symmetry $\mathrm{USp}(2Q)_x$ down to $\mathrm{SU}(2)_v$. Notice that since $O_H$ is flipping the operator $H$, the F-term equations for $O_H$ now imply the following vev:

$$\langle H \rangle = J_Q \, . \tag{A.12}$$

This vev makes the original $FE[\mathrm{USp}(2Q)]$ theory flow to a simple WZ model. This can be understood from Corollary 2.8 of [34]

$$
\begin{aligned}
\mathcal{I}_{FE[\mathrm{USp}(2Q)]}(v, t\,v, \cdots, t^{Q-1}v; y_n; c, t) &= \prod_{n=2}^{Q} \frac{1}{\Gamma_e(t^n)} \prod_{n=1}^{Q} \frac{\Gamma_e\left(v\,c\,y_n^{\pm 1}\right)\Gamma_e\left(v^{-1}c\,t^{1-Q}y_n^{\pm 1}\right)}{\Gamma_e(t^{1-n}c^2)} \\
&\rightarrow \prod_{n=2}^{Q} \frac{1}{\Gamma_e(t^n)} \prod_{n=1}^{Q} \frac{\Gamma_e\left(c\,t^{\frac{1-Q}{2}}v^{\pm 1}y_n^{\pm 1}\right)}{\Gamma_e(c^2 t^{1-n})} \, ,
\end{aligned}
\tag{A.13}
$$

where at the second step we redefined $v \rightarrow t^{\frac{1-Q}{2}}v$ to make the residual $\mathrm{SU}(2)_v$ symmetry manifest. As it was pointed out in [19], some of the fields in this theory are decoupled from the others and were removed in that paper with a flipping procedure. We instead keep all of them in the construction of the E-string models, as they are crucial to match the anomalies from 6d computed in Appendix B. Indeed, there is a priori no reason why the compactification of the 6d theory should lead to a fully interacting theory without a decoupled free sector.

# B  't Hooft Anomalies for the E-string theory from 6d

In this appendix we derive the predicted anomalies from 6d for the 4d models obtained compactifying the E-string theory on a generic Riemann surface of genus $g$ with $s$ punctures and a general flux for the 6d global symmetry. We also compute the anomalies for the flavor symmetry that descend from the $\mathrm{SU}(2)$ isometry in the case in which the Riemann surface is $\mathbb{S}^2$.

We start by writing the 6d anomaly polynomial eight-form for a rank $Q$ E-string theory as given in [8],

$$
\begin{aligned}
I_8^{E-string} &= \frac{Q\left(4Q^2+6Q+3\right)}{24}C_2^2(R)_{\mathbf{2}} + \frac{(Q-1)\left(4Q^2-2Q+1\right)}{24}C_2^2(L)_{\mathbf{2}} \\
&\quad - \frac{Q\left(Q^2-1\right)}{3}C_2(R)_{\mathbf{2}}\,C_2(L)_{\mathbf{2}} + \frac{(Q-1)(6Q+1)}{48}C_2(L)_{\mathbf{2}}\,p_1(T) \\
&\quad - \frac{Q(6Q+5)}{48}C_2(R)_{\mathbf{2}}\,p_1(T) + \frac{Q(Q-1)}{120}C_2(L)_{\mathbf{2}}\,C_2(E_8)_{\mathbf{248}} \\
&\quad - \frac{Q(Q+1)}{120}C_2(R)_{\mathbf{2}}\,C_2(E_8)_{\mathbf{248}} + \frac{Q}{240}p_1(T)\,C_2(E_8)_{\mathbf{248}} \\
&\quad + \frac{Q}{7200}C_2(E_8)_{\mathbf{248}}^2 + (30Q-1)\frac{7p_1(T)-4p_2(T)}{5760} \, ,
\end{aligned}
\tag{B.1}
$$

where $C_n(G)_{\mathbf{R}}$ is the $n$-th Chern class of the group $G$ in the representation $\mathbf{R}$. In particular, $C_2(R)_{\mathbf{2}}$ and $C_2(L)_{\mathbf{2}}$ stand for the second Chern class of $SU(2)_R$ and $SU(2)_L$, respectively, in the doublet representation. Moreover, $p_1(T)$ and $p_2(T)$ denote the first and second Pontryagin classes of the tangent bundle $T$.

We wish to write the anomalies under the decomposition

$$
E_8 \rightarrow E_7 \times U(1)_c \rightarrow SU(8)_u \times U(1)_c \rightarrow \prod_{a=1}^{8} U(1)_{u_a} \times U(1)_c \, ,
\tag{B.2}
$$

with the constraint that $U(1)_{u_8} = -\sum_{a=1}^{7} U(1)_{u_a}$. Thus, we first want to decompose the $E_8$ Chern classes to the Chern classes of $E_7$ and $U(1)_c$. Using the decomposition

$$\mathbf{248} \to (\mathbf{133}) \oplus (\mathbf{56})\left(c + c^{-1}\right) \oplus \left(c^2 + 1 + c^{-2}\right), \tag{B.3}$$

we can substitute the following Chern classes

$$C_2(E_8)_{\mathbf{248}} = -60\, C_1^2(c) + C_2\left(E_7\right)_{\mathbf{133}} + 2\, C_2\left(E_7\right)_{\mathbf{56}}. \tag{B.4}$$

Next, we further break $E_7 \to SU(8)_u$ using the branching rules

$$\begin{aligned}
\mathbf{56} &\to (\mathbf{28}) \oplus \left(\overline{\mathbf{28}}\right), \\
\mathbf{133} &\to (\mathbf{63}) \oplus (\mathbf{70}).
\end{aligned} \tag{B.5}$$

Translating to the Chern classes substitutions

$$\begin{aligned}
C_2\left(E_7\right)_{\mathbf{56}} &= C_2(SU(8))_{\mathbf{28}} + C_2(SU(8))_{\overline{\mathbf{28}}} = 12\, C_2(SU(8))_{\mathbf{8}}, \\
C_2\left(E_7\right)_{\mathbf{133}} &= C_2(SU(8))_{\mathbf{63}} + C_2(SU(8))_{\mathbf{70}} = 36\, C_2(SU(8))_{\mathbf{8}},
\end{aligned} \tag{B.6}$$

where in the second equalities we used the fact that $C_2(G)_{\mathbf{R_1}}/T_G(\mathbf{R_1}) = C_2(G)_{\mathbf{R_2}}/T_G(\mathbf{R_2})$, with $T_G(\mathbf{R})$ standing for the Dynkin index of the representation $\mathbf{R}$ of the group $G$ (a.k.a. quadratic Casimir).

The final step in the decomposition is the one taking $SU(8)_u \to U(1)^7$ with the well known branching rule

$$\mathbf{8} \to \sum_{i=1}^{8} u_a, \tag{B.7}$$

with the constraint $u_8 = \prod_{i=1}^{7} u_i^{-1}$. In terms of Chern classes it translates into

$$C_2(SU(8))_{\mathbf{8}} = -\frac{1}{2}\sum_{a=1}^{8} C_1^2\left(U(1)_{u_a}\right). \tag{B.8}$$

Gathering all the decompositions together, we find that we simply need to substitute

$$C_2(E_8)_{\mathbf{248}} = -60\, C_1^2(c) - 30\sum_{a=1}^{8} C_1^2\left(U(1)_{u_a}\right) \tag{B.9}$$

in the above anomaly polynomial.

In the main text we also consider models with flux in the Cartan of $SU(2)_L$ global symmetry. In this case we should break $SU(2)_L \to U(1)_t$ by setting the Chern classes as follows

$$C_2(L)_{\mathbf{2}} = -C_1^2(t). \tag{B.10}$$

After the above substitutions, we obtain the 6d 8-form anomaly polynomial written in terms of Chern classes for the $U(1)$ symmetries in the Cartan of the 6d global symmetry, including the R-symmetry. The next step consists of compactifying the theory on a Riemann surface $C_g$ of genus $g$. At the level of the Pontryagin classes this means

$$p_1(T_{6d}) = p_1(T_{4d}) + e^2, \qquad p_2(T_{6d}) = p_2(T_{4d}) + p_1(T_{4d})e^2, \tag{B.11}$$

where $e$ is the Euler class of the Riemann surface. We also want to turn on fluxes $n_t$ for $U(1)_t$, $n_c$ for $U(1)_c$ and $n_a$ for $U(1)_{u_a}$ through the Riemann surface, meaning

$$C_1^{6d}(t) = C_1^{4d}(t) - n_t \frac{e}{2(1-g)}$$

$$C_1^{6d}(c) = C_1^{4d}(c) - n_c \frac{e}{2(1-g)}$$

$$C_1^{6d}(u_a) = C_1^{4d}(u_a) - n_a \frac{e}{2(1-g)}, \tag{B.12}$$

where the Chern classes for the SU(8) Cartan satisfy $C_1^{4d}(u_8) = -\sum_{a=1}^{7} C_1^{4d}(u_a)$. For the R-symmetry we don't turn on any flux, but we have to consider that, in order to preserve half of the supercharges in the compactification, we need to perform a topological twist, which amounts to

$$C_2^{6d}(R)_{\mathbf{2}} = -C_1^{6d}(R)^2, \qquad C_1^{6d}(R) = C_1^{4d}(R) - \frac{e}{2}. \tag{B.13}$$

When we compactify our theory on a generic Riemann surface, only the terms linear in $e$ out of the full 8-form anomaly polynomial contribute and their contribution can be computed using the Gauss–Bonnet theorem

$$\int_{C_g} e = 2(1-g). \tag{B.14}$$

In this way we get the 6-form anomaly polynomial $I_6$ of the 4d theory, out of which we can read all the anomalies for the 4d global symmetries that descend from 6$d$

$$\text{Tr}\, U(1)_i = -24 I_6\big|_{C_1^{4d}(U(1)_i)p_1(T_{4d})}, \quad \text{Tr}\, U(1)_i U(1)_j U(1)_k = d_{ijk} I_6\big|_{C_1^{4d}(U(1)_i)C_1^{4d}(U(1)_j)C_1^{4d}(U(1)_k)}, \tag{B.15}$$

where $d_{ijk} = m!$ with $m$ the number of equal indices between $i$, $j$ and $k$. Eventually we find the following 4d anomalies:

$$\text{Tr}\left(U(1)_R^3\right) = (g-1)Q\left(4Q^2 + 6Q + 3\right), \quad \text{Tr}(U(1)_R) = -(g-1)Q(6Q+5),$$

$$\text{Tr}\left(U(1)_c^3\right) = -12Qn_c, \quad \text{Tr}(U(1)_c) = -12Qn_c,$$

$$\text{Tr}\left(U(1)_{u_a}^3\right) = -6Q(n_a - n_8), \quad \text{Tr}\left(U(1)_{u_a}\right) = -6Q(n_a - n_8),$$

$$\text{Tr}\left(U(1)_R U(1)_c^2\right) = -2(g-1)Q(Q+1), \quad \text{Tr}\left(U(1)_R^2 U(1)_c\right) = 2Q(Q+1)n_c,$$

$$\text{Tr}\left(U(1)_R U(1)_{u_a}^2\right) = -2(g-1)Q(Q+1), \quad \text{Tr}\left(U(1)_R^2 U(1)_{u_a}\right) = Q(Q+1)(n_a - n_8),$$

$$\text{Tr}\left(U(1)_R U(1)_{u_a} U(1)_{u_b}\right) = -(g-1)Q(Q+1),$$

$$\text{Tr}\left(U(1)_c U(1)_{u_a}^2\right) = -4Qn_c, \quad \text{Tr}\left(U(1)_c^2 U(1)_{u_a}\right) = -2Q(n_a - n_8),$$

$$\text{Tr}\left(U(1)_{u_a} U(1)_{u_b}^2\right) = -2Q(n_a + n_b - 2n_8), \quad \text{Tr}\left(U(1)_c U(1)_{u_a} U(1)_{u_b}\right) = -2Qn_c,$$

$$\text{Tr}\left(U(1)_{u_a} U(1)_{u_b} U(1)_{u_d}\right) = -Q(n_a + n_b + n_d - 3n_8),$$

$$\text{Tr}\left(U(1)_t^3\right) = -(Q-1)\left(4Q^2 - 2Q + 1\right)n_t, \quad \text{Tr}(U(1)_t) = -(Q-1)(6Q+1)n_t,$$

$$\text{Tr}\left(U(1)_R U(1)_t^2\right) = -\frac{4}{3}(g-1)Q\left(Q^2-1\right), \quad \text{Tr}\left(U(1)_R^2 U(1)_t\right) = \frac{4}{3}Q\left(Q^2-1\right)n_t,$$

$$\text{Tr}\left(U(1)_t U(1)_{c/u_a}^2\right) = -2Q(Q-1)n_t, \quad \text{Tr}\left(U(1)_t^2 U(1)_c\right) = -2Q(Q-1)n_c,$$

$$\text{Tr}\left(U(1)_t^2 U(1)_{u_a}\right) = -Q(Q-1)(n_a - n_8), \quad \text{Tr}\left(U(1)_t U(1)_{u_a} U(1)_{u_b}\right) = -Q(Q-1)n_t, \tag{B.16}$$

with the constraint $n_8 = -\sum_{a=1}^{7} n_a$. The rest of the anomalies that don't appear in (B.16) vanish. Moreover, the 6d R-symmetry used in (B.16) and the 4d R-symmetry used in the main text are related by $R_{6d} = R_{4d} + q_t$, where $q_t$ denotes the charge under $U(1)_t$.

When we compactify the E-string theory on a Riemann surface with punctures the anomalies in (B.16) should be modified by $g \to g + \frac{s}{2}$, where $s$ is the number of punctures. Moreover, we have additional contributions coming from the punctures. These have been already worked out in [8] for the rank one case and in [18] for the case of generic rank $Q$ and here we shall only report the final result. For each maximal puncture we have the following contributions to the anomalies:

$$
\begin{aligned}
\text{Tr}(U(1)_R^3) &= -\frac{Q(2Q+1)}{2}, & \text{Tr}(U(1)_R) &= -\frac{Q(2Q+1)}{2}, \\
\text{Tr}(U(1)_t^3) &= q^3 \frac{(Q(2Q-1)-1)}{2}, & \text{Tr}(U(1)_t) &= q \frac{(Q(2Q-1)-1)}{2}, \\
\text{Tr}(U(1)_R \text{USp}(2Q)^2) &= -\frac{(Q+1)}{2}, & \text{Tr}(U(1)_t \text{USp}(2Q)^2) &= q \frac{(Q-1)}{2}. \\
\text{Tr}(U(1)_c^3) = Q \sum_{a=1}^{8} q_a^3, & \quad \text{Tr}(U(1)_c) = Q \sum_{a=1}^{8} q_a, & \quad \text{Tr}(U(1)_c \text{USp}(2Q)^2) = \frac{1}{4} \sum_{a=1}^{8} q_a \\
\text{Tr} U(1)_c U(1)_{u_a}^2 = 2Q q_a & \quad a = 1,\cdots,7, & \quad \text{Tr} U(1)_c U(1)_{u_a} U(1)_{u_b} = Q q_a & \quad a \neq b \\
\text{Tr} U(1)_{u_a} U(1)_{u_b} U(1)_{u_d} = Q, & \quad a \neq b, d\,,
\end{aligned}
$$
(B.17)

where $q$ and $q_a$ are normalization factors that can be fixed when trying to match with the 4d anomalies of the models discussed in the main text. In particular in our conventions we should set $q = 1$ and $q_a = -\frac{1}{2}$. The anomalies that don't appear in (B.17) receive no contribution from the punctures.

When the Riemann surface possesses some isometry, this manifests itself as a flavor symmetry from the point of view of the 4d theory. We can then compute the anomalies for such a symmetry starting from the 8-form anomaly polynomial of the 6d theory [45] and for this we will follow the discussion of [35] (See [46–48] for earlier application in physics.). In the main text we are interested in the case in which the surface is a two-sphere, whose isometry group is $SO(3)_{\text{ISO}} \simeq SU(2)_{\text{ISO}}$. The anomalies for this symmetry can be computed following the same procedure that lead us to (B.16) by just taking into account that, from the 8-form anomaly polynomial, we now receive contributions also from terms which are cubic in the Euler class $e$. This follows from the fact that now $e$ fibers the surface in a non-trivial way over the 4d space. Their contribution can be computed using the Bott–Cattaneo formula [23], which in general states that

$$
\int_{\mathbb{S}^2} e^{2s+1} = 2 p_1(SO(3)_{\text{ISO}})^s, \qquad \int_{\mathbb{S}^2} e^{2s} = 0\,,
$$
(B.18)

where $p_1(SO(3)_{\text{ISO}})$ is the first Pontryagin class of the real vector bundle for the $SO(3)_{\text{ISO}}$ isometry of $\mathbb{S}^2$. In particular, for $s = 0$ we recover the usual Gauss–Bonnet theorem (B.14) for the two-sphere, while for $s = 1$ we get the formula that we need for evaluating the integral of $e^3$

$$
\int_{\mathbb{S}^2} e^3 = -2 C_2(SU(2)_{\text{ISO}})_3\,,
$$
(B.19)

where we used that $p_1(SO(3)_{\text{ISO}}) = -C_2(SU(2)_{\text{ISO}})_3$ when we think of $SO(3)_{\text{ISO}} \simeq SU(2)_{\text{ISO}}$. In this way we get additional terms in the 6-form anomaly polynomial of the 4d theory that

are proportional to $C_2(\mathrm{SU}(2)_{\mathrm{ISO}})_\mathbf{3}$, from which we can read off the anomalies for the 4d flavor symmetry $\mathrm{SU}(2)_{\mathrm{ISO}}$. Using the fact that

$$\mathrm{Tr}\left(G^2\,\mathrm{U}(1)\right) = -T_G(\mathbf{R})I_6\big|_{C_2(G)_\mathbf{R} C_1(\mathrm{U}(1))}, \tag{B.20}$$

and that the Dykin index of the adjoint representation of $\mathrm{SU}(2)_{\mathrm{ISO}}$ is 2, we find

$$\mathrm{Tr}\left(\mathrm{SU}(2)_{\mathrm{ISO}}^2\mathrm{U}(1)_R\right) = \frac{Q(Q+1)}{12}\left(-8 + 6n_c^2 + 4(Q-1)n_t^2 - 4Q + 3\sum_{a=1}^{8} n_a^2\right)$$

$$\mathrm{Tr}\left(\mathrm{SU}(2)_{\mathrm{ISO}}^2\mathrm{U}(1)_t\right) = -\frac{Q-1}{12}n_t$$

$$\times\left(-1 + 2Q(3n_c^2 - 5) + (4Q^2 - 2Q + 1)n_t^2 + Q(-4Q + 3\sum_{a=1}^{8} n_a^2)\right)$$

$$\mathrm{Tr}\left(\mathrm{SU}(2)_{\mathrm{ISO}}^2\mathrm{U}(1)_c\right) = -\frac{Q}{2}n_c\left(-3 + 2n_c^2 + (Q-1)n_t^2 - Q + \sum_{a=1}^{8} n_a^2\right)$$

$$\mathrm{Tr}\left(\mathrm{SU}(2)_{\mathrm{ISO}}^2\mathrm{U}(1)_{u_a}\right) = -\frac{Q}{4}(n_a - n_8)\left(-3 + 2n_c^2 + (Q-1)n_t^2 - Q + \sum_{a=1}^{8} n_a^2\right). \tag{B.21}$$

Note that the anomalies of $\mathrm{SU}(2)_{\mathrm{ISO}}$ depend on fluxes qualitatively in a different way than the ones for other symmetries. Namely these are non-linear in the fluxes. In particular this implies that the correct relation of the $U(1)_f$ symmetry to the Cartan of $\mathrm{SU}(2)_{\mathrm{ISO}}$ depends on the fluxes.

We would like to give a prediction for the mixing of the $\mathrm{U}(1)_f$ symmetry appearing in (4) with fugacity $f$ and other U(1) symmetries. Such mixing is required when generating a flux sphere in order to get an enhancement of symmetry to the isometry symmetry of a sphere given by $\mathrm{SU}(2)_{\mathrm{ISO}}$. Finding the mixing can be done in an algorithmic fashion by comparing the anomalies involving a generally mixed $\mathrm{U}(1)_f$ with the ones predicted from 6d given in (B.21). Building a sphere involves generally two caps and a series of tubes in glued in between them. We will fix the initial isometry $\mathrm{U}(1)_f$ symmetry on one of the caps to be the one naturally derived from the tube in (4) and set the rest of the $\mathrm{U}(1)_f$ symmetries of the tubes and the other cap to mix with the other U(1)'s such that all the mixed gauge anomalies vanish. In this construction the mixing is given by

$$q_f^{mixed} = q_f + (n_t + 1)q_t + (n_c - 1)q_c + \sum_{i=1}^{7}\left(n_{a_i} - \frac{1}{4}\right)q_{a_i}, \tag{B.22}$$

where $q_m$ denotes the charge of a chiral multiplet under $\mathrm{U}(1)_m$, and $n_m$ denote the flux of the sphere under $\mathrm{U}(1)_m$. One can see that all the expected examples in Section 3 obey this mixing prescription.

## C   $\mathrm{SU}(2)_{\mathbf{ISO}}$ symmetry of $\mathbb{S}^2$ compactifications of $\mathcal{N}=1$ class $\mathcal{S}$

In this appendix we use the machinery of [35] that we have just used for the E-string theory, but to compute the anomalies in sphere compactifications of $\mathcal{N}=1$ class $\mathcal{S}$ theories [2, 3, 49] that involve the isometry of the $\mathbb{S}^2$[20]. We will then use this result to check, in type $A_1$ the case in which the 4d theory is Lagrangian, that there is an additional flavor symmetry that can be identified with such isometry.

---

[20]The computation starting from string theory was previously performed in [45].

The starting point is the anomaly polynomial of the 6d $\mathcal{N} = (2,0)$ $A_{N-1}$ SCFT [46,50]

$$I_8 = \frac{N-1}{48}\left[ p_2(N) - p_2(T) + \frac{1}{4}(p_1(T) - p_1(N))^2 \right] + \frac{N(N^2-1)}{24} p_2(N). \tag{C.1}$$

The theory has a symmetry group $SO(5,1) \times SO(5)_R$, with the $SO(5,1)$ Lorentz symmetry acting on the tangent bundle and the $SO(5)_R$ R-symmetry acting on the normal bundle. It is possible to compactify the theory on a Riemann surface with a topological twist preserving either $\mathcal{N} = 2$ or $\mathcal{N} = 1$ supersymmetry in 4d. Here we will focus on the latter. Specifically, we decompose $SO(3,1) \times SO(2)_s \subset SO(5,1)$ and $U(1)_R \times SU(2)_L \subset SU(2)_R \times SU(2)_L \simeq SO(4) \subset SO(5)_R$ and perform a twist that mixes $U(1)_R$ with the spin connection over the Riemann surface $SO(2)_s$. This leads to a 4d $\mathcal{N} = 1$ theory with a flavor symmetry $SU(2)_L$. We are going to show that when the Riemann surface is a sphere, the resulting 4d theory possesses an additional $SU(2)$ flavor symmetry coming from the isometry group of the $\mathbb{S}^2$.

The anomaly polynomial of the 4d $\mathcal{N} = 1$ class $\mathcal{S}$ theory of type $A_{N-1}$ was computed integrating the 6d anomaly polynomial (C.1) over the Riemann surface in [2]. Here we are going to extend their computation for the case of the two-sphere by including its isometry group $SU(2)_{ISO}$ and also a possible flux for the Cartan of the $SU(2)_L$ symmetry. Remembering the decomposition of the $SO(5)_R$ R-symmetry for the $\mathcal{N} = 1$ twist, we can rewrite the Pontryagin classes of the normal bundle as

$$p_1(N) = -2\left(C_2(R)_{\mathbf{2}} + C_2(R)_{\mathbf{2}}\right) = 2\left(C_1(R)^2 + C_1(L)^2\right),$$
$$p_2(N) = \left(C_2(R)_{\mathbf{2}} - C_2(L)_{\mathbf{2}}\right)^2 = C_1(R)^4 + C_1(L)^4 - 2C_1(R)^2 C_1(L)^2. \tag{C.2}$$

When we compactify on the sphere, the Pontryagin classes of the tangent bundle decompose as

$$p_1(T_{6d}) = p_1(T_{4d}) + e^2, \qquad p_2(T_{6d}) = p_2(T_{4d}) + p_1(T_{4d})e^2, \tag{C.3}$$

where $e$ is the Euler class of the sphere. As in the case of the E-string theory, the topological twist can be implemented by

$$C_2^{6d}(R)_{\mathbf{2}} = -C_1^{6d}(R)^2, \qquad C_1^{6d}(R) = C_1^{4d}(R) - \frac{e}{2}, \tag{C.4}$$

while the flux $n_L$ for $U(1)_L \subset SU(2)_L$ can be implemented by

$$C_2^{6d}(L)_{\mathbf{2}} = -C_1^{6d}(L)^2, \qquad C_1^{6d}(L) = C_1^{4d}(L) - n_L \frac{e}{2(1-g)}. \tag{C.5}$$

After these replacements, we can integrate over the surface. As we saw in the previous appendix for the case of the E-string, when we integrate over $\mathbb{S}^2$ we receive contributions to the resulting 4d 6-form anomaly polynomial not only from terms linear in $e$, but also from cubic terms. For the former we use the standard Gauss–Bonnet formula (B.14), while for the latter we use the Bott–Cattaneo formula (B.19). This gives us the following 6-form anomaly polynomial for the 4d $\mathcal{N} = 1$ class $\mathcal{S}$ theory of type $A_{N-1}$ on $\mathbb{S}^2$ in the presence of flux $n_L$ for $U(1)_L$ using (B.20)

$$\begin{aligned}
I_6 = \frac{1}{24}\Big( & 4N(N^2-1)n_L C_1(L)C_1(R)^2 - 4(N^3-1)C_1(R)^3 + \\
+ & (N-1)C_1(R)\big(p_1(T) + N(N+1)\big(4C_1(L)^2 - (n_L^2-1)C_2(SU(2)_{ISO})_{\mathbf{3}}\big)\big) + \\
- & n_L C_1(L)\big((1-N)p_1(T) + (N^3-1)\big(4C_1(L)^2 - (n_L^2-1)C_2(SU(2)_{ISO})_{\mathbf{3}}\big)\big)\Big).
\end{aligned} \tag{C.6}$$

From this we can, for example, extract the anomalies for the $SU(2)_{ISO}$ symmetry with $U(1)_R$ and $U(1)_L$

$$\text{Tr}\left(SU(2)_{ISO}^2 U(1)_R\right) = \frac{N(N^2-1)}{12}(n_L^2 - 1), \qquad \text{Tr}\left(SU(2)_{ISO}^2 U(1)_L\right) = -\frac{N^3-1}{12}n_L(n_L^2 - 1).$$
(C.7)

Notice that the second anomaly vanishes in the case of no flux $n_L = 0$, since the $SU(2)_L$ symmetry is fully preserved.

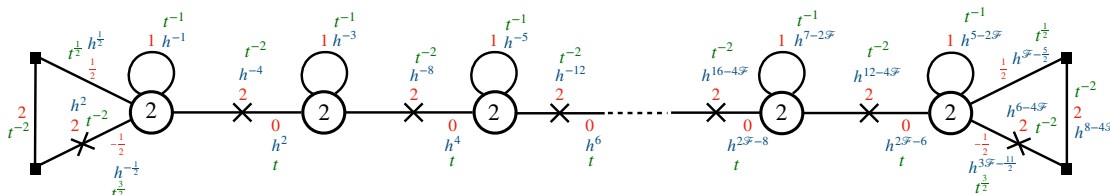

Figure 6: The Lagrangian for the sphere with $n_L = \mathcal{F} > 2$ for the case of $N = 2$. The number of gauge nodes is $\mathcal{F} - 2$. Here $t$ is the fugacity for the Cartan of $SU(2)_L$ and the 6d R-charges are written in red. The fugacity $h$ is for an additional $U(1)$ symmetry that will be related to $SU(2)_{ISO}$ in the text. One should turn on all the gauge invariant superpotentials consistent with the charges denoted in the quiver. The $x$s denote gauge singlet fields flipping certain gauge invariant operator.

Let us compare the above anomalies to the explicit field theoretic construction in the case of $N = 2$. Here the theories corresponding to a sphere with $n_L = \mathcal{F} > 2$ are given by the gauge theories of Figure 6, see [30, 51–53]. These are built from two caps, the triangles at the ends, and tube theories, the bifundamentals with a flip. The tube is obtained from the trinion, the trifundamental of $SU(2)$, by flipping one of the three moment maps and then closing the puncture giving a vev to one of the flip fields. The trinion has flux $+\frac{1}{2}$ and the procedure shifts the flux by $+\frac{1}{2}$ to give $n_L = +1$ for the tube. The cap is obtained similarly by flipping another puncture moment map and closing and one obtains flux $+\frac{3}{2}$. One can compute the index for these theories and find that for general value of flux $\mathcal{F}$ the index is invariant under the Weyl group of $SU(2)$ Cartan of which is parametrized by fugacity $h$ taking the shift $t \to t h^{2-\mathcal{F}}$. The Weyl symmetry acts on the index as $h \to 1/h$ after performing the shift. Moreover at order $qp$ of the index one observes a contribution of the form $-1-1-h^2-h^{-2}$ which is to be identified with the contribution of the conserved currents of $U(1)_L \times SU(2)$ [38][21]. We identify this $SU(2)$ with $SU(2)_{ISO}$ with the fugacity $h$ as corresponding to the Cartan. Then we can compute the anomalies for the $SU(2)_{ISO}$ symmetry, or rather for the Cartan as it is the symmetry appearing explicitly in the Lagrangian. The results are given by,

$$\frac{1}{4}\text{Tr}\,U(1)_h^2 R = \frac{1}{2}(\mathcal{F}^2 - 1), \qquad \frac{1}{4}\text{Tr}\,U(1)_h^2 U(1)_L = \frac{7}{12}\mathcal{F}(1 - \mathcal{F}^2), \qquad (C.8)$$

$$\text{Tr}\,U(1)_h R^2 = 0, \qquad \text{Tr}\,U(1)_h U(1)_L^2 = 0, \qquad \text{Tr}\,U(1)_h = 0, \qquad \text{Tr}\,U(1)_h^3 = 0.$$

---

[21]Notice that these theories have also states with 6d R-charge $-1$ which come from operators winding from end to end on the quiver. Such states also exist in theories corresponding to E-string on the sphere we discussed in the bulk of the paper. The charges of these states with respect to abelian symmetries scale with the fluxes. It will be interesting to understand the 6d origin of such states and a natural guess is that these come from defects wrapping the surface. A similar conjecture was put forward regarding states with 6d R-charge zero for tori compactifications [8].

This is consistent with $U(1)_h$ being the Cartan of $SU(2)_{ISO}$ and with the anomalies for this symmetry computed from six dimensions in (C.7)[22].

For flux $n_L = 2$ the theory in 4d is a WZ model with manifest $SU(2)$ symmetry which we identify with $SU(2)_{ISO}$ [30]. This theory is obtained by flipping the puncture of the cap and closing. The matter content is a doublet $Q$ of $SU(2)$ (6d R-charge 0 and $U(1)_L$ charge $+1$), an adjoint $\Phi$ (6d R-charge $+2$ and $U(1)_L$ charge $-2$), and a singlet $\phi$ (6d R-charge $-1$ and $U(1)_L$ charge $+2$). The superpotential is $Q^2\Phi + \phi^2 \operatorname{Tr}\Phi^2$. With these charges we easily compute the anomalies,

$$\operatorname{Tr}\left(SU(2)_{ISO}^2 U(1)_R\right) = \frac{1}{2} \times (0-1) + 2 \times (2-1) = \frac{3}{2}, \tag{C.9}$$
$$\operatorname{Tr}\left(SU(2)_{ISO}^2 U(1)_L\right) = \frac{1}{2} \times (+1) + 2 \times (-2) = -\frac{7}{2}.$$

This again matches perfectly with (C.7) for $n_L = 2$.

Finally for flux $n_L = 1$ the theory is constructed by closing a puncture of the cap without flipping. This shifts the flux $+\frac{3}{2}$ of the cap by $-\frac{1}{2}$ and we obtain sphere with $n_L = +1$. Performing this closure we obtain just two chiral fields with charges which can be encoded in fugacities as $\frac{pq}{t^2}h^2$ and $(pq)^{-\frac{1}{2}}th^{-1}$. Note that performing here the shift $t \to t\,h^{2-n_L}$ to identify $h$ with the Cartan of the $SU(2)_{ISO}$ the dependence on $h$ drops out completely. In particular this implies that anomalies of $SU(2)_{ISO}$ vanish consistently with (C.7) for $n_L = 1$. Moreover one can check other anomalies to match with (C.6): *e.g.* $\operatorname{Tr} R^3 = -7$ and $\operatorname{Tr} R = -1$. Note that having flux $+1$ for a sphere we expect to preserve $\mathcal{N} = 2$ supersymmetry. The reason is that the $\mathcal{N} = 2$ supersymmetric twist on a closed surface of genus $g$ corresponds to flux $\pm(g-1)$. $\mathcal{N} = 2$ sphere with no punctures is subtle, see [55], but the procedure discussed here provides a concrete suggestion for the model. The above charges seem to be unconventional but upon closer inspection we can recognize that this is just a free hypermultiplet. To see this let us transform the charges after the shift to the usual $\mathcal{N} = 2$ notations[23]. This is achieved here by further shifting $t \to (qp)^{\frac{1}{2}} t^{-1}$[24]. Doing so the two fields have charges $t^{\frac{1}{2}} \times t^{\frac{3}{2}}$ and $t^{\frac{1}{2}} \times t^{-\frac{3}{2}}$, which are just the charges of a free hypermultiplet with $SU(2)$ global symmetry parametrized by $t^{\frac{3}{2}}$. In other words we get a free hypermultiplet with emergent $SU(2)$ symmetry rotating the half hypermultiplets, Cartan of which in the compactification procedure is locked with $U(1)_L$[25].

It would be interesting to study the emergent $SU(2)_{ISO}$ symmetry for higher values of $N$. *E.g.* for $N = 3$ one might try to use the Lagrangian for $T_3$ of [57] or the construction of [58]. It will be also interesting to understand any interplay the $SU(2)_{ISO}$ symmetry has here, in the rank $Q$ E-string and any other 6d compactifications, with the higher form symmetries through the higher group structures, see [59–62].

---

[22]It is interesting to note that the $SU(2)_{ISO}$ symmetry further enhances to $SU(n_L)$ upon reduction to 3d [30]. The claim is that in 3d one has a mirror dual in terms of $SO(3)$ gauge theory with $n_L$ adjoint fields rotated by $SU(n_L)$ [30,54].

[23]Remember that using 6d R-symmetry the free hypermultiplet has R-charge $+\frac{1}{2}$ and the chiral adjoint in $\mathcal{N} = 2$ vector has R-charge $+1$. While for the computations of $\mathcal{N} = 2$ index one uses assignment of R-charge 0 to hypermultiplet and $+2$ to the adjoint chiral, see *e.g.* [56].

[24]The negative power of $t$ here is due to the fact that the flux is $+1 = 1 - g$ while for the trinion from which the construction starts the flux is $+\frac{1}{2} = g - 1 + \frac{s}{2}$.

[25]Note that upon compactification to 3d this model is supposed to be mirror dual to an $SO(3)$ gauge thory with an adjoint chiral [30], which is an $\mathcal{N} = 4$ pure $SO(3)$ gauge theory. This theory is "bad" in the sense that the partition functions are divergent/ superconformal R-symmetry is not identifiable in the UV. The claim here thus suggests that pure $SO(3)$ $\mathcal{N} = 4$ SYM flows to a free hypermultiplet in the IR.

# D  Index notations

In this appendix we define the $\mathcal{N} = 1$ superconformal index [63–65]. In addition we give some useful notations and results that were used throughout this paper. See [66] for more details. We define an SCFT index by the Witten index of the theory in radial quantization. In 4d it translates to a trace over the Hilbert space of the theory quantized on $\mathbb{S}^3$,

$$\mathcal{I}(\mu_i) = Tr(-1)^F e^{-\beta\delta} e^{-\mu_i \mathcal{M}_i}, \tag{D.1}$$

where $\delta = \frac{1}{2}\left\{\mathcal{Q}, \mathcal{Q}^\dagger\right\}$, with $\mathcal{Q}$ and $\mathcal{Q}^\dagger = \mathcal{S}$ one of the Poincaré supercharges, and its conjugate conformal supercharge, respectively. $\mathcal{M}_i$ are $\mathcal{Q}$-closed conserved charges and $\mu_i$ their associated chemical potentials. This expression gets non-vanishing contributions only from states with $\delta = 0$ making the index independent on $\beta$.

Focusing on $\mathcal{N} = 1$, the supercharges are $\left\{\mathcal{Q}_\alpha, \mathcal{S}^\alpha = \mathcal{Q}^{\dagger\alpha}, \widetilde{\mathcal{Q}}_{\dot\alpha}, \widetilde{\mathcal{S}}^{\dot\alpha} = \widetilde{\mathcal{Q}}^{\dagger\dot\alpha}\right\}$, with $\alpha = \pm$ and $\dot\alpha = \pm$ the respective $SU(2)_1$ and $SU(2)_2$ indices of the isometry group of $\mathbb{S}^3$ ($Spin(4) = SU(2)_1 \times SU(2)_2$). Since all choices of supercharges give equivalent indices in the setups we discuss, let us explicitly choose $\mathcal{Q} = \widetilde{\mathcal{Q}}_-$. For this choice the index formula transforms to,

$$\mathcal{I}(p,q) = Tr(-1)^F p^{j_1+j_2+\frac{1}{2}r} q^{j_2-j_1+\frac{1}{2}r}, \tag{D.2}$$

where $j_1$ and $j_2$ are the Cartan generators of $SU(2)_1$ and $SU(2)_2$, and $r$ is the generator of the $U(1)_r$ R-symmetry.

The index is computed by listing all gauge invariant contributions one can construct from modes of the fields. The modes and operators are conventionally called "letters" and "words". The single-letter indices for a vector multiplet and a chiral multiplet transforming in the $\mathcal{R}$ representation of the gauge and flavor groups are,

$$
\begin{aligned}
i_V(p,q,U) &= \frac{2pq-p-q}{(1-p)(1-q)}\chi_{adj}(U), \\
i_{\chi(r)}(p,q,U,V) &= \frac{(pq)^{\frac{1}{2}r}\chi_{\mathcal{R}}(U,V) - (pq)^{\frac{2-r}{2}}\chi_{\overline{\mathcal{R}}}(U,V)}{(1-p)(1-q)},
\end{aligned}
\tag{D.3}
$$

where $\chi_{\mathcal{R}}(U,V)$ and $\chi_{\overline{\mathcal{R}}}(U,V)$ denote the characters of representation $\mathcal{R}$ and the conjugate representation $\overline{\mathcal{R}}$, with $U$ and $V$ being the gauge and flavor group matrices, respectively.

The single letter indices allow us to write the full index by listing all the words and projecting them to gauge invariant operators by integrating over the Haar measure of the gauge group. This takes the form

$$\mathcal{I}(p,q,V) = \int [dU] \prod_k PE\left[i_k(p,q,U,V)\right], \tag{D.4}$$

where $k$ labels the different multiplets in the theory, and $PE[i_k]$ is the plethystic exponent of the single-letter index of the $k$-th multiplet. This plethystic exponent lists all the words and is defined by

$$PE\left[i_k(p,q,U,V)\right] \triangleq \exp\left\{\sum_{n=1}^\infty \frac{1}{n} i_k(p^n,q^n,U^n,V^n)\right\}. \tag{D.5}$$

Concentrating on the cases of $SU(N_c)/USp(2N_c)$ groups relevant to our paper the full contribution of a chiral superfield in the fundamental representation with R-charge $r$ can be writ-

ten in terms of elliptic gamma functions, as follows:

$$\text{SU}(N_c) \quad : \quad PE\left[i_k(p,q,U)\right] \equiv \prod_{i=1}^{N_c} \Gamma_e\left((pq)^{\frac{1}{2}r} z_i\right),$$

$$\text{USp}(2N_c) : \prod_{a=1}^{N_c} \Gamma_e\left((pq)^{\frac{1}{2}r} z_a^{\pm 1}\right),$$

$$\Gamma_e(z) \triangleq \Gamma(z;p,q) \equiv \prod_{n,m=0}^{\infty} \frac{1 - p^{n+1}q^{m+1}/z}{1 - p^n q^m z}, \tag{D.6}$$

where $\{z_i\}$ with $i = 1, ..., N_c$ are the fugacities parameterizing the Cartan subalgebra of $\text{SU}(N_c)/\text{USp}(2N_c)$, with $\prod_{i=1}^{N_c} z_i = 1$ for the $\text{SU}(N_c)$ case and no constraint on the $\text{USp}(2N_c)$ case. In addition, in many occasions we will use the shorten notation

$$\Gamma_e\left(uz^{\pm n}\right) = \Gamma_e\left(uz^n\right)\Gamma_e\left(uz^{-n}\right). \tag{D.7}$$

Next we write the full contribution of the vector multiplet in the adjoint of $\text{USp}(2N_c)$, together with the matching Haar measure and integration responsible for projecting to gauge invariant contributions as

$$\frac{\kappa^{N_c}}{2^{N_c}N_c!} \oint_{\mathbb{T}^{N_c}} \prod_{a=1}^{N_c} \frac{dz_a}{2\pi i z_a} \frac{1}{\Gamma_e\left(z_a^{\pm 2}\right)} \prod_{1 \leq a < b \leq N_c} \frac{1}{\Gamma_e\left(z_a^{\pm 1} z_b^{\pm 1}\right)} \cdots, \tag{D.8}$$

where the dots denote that it will be used in addition to the full matter multiplets transforming under the gauge group. The integration is a contour integration over the maximal torus of the gauge group. $\kappa$ is the index of $\text{U}(1)$ free vector multiplet defined as

$$\kappa \triangleq (p;p)(q;q), \tag{D.9}$$

where

$$(a;b) \triangleq \prod_{n=0}^{\infty} (1 - ab^n) \tag{D.10}$$

is the q-Pochhammer symbol.

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
