# Peer review of "Rank $Q$ E-String on Spheres with Flux"

_SciPost Physics, doi:SciPost Phys. 11, 044 (2021)_

## Round 1 · Referee Report · Anonymous · 2021-5-9

Strengths
1 - provides a new construction of a class of theories that opens up potential new vantage points on certain symmetries and features
2 - concrete checks of the proposed construction via detailed computation of anomalies
3 - clarity of the presentation, well written
Weaknesses
1 - it might be useful to add a more concrete discussion on how these results can be used for future computations and new directions
Report
In this paper the authors construct compactifications of the rank $Q$ E-string theory on a sphere with different amounts of flux. These theories can be obtained from a 6-dimensional $\mathcal{N}=(1,0)$ SCFT that is realised by $Q$ M5-branes probing an M9-brane. In this paper the authors present a different construction by gluing together different theories on punctured spheres with specific flux: indeed, starting from the so-called tube model (a theory on a two-punctured sphere) considered previously in the literature (by some of the current authors), the authors create the so-called cap model by 'closing' one of the punctures by introducing suitable vevs and adding gauge invariant flipping fields to obtain the correct anomalies. Compactifications of the E-string theories can be obtained by gluing cap- and tube models via adding chiral fields in a suitable representation of various $Usp(2Q)$ symmetries appearing in the tube models. The authors analyse the symmetry content of the resulting theories and calculate the anomalies to check whether they indeed correspond to the rank $Q$ E-string theories.
As far as I can tell, the computations in this paper are genuine and lead to interesting and novel results: they lead to a new construction of a class of interesting theories that gives a new angle on certain of its properties and symmetries, for example a geometric interpretation of the appearance of an $E_6$ symmetry in a sphere model that was previously discussed in the literature. Furthermore, the paper is well structured and the various appendices make it fairly accessible. Based on this, I recommend it for publication.
There are only very few minor points that perhaps the authors could comment on:
-) it would be helpful to discuss in the main text some of the notation that is used throughout. While there is appendix D, it might help, for example to clarify in eq.(3) which chemical potentials $p$ and $q$ encode
-) after eq.(12): it would be very helpful to elaborate to what degree checking that the two models have the same anomalies guarantees that indeed the theory constructed here is a compactification of the E-string theory, or (if so) which possible ambiguities are left
-) eq.(19): comparing with eq.(12), I don't understand why the argument of the $\Gamma_e$ in the first product contains $x_n$ rather than $y_n$
-) I don't understand how the numerical value of $R_b$ in eq.(63) has been obtained (or similar results in subsequent sections)
-) When gluing various cap- and tube theories together by adding chiral fields, are there any constraints on the choice of $\Phi$- or $S$-gluing? While I understand that in order to reproduce the correct E-string models, a particular choice is required, is it clear that other choices lead to inherently inconsistent models or are there a priori also other viable theories? Does this for example produce inconsistencies at the level of the obtained anomalies, thus providing intrinsic selection rules?
-) The authors have not included a Conclusion section: I agree with their decision that they have (notably in the Introduction) outlined their results and put them into context (such that a Conclusion would be a repetition of what has been already said). However, an aspect which might be interesting to discuss is what other possibilities the description of the E-string theories in the current theory entails: since the tube theory is well studied (and runs to a simple WZW model), does this allow other quantities (i.e. other than anomalies) to be computed which are inaccessible by other methods? Can the current methods be applied to other classes of compactifications?
Author: Chiung Hwang on 2021-05-24 [id 1464]
(in reply to Report 1 on 2021-05-09)We would like to thank the referee for thoughtful and detailed comments that help us to improve the manuscript. We intend to implement the suggestions in the revised version as follows:
We will give a brief introduction to the superconformal index in the main text as well.
The anomaly matching we conducted after eq. (12) is a necessary condition one has to check for the duality between the the 4d model we propose and the 6d E-string theory compactified on the cap. We also provide further evidence that on can construct various 4d models corresponding to the E-string theory on a sphere with different fluxes by gluing the cap models we propose, which exhibit the expected (enhanced) symmetries and the spectrums of operators perfectly consistent with the 6d theory compactified on a sphere with a given flux. We will emphasize this point after eq. (12) in the revised version.
As the referee pointed out, there is a typo in eq. (19), which we will correct in the revised version.
$R_b$ in eq. (63) is obtained by the a-maximization as we explained in the paragraph before eq. (63).
In general, the gluing should be taken in such a way that the resulting theory is anomaly-free. For example, we have considered the S-gluing only for an even number of octet moment maps to avoid the Witten anomaly of the resulting theory. We will emphasize this point in the revised version.
We intend to discuss the possible extension of our result in the introduction section, which would include the generalization to other 6d theories, the study of discrete subgroups of $U(1)_{ISO}$ preserved on a tube, the geometric interpretation of other types of Seiberg dualities, and finally the 3d reduction of our models.

---

## Round 1 · Referee Report · Anonymous · 2021-6-29

Strengths
1 - The paper analyzes the cap model originating from reduction of the rank-Q E-string theory on a one-punctured sphere with suitable fluxes; this cap model is a useful "building block" in the construction of 4d theories from reduction of 6d theories on a Riemann surface.
2 - The paper studies non-trivial applications of this "building block" in the context of the rank-Q E-string theory reduced on a sphere with fluxes.
3 - The constructions of the paper offer a geometric/flux interpretation of a class of IR dualities in field theory.
Weaknesses
1 - It might be useful to add a brief discussion of some future directions of interest.
Report
The paper is well-written and presents original and interesting results in the field. It is thus recommended for publication. Here are some minor comments for the authors.
- I have noticed a few minor typos in the draft: "natrural" on page 5; "bt" for by in footnote 11; "an unitary" between eqs (73) and (74); "transfrom" on page 46.
- It might be useful if the authors comment briefly on why it is appropriate to use rational approximations for the exact R-symmetry mixing coefficients in the analysis of the index, as done e.g. on page 18, and subsequently in other subsections.
- Are the entries of the flux vectors subject to quantization conditions? It might be beneficial to comment briefly on this point.

---

## Round 2 · Referee Report · Anonymous (Referee 2) · 2021-8-9

Report

In this resubmitted version, the authors have carefully implemented the referees' suggestions. The paper is therefore recommended for publication without further modifications.

---

## Round 2 · Referee Report · Anonymous (Referee 1) · 2021-8-12

Report

In this revised version, the authors have fully addressed all the (minor) points that had been raised in the previous reports. I therefore recommend to publish the article in its current form.

---

## Round 2 · Author Response

We would like to thank the referees for thoughtful and detailed comments that help us to improve the manuscript. We have implemented the referees' suggestions in the resubmitted version.

---

## Round 2 · List of Changes

Report I:

  1. We have added the definition of the superconformal index in eq. (3) with a brief explanation of the notation after that.

  2. We have commented in the paragraph after eq. (14) on the evidence we provide to support our cap model, which is supposed to flow to the same IR fixed point as the E-string theory compactified on a sphere with a puncture.

  3. We have corrected the typos the referee pointed out.

  4. $R_b$ in eq. (65), which was (63) in the previous version, is obtained by the a-maximization as we explained in the paragraph before eq. (64).

  5. We have added footnote 14 explaining the gluing should be taken in such a way that the resulting theory is anomaly-free.

  6. We have added discussions for the possible extension of our result in the last paragraph of introduction.

Report II:

  1. We have corrected the typos the referee pointed out.

  2. We have briefly commented after eq. (66) why the rational approximation of the mixing coefficients is used in our analysis.

  3. We have added an explanation for the quantization of the flux in footnote 5, which was footnote 4 in the previous version.

We have also fixed the typos in eqs. (6-7) because we missed $i \sigma_2$ in (6) and the vev $\mathsf{J}_Q$ in (7) is for $\mathsf{H}$ instead of $\mathsf{O_H}$, which is the flip field of $\mathsf{H}$. We have added eq. (5) to elaborate the explanation accordingly.

---

## Editorial Decision

published